# An in-situ peptide-antibody self-assembly to block CD47 and CD24 signaling enhances macrophage-mediated phagocytosis and anti-tumor immune responses

Weiqi Zhang ©[1,2,3], Yinghua Zeng ©[1,3,4], Qiuqun Xiao[3,4], Yuanyuan Wu[5], Jiale Liu[3,4], Haocheng Wang[6], Yuting Luo[2], Jie Zhan ©[7,8,10] ✉, Ning Liao[2,10] ✉ & Yanbin Cai ©[3,4,9,10] ✉

Targeted immunomodulation for reactivating innate cells, especially macrophages, holds great promise to complement current adaptive immunotherapy. Nevertheless, there is still a lack of high-performance therapeutics for blocking macrophage phagocytosis checkpoint inhibitors in solid tumors. Herein, a peptide-antibody combo-supramolecular in situ assembled CD47 and CD24 bi-target inhibitor (PAC-SABI) is described, which undergoes biomimetic surface propagation on cancer cell membranes through ligand-receptor binding and enzyme-triggered reactions. By simultaneously blocking CD47 and CD24 signaling, PAC-SABI enhances the phagocytic ability of macrophages in vitro and in vivo, promoting anti-tumor responses in breast and pancreatic cancer mouse models. Moreover, building on the foundation of PAC-SABI-induced macrophage repolarization and increased CD8[+] T cell tumor infiltration, sequential anti-PD-1 therapy further suppresses 4T1 tumor progression, prolonging survival rate. The in vivo construction of PAC-SABI-based nano-architectonics provides an efficient platform for bridging innate and adaptive immunity to maximize therapeutic potency.

Adaptive immune checkpoint (IC) inhibitors targeting programmed cell death protein 1 (PD-1) or its ligand programmed death-ligand 1 (PD-L1) have shown efficacy in treating some cancer types by disrupting inhibitory T cell pathways[1–3]. Unfortunately, their effectiveness remains limited against immune quiescent tumors, such as breast and pancreatic cancers (BCs and PCs), underscoring the importance of innate immune cells and enhanced antigen processing to overcome this challenge[4–6]. As the highly prevalent non-malignant cells in tumor microenvironment (TME), macrophages play a crucial innate immune role via phagocytosis, antigen presentation, and inflammatory

[1]Guangdong Cardiovascular Institute, Guangdong Provincial People's Hospital, Guangdong Academy of Medical Sciences, Southern Medical University, Guangzhou, China. [2]Department of Breast Surgery, Guangdong Provincial People's Hospital, Guangdong Academy of Medical Sciences, Southern Medical University, Guangzhou, China. [3]Guangdong Provincial Biomedical Engineering Technology Research Center for Cardiovascular Disease, Zhujiang Hospital, Southern Medical University, Guangzhou, China. [4]Department of Cardiology and Laboratory of Heart Center, Zhujiang Hospital, Southern Medical University, Guangzhou, China. [5]Department of Hepatobiliary Surgery, Zhujiang Hospital, Southern Medical University, Guangzhou, China. [6]Department of Gastro-intestinal Surgery, Zhujiang Hospital, Southern Medical University, Guangzhou, China. [7]Department of Laboratory Medicine, Nanfang Hospital, Southern Medical University, Guangzhou, China. [8]Guangdong Engineering and Technology Research Center for Rapid Diagnostic Biosensors, Nanfang Hospital, Southern Medical University, Guangzhou, China. [9]Department of Cardiovascular Surgery, Zhujiang Hospital, Southern Medical University, Guangzhou, China. [10]These authors jointly supervised this work: Jie Zhan, Ning Liao, Yanbin Cai. ✉e-mail: zhanjie0507@smu.edu.cn; drningliao@163.com; skyer1@smu.edu.cn

cytokine production, linking innate and adaptive immunity[5,7]. Yet cancer cells can evade macrophage clearance by upregulating antiphagocytic "don't eat me" membrane proteins[8–11]. Thus, inhibiting these antiphagocytic signals or receptors represents a promising immunotherapeutic approach that could synergize with cancer cell elimination, activate CD8[+] T cells, and initiate an antitumor immune response.

The CD47-signal regulatory protein alpha (SIRPα) axis, discovered in the late 2000s, was the first identified checkpoint inhibiting cancer phagocytosis through innate immunity[12–14]. Clinical trials are ongoing to test inhibitors and antibodies targeting this axis for cancer therapy. However, two primary challenges have hindered their successful clinical application[15]. One is the ubiquitous CD47 expression on normal cells ("antigen sink"), especially red blood cells and platelets. High maintenance dosing required to saturate peripheral antigens causes severe toxicity like anemia and thrombocytopenia[16]. The other challenge is the limited treatment response in solid tumors[17,18], likely due to additional immunomodulatory receptors beyond CD47 and the inefficiency of blocking antiphagocytic membrane proteins given the TME complexity. Consequently, interest has grown in identifying other macrophage-associated ICs to expand the scope of innate immunotherapy[19]. Recently, CD24 has been identified as a cancer cell "don't eat me" signal via binding the macrophage sialic acid-binding Ig-like lectin 10 (Siglec-10)[20]. Both CD24-Siglec-10 and CD47-SIRPα pathways enable immunoreceptor tyrosine-based inhibitory motif phosphorylation[15,20], highlighting the potential of blocking both CD47 and CD24 to foster optimal antitumor innate immunity.

Current inhibitor designs against membrane protein targets in solid tumors are primarily based on the proteins' intrinsic properties, without adequately accounting for the native curved topological structure of cell membranes[21–23]. Owing to the heightened molecular requisites for synergistic engagement of two distinct proteinaceous targets, this phenomenon may impede binding kinetics and attenuate pharmacological potency of the bi-target inhibitor[24]. By incorporating more information on membrane contexts into structure design, the bi-target inhibitors are expected to achieve improved specificity and efficacy. Recent studies have demonstrated that alkaline phosphatase (ALP)-responsive peptide self-assembly enables interaction with proteins on cell membranes, making it a crucial tool for influencing cell-cell interactions in multicellular systems[25–27]. The enzyme-instructed self-assembly (EISA) effectively modulates the phosphorylated precursors, enabling precise control over their distinctive spatiotemporal functions within TME while concurrently reducing toxicity in the circulatory system[28,29]. By incorporating active targeting motifs, these precursors can be directed toward the formation of self-assembly units with well-organized structures at the subcellular level within the cell or pericellular space[30–33]. Furthermore, the plasma membrane's topological structure tends to facilitate the formation of subsequent peptide assemblies through the coordination of ligand-receptor binding, which is advantageous in achieving maximum inhibitory effects on multiprotein signaling[34–37].

In this work, we develop a peptide-antibody combo-supramolecular in situ assembled CD47 and CD24 bi-target inhibitors (PAC-SABIs) to stimulate macrophage phagocytic activity against malignant cells, thus eliciting potent antitumor immunity. As illustrated in Fig. 1, PAC-SABIs undergo biomimetic, ligand-directed surface propagation like lichens on cancer cell membranes, with conformational maturation mediated by target binding and enzyme catalysis to achieve in situ self-assembly and presentation of bioactive motifs. Our designed modular system demonstrates that responsive nano-architectonics enable meticulous molecular manipulation to construct versatile peptide-antibody therapeutics with enhanced performance. Looking ahead, this proof-of-concept establishes supramolecular peptide engineering as a facile yet powerful approach to merge multitargeting, programmability, and spatial control within a singular nanoscale platform for more efficacious next-generation immunotherapies.

## Results

### Molecular design and self-assembly behavior

To test the feasibility of our concepts and investigate the assembly behavior, we synthesized the peptide molecules as depicted in Fig. 1a. The multifunctional peptide-based inhibitor employed a modular design. In the molecular structure, the fluorophores nitrobenzoxadiazole (NBD) and cyanine 5.5 (Cy5.5) were incorporated as capping groups at the ends of the peptides, and enabled real-time tracking of molecular assemblies in vitro and in vivo through fluorescence imaging. The typical self-assembled peptide, Lys-Leu-Val-Phe-Phe, with a sequence derived from Alzheimer's disease-associated β-amyloid, was the peptide backbone that accomplished molecular assembly. The hydrophilic $PEG_{2000}$ was introduced and implemented to modify the prodrug assemblies for prolonging circulation in blood and increasing accumulation in tumor. Pep-20 (Ala-Trp-Ser-Ala-Thr-Trp-Ser-Asn-Tyr-Trp-Arg-His) incorporated in our molecular design was identified to specifically bind to both human and murine CD47 for blockage of CD47/SIRPα interaction[38]. The peptide monomers were assembled into micelles in the aqueous solution, which were subsequently connected to the anti-CD24 monoclonal antibody (mAb). The linkage allowed for active targeting of CD24 overexpressing cancer cells, thereby exerting a synergistic macrophage immunomodulation (Fig. 1b). Moreover, the tyrosine of Pep-20 enabled the synthesis of phosphorylated analogs (pTyr) to promote in situ rearrangement and growth of molecular assemblies, leading to the formation of final PAC-SABIs upon ALP mediated dephosphorylation (Fig. 1c, d). Expanding upon our supramolecular platform, we engineered additional supramolecular assembled CD47 mono-target inhibitors (SAMIs) to validate the efficacy of PAC-SABIs. Solid-phase peptide synthesis and liquid synthesis were employed, and the detailed procedures were described in Supplementary Figs. 1 and 2. The molecular identification of the intermediates and final products were confirmed in Supplementary Figs. 3–12.

In PBS (pH = 7.4), the critical assembly concentrations (CACs) of peptide molecules (Pep) and conjugates (Pep-PEG) were determined to be 42.6 and 26.6 $10^{-6}$ M, respectively (Fig. 2a). The lower CAC of Pep-PEG could be attributed to the presence of hydrophilic PEG chains, which imparted the conjugate with amphiphilic character and enhanced its self-assembly capability in aqueous medium. The fluorescence associated with NBD potentially serves as a reliable indicator for monitoring the self-assembly of Pep and Pep-PEG. In comparison to concentrations below the CACs, we discerned a noticeable rise in detectable NBD fluorescence at CAC concentrations, which was further amplified under ALP co-incubation (Fig. 2b, c). The heightened fluorescence quantum yields implied that the dephosphorylation process mediated by ALP facilitated the reassembly of Pep and Pep-PEG. After the PEG modification, the zeta potential of the peptide molecules in PBS (pH = 7.4) shifted from $-21.1 \pm 1.1$ mV (Pep) to $-9.1 \pm 1.2$ mV (Pep-PEG) due to the surface charge screening effect (Fig. 2d). The increasing conjugation between Pep-PEG and negatively charged anti-CD24 mAb resulted in a gradual decrease in zeta potential. When the ratio of mAb to Pep-PEG reached 1:10, PAC-SABIs exhibited a potential of $-11.7 \pm 0.8$ mV, and remained stable despite the escalation of Pep-PEG (Supplementary Table 1). Therefore, the ratio of 1:10 was considered to be the saturation point for our following study.

To accurately monitor the secondary structure transformation of the peptide molecules, we conducted circular dichroism (CD) spectroscopy and Fourier transform infrared spectroscopy (FTIR) analysis. The CD spectrum analysis revealed prominent negative peaks at 217 nm for both Pep and PAC-SABIs under ALP co-incubation, with Pep plus ALP group additionally exhibiting a strong positive peak around 198 nm (Fig. 2e). These spectral characteristics were indicative of β-

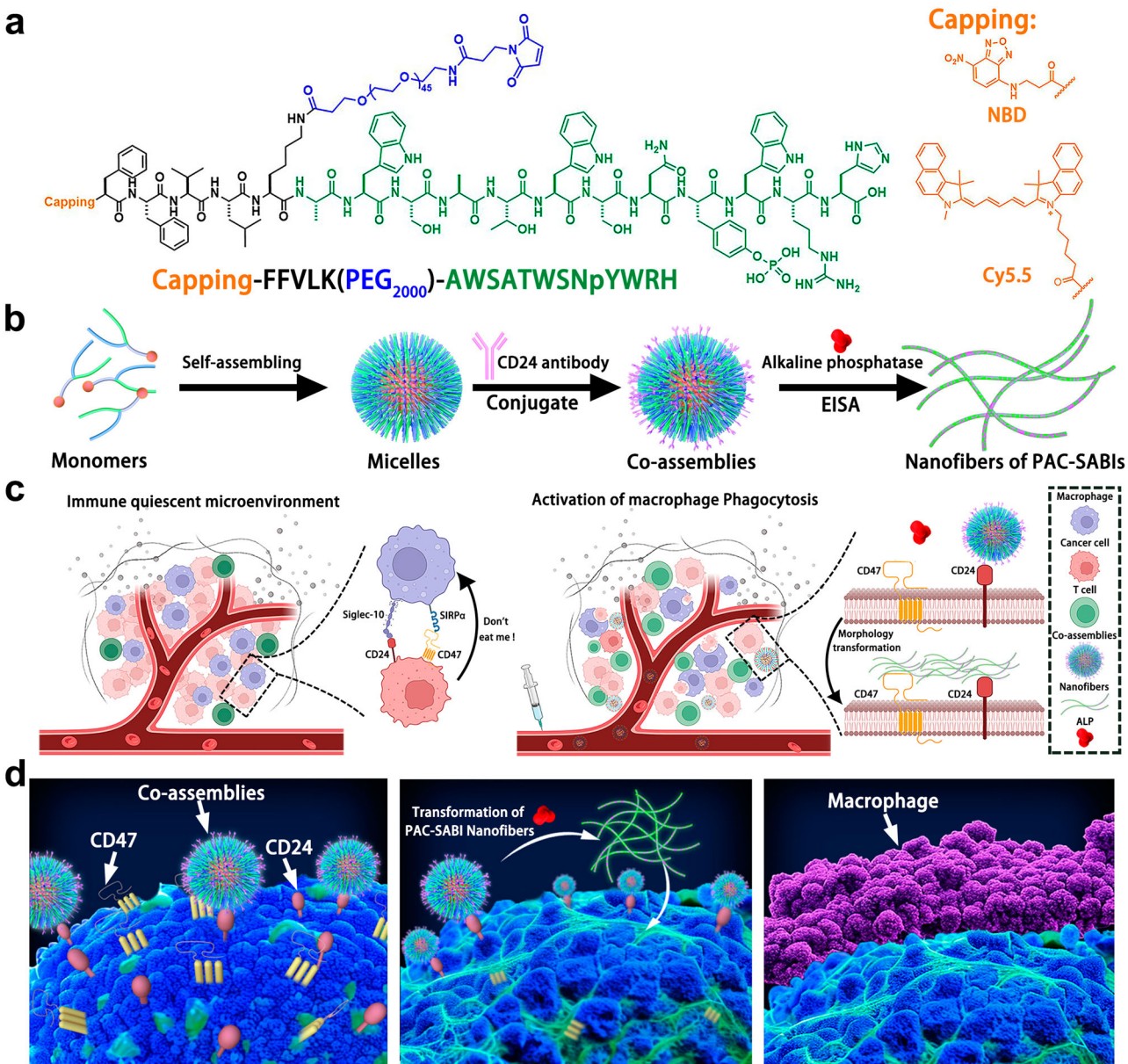

**Fig. 1 | Design and proposed mechanism of PAC-SABIs. a** Peptide molecular structure design of Pep-PEG. **b** Schematic illustration of PAC-SABIs formation process including peptide self-assembling, antibody modification, and ALP catalysis. **c** Schematic illustration of immune quiescent microenvironment and in vivo construction of PAC-SABIs. Figure was created with BioRender.com and released under a Creative Commons Attribution-NonCommercial-NoDerivs 4.0 international license. **d** Proposed mechanism of PAC-SABIs-mediated activation of macrophage phagocytosis against cancer cell.

sheet formations, suggesting that ALP exposure triggered conformational transitions within Pep and PAC-SABIs, leading to an increased proportion of β-sheet content. Subsequent quantitative analysis of the secondary structures underscored this transformation in the ALP-dependent secondary structure of Pep and PAC-SABIs. Specifically, we observed a substantial increase in β-sheet content: from 11.3% to 59.3% in Pep, and from 7.1% to 47.5% in PAC-SABIs post-ALP treatment (Fig. 2f). The FTIR results showed strong absorption bands around 1629 cm$^{-1}$ for Pep and PAC-SABIs in interaction with ALP (Fig. 2g). These bands, attributable to the stretching vibrations of C=O and C-N groups, indicated the occurrence of a high β-sheet structure content. Then, high-performance liquid chromatograph (HPLC) analysis was carried out to monitor the kinetics of EISA. Although the PEG$_{2000}$ and mAb modification might produce steric hindrance for molecular recognition and ALP reaction, Pep, Pep-PEG, and PAC-SABIs all displayed high conversion efficiency, with a conversion rate of

approximately 90% at 60 min (Fig. 2h). These results demonstrated that ALP mediated enzymatic reactions acted as a trigger for initiating the rearrangement of peptide molecules, promoting the rapid formation of ordered superstructures.

Subsequently, we conducted an extensive assessment of the micromorphology of Pep-PEG and their intermediates utilizing transmission electron microscopy (TEM). The TEM imaging results, as illustrated in Supplementary Fig. 13, revealed that the modular design of Pep-PEG manifested distinct variations in self-assembly dynamics and microstructure corresponding to changes in module components. Notably, both Pep and PAC-SABIs displayed time-dependent morphological transformations under ALP co-incubation (Fig. 2i). Pep exhibited a quasi-circular aggregate, wherein these peptide assemblies subsequently aggregated and manifested a spherical structure after a 10-min incubation period with ALP. Then, the advent of a short fiber structure was observed at 30 min. By the 60-min mark, the majority of

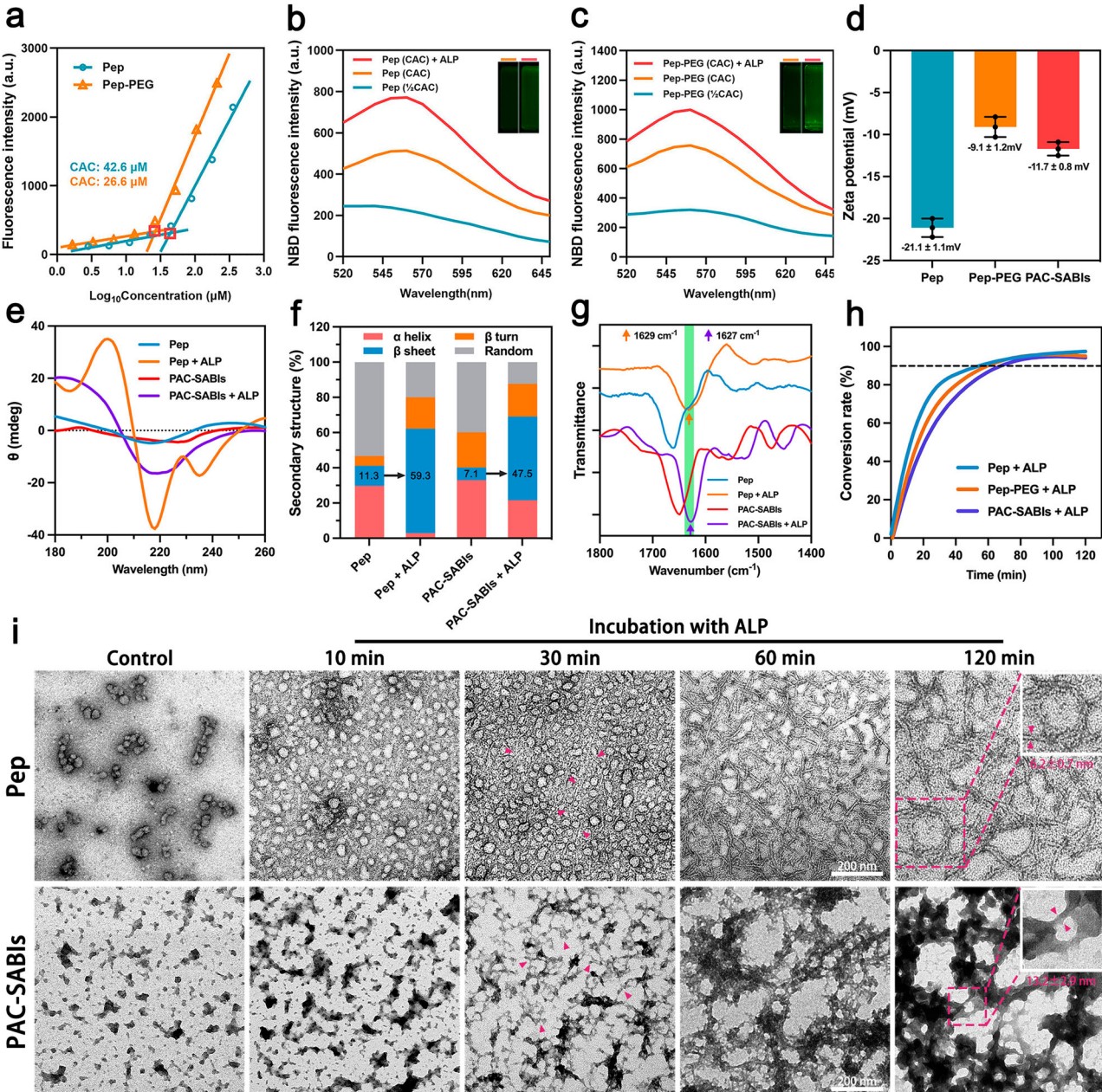

**Fig. 2 | Characterization of PAC-SABIs. a** CACs of Pep and Pep-PEG. Three independent experiments were performed. **b** Fluorescence spectrum of NBD-labeled Pep. Insert: fluorescence emission image of NBD-labeled Pep and Pep plus ALP (1 U/ml). Three independent experiments were performed. **c** Fluorescence spectrum of NBD-labeled Pep-PEG. Insert: fluorescence emission image of NBD-labeled Pep-PEG and Pep-PEG plus ALP (1 U/ml). Three independent experiments were performed. **d** Zeta potentials of Pep, Pep-PEG and PAC-SABIs detected by dynamic light scattering. The error bars represent the mean ± SD ($n = 3$ independent experiments). **e** CD spectrum of Pep and PAC-SABIs in the presence or absence of ALP (1 U/ml). Three independent experiments were performed. **f** Secondary structure calculation of Pep and PAC-SABIs in the presence or absence of ALP (1 U/ml). Three independent experiments were performed. **g** FTIR spectra of Pep and PAC-SABIs in the presence or absence of ALP (1 U/ml). The green area refers to absorption band around 1629 cm⁻¹. Three independent experiments were performed. **h** Conversion rates of Pep, Pep-PEG and PAC-SABIs incubated with ALP (1 U/ml) over time. The black dashed line refers to conversion rates of 90%. Three independent experiments were performed. **i** Time-dependent TEM images of Pep and PAC-SABIs. The red arrows point to the formation of nanofibers. Scale bar: 200 nm. Three independent experiments were performed. Source data are provided as a Source Data file.

the spherical structure was dissipated, giving way to a substantial presence of fibrous structures. At the 120-min interval, the entire assembly system was predominantly occupied by characteristic nanofibers with a diameter of 6.2 ± 0.7 nm (Fig. 2i). However, PAC-SABIs were observed as irregular aggregates. During the initial 30 min of enzyme catalyzed reactions, these aggregates rapidly coalesced into entangled nanofibers. Subsequently, these nanofibers underwent further extension and bundling, resulting in the formation of a three-dimensional (3D) network after 60 min. At 120 min, nearly all PAC-SABIs transformed into a larger network scaffold with interlaced nanofibers presenting a diameter measuring 13.2 ± 2.9 nm (Fig. 2i). TEM analysis showed that the final morphology of PAC-SABIs was closely related to the proportion of peptide molecules in the system. At a mAb: peptide ratio of 1:1, the TEM image depicted large, irregular aggregates. Progressing to a ratio of 1:5, these aggregates appeared denser and more branched, while at a ratio of 1:10, they evolved into a

fibrous network structure, distinguished by the high degree of cross-linking and framework complexity (Supplementary Fig. 14). Thioflavin T (ThT) assays further confirmed the critical role of primary peptide components in forming β-sheet structures within the desired PAC-SABIs system (Supplementary Fig. 15). The notable increase in β-sheet content, characterized by parallel or antiparallel amino acid chains stabilized by hydrogen bonds, facilitated stacking conducive to fiber morphology assembly. Moreover, scanning electron microscopy (SEM) imaging offered a more comprehensive view and surface feature information of PAC-SABIs, implying their potential for 3D network scaffold formation via in situ self-assembly (Supplementary Fig. 16).

## Expression of target innate ICs in BC and PC

To assess the impact of CD47 and CD24 signaling on the regulation of macrophage-mediated immune response against cancer, we investigated the expression of CD47 and CD24 in different tumor types. Analysis of RNA-seq data obtained from The Cancer Genome Atlas (TCGA) revealed a significant overexpression of CD47 and CD24 in nearly all examined tumors (Supplementary Fig. 17a). Furthermore, when compared to the well-established adaptive PD-L1, CD47 and CD24 exhibited a consistent upregulation in BC and PC (Supplementary Fig. 17a). The levels of CD47 and CD24 expression in PC and BC were notably elevated compared to their corresponding normal tissues, respectively, while no significant variation was detected in PD-L1 expression (Supplementary Fig. 17b–d). Stratification of patients based on their CD47 and CD24 expression levels demonstrated a significant correlation with improved overall survival in individuals with lower CD47 expression among PC patients and lower CD24 expression among BC patients (Supplementary Fig. 17e, f). Single-cell RNA-seq analysis of a primary BC sample, focusing on examining the expression patterns of CD47, CD24, CD274, SIRPα and Siglec-10 at the individual cell level, revealed a prominent expression of CD47 and CD24 in cancer cells, notably higher than that of CD274 (Supplementary Fig. 17g, h). Importantly, SIRPα and Siglec-10 were predominantly concentrated within the non-cancerous cellular cohorts, with a marked expression discernible in clusters of tumor-associated macrophages (TAMs) (Supplementary Fig. 17h). These results suggested that CD47 and CD24 may serve as potential markers specific to cancer cells. Moreover, immunofluorescence (IF) analysis of two cancer nodule sections obtained from patient 1 with BC indicated the presence of CD47 and CD24, which had a broad distribution within the cytoplasm and membrane of malignant cells (Supplementary Fig. 17i, j). IF results acquired from consecutive sections of BC tissue in patient 2 showed robust CD47 and CD24 protein expression by cancer cells, indicating redundancy of the two membrane-bound "don't-eat-me" signals (Supplementary Fig. 17k, l).

## In situ self-assembly of PAC-SABIs on cancer cell membranes

As illustrated in Fig. 3a, CD47 and CD24 function as innate ICs, diminishing the phagocytic action of macrophages on cancer cells in a synergistic manner. We hypothesize that by utilizing the interface receptor ligand interaction and ALP catalysis, PAC-SABIs can precisely identify the cancer cell membrane and establish stable assemblies. This process efficiently inhibits CD47 and CD24 signals, regulating TAM phagocytic activity and, eventually, boosting antitumor immune responses. Having determined the effective assembly of PAC-SABIs under static external conditions, we proceeded to investigate the self-assembly behavior of PAC-SABIs within BC and PC cellular environments using the NBD fluorescence. Firstly, the widespread expression of ALP in 4T1 and PAN02 subcutaneous tumor, as well as clinical BC and PC samples, was verified using the BCIP/NBT color development kit (Supplementary Fig. 18). Then, 4T1 and PAN02 cells were co-incubated with NBD-labeled PAC-SABIs at 37 °C, and changes in NBD fluorescence were directly monitored using confocal laser scanning microscopy (CLSM) imaging at different time points (Fig. 3b, c). As

anticipated, the treated 4T1 and PAN02 cells exhibited a time-dependent augmentation in NBD fluorescence emission on the cellular surface. Following a 30-min co-incubation period, the fluorescence appeared as a diffuse signal surrounding the cancer cells as PAC-SABIs bound to the membrane's outer surface. Then, a prompt aggregation of fluorescent clusters on the membrane was observed at the 60-min mark. By the 120-min interval, fluorescence became uniform and encompassed the cell surface, signifying the in situ self-assembly of PAC-SABIs on cancer cell membrane (Fig. 3b, c). This observation was further supported by the findings from bright field and CLSM imaging (Fig. 3d). As shown in Fig. 3e–g, the green fluorescence from NBD exhibited a strong co-localization with the red fluorescence from Dil dye, suggesting that a majority of PAC-SABI molecules were assembled and localized on the 4T1 cell membrane. Under identical conditions, 4T1 cells treated with SAMIs exhibited robust intracellular green fluorescence, reflecting the internalization of SAMIs into the cells, a phenomenon commonly observed when other nanostructures are incubated with live cells (Supplementary Fig. 19). After confirming that anti-CD24 mAbs effectively disrupted the CD24-Siglecs interaction (Supplementary Fig. 20), we conducted a blocking experiment utilizing these anti-CD24 mAbs. The surface fluorescence signals of pretreated 4T1 cells were significantly reduced during dynamic incubation with PAC-SABIs, indicating that the specificity of interaction between CD24 and PAC-SABIs promoted the membrane in situ self-assembly process (Supplementary Fig. 21). Additionally, as shown in Supplementary Fig. 22, TEM revealed the morphologies of PAC-SABIs assembled on the top surface of cell membranes. We also observed the formation of superstructure networks over time on the 4T1 and PAN02 cell membranes treated with PAC-SABIs through SEM imaging. During the co-incubation process from 30 to 120 min, PAC-SABIs, like the proliferation process of lichens, first attached to the cell's adherent edge, then migrated along the cell surface, and finally formed a 3D network scaffold covering the membrane (Fig. 3h, i).

To further emulate the TME and 3D spatial architecture of tumors in vivo, we constructed 4T1 spheroid models. After incubating Cy5.5-labeled PAC-SABIs ([Cy5.5]PAC-SABIs) with the 3D cellular spheres for 120 min, fluorescence monitoring by CLSM revealed Cy5.5 envelopment of the outermost Calcein-AM stained cell layer (Fig. 3j). Scanning multiple confocal planes along the z-axis showed Cy5.5 fluorescent binding on the surfaces of sequential cross-sections, corroborating [Cy5.5]PAC-SABIs assembly on the membrane exteriors. Consistent with 2D culture results, blocking with anti-CD24 mAb markedly reduced external fluorescence of the layered spheroids (Fig. 3k). These results demonstrated the impressive capacity of PAC-SABIs for in situ self-assembly within dynamic physiological milieus, potentially enabling productive IC binding and multiprotein signal inhibition.

Due to the ability of PAC-SABIs to form network scaffold on cancer cell surface, we conducted a series of in vitro experiments including wound healing, cell invasion, and clone formation assays to verify whether PAC-SABIs affect the physiological activity of BC and PC cells. Wound healing assays on the inhibitory ability of SAMIs and PAC-SABIs on cancer cell movement are shown in Supplementary Figs. 23 and 24. The control group, consisting of highly metastatic 4T1 and PAN02 cells, exhibited robust migration and healing capabilities, as evidenced by a wound healing rate of 100% within 24 h post-scratching. However, PAC-SABIs significantly decreased the wound healing rates of 4T1 and PAN02 cells to 45.9% and 20.1%, respectively, which were notably lower than the rates of 72.7% and 69.8% observed in the SAMIs group (Supplementary Figs. 23 and 24). Cell invasion assay was conducted by using Transwell chambers with pre-coating Matrigel. Based on the findings from the control group, it was concluded that the highly metastatic 4T1 and PAN02 cells possessed the ability to break through and dissolve the extracellular matrix barrier, facilitating their migration from the primary tumor site and

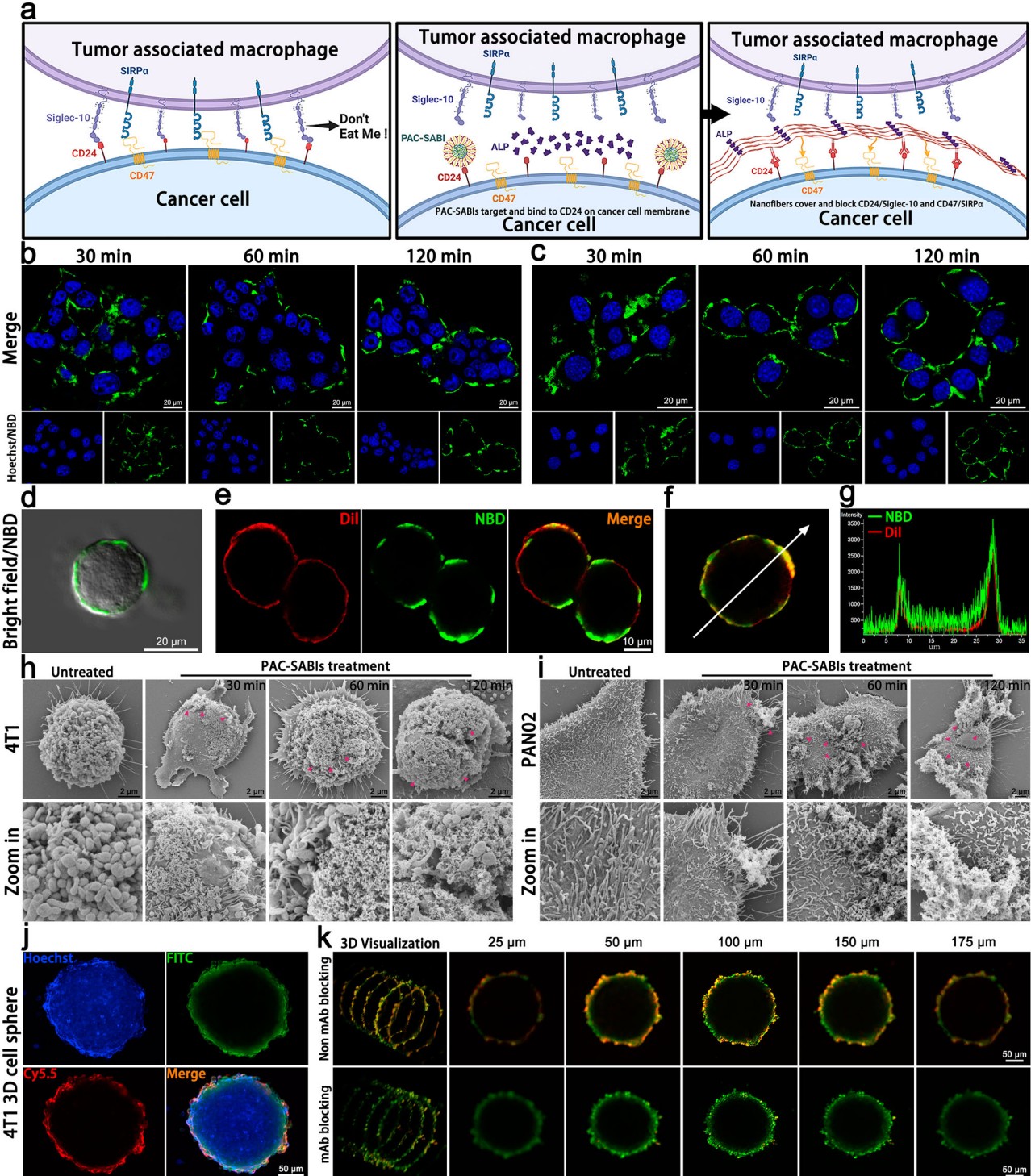

**Fig. 3 | In situ self-assembly of PAC-SABIs on BC and PC cell membranes.**
**a** Schematic illustration of tumor immune quiescent microenvironment, in situ assembly of PAC-SABIs on the cancer cell surface, and blockage of CD47 and CD24 phagocytic checkpoints by PAC-SABIs. Figure was created with BioRender.com and released under a Creative Commons Attribution-NonCommercial-NoDerivs 4.0 international license. **b** Time-dependent CLSM images of 4T1 cells treated with NBD-labeled PAC-SABIs. Scale bar: 20 μm. Three independent experiments were performed. **c** Time-dependent CLSM images of PAN02 cells treated with NBD-labeled PAC-SABIs. Scale bar: 20 μm. Three independent experiments were performed. **d** Merged bright field and fluorescence images of 4T1 cell treated with NBD-labeled PAC-SABIs for 120 min. Scale bar: 20 μm. Three independent experiments were performed. **e** CLSM images of 4T1 cell treated with DiI dye (red) and NBD-labeled PAC-SABIs for 120 min. Scale bar: 10 μm. Three independent experiments were performed. **f**, **g** Fluorescence distribution of NBD-labeled PAC-SABIs on 4T1 cell. Three independent experiments were performed. **h** Time-dependent SEM images of 4T1 cells treated with PAC-SABIs. The red arrows point to PAC-SABIs on cell membrane. Scale bar: 2 μm. Three independent experiments were performed. **i** Time-dependent SEM images of PAN02 cells treated with PAC-SABIs. The red arrows point to the PAC-SABIs on cell membrane. Scale bar: 2 μm. Three independent experiments were performed. **j** CLSM images of 3D 4T1 spheroids treated with $^{Cy5.5}$PAC-SABIs, Calcein-AM, and Hoechst 33342. Scale bar: 50 μm. Three independent experiments were performed. **k** CLSM images of 3D 4T1 spheroids along the z-axis position. Scale bar: 50 μm. Three independent experiments were performed.

subsequent establishment of new metastatic sites. PAC-SABIs yielded a substantial inhibitory effect on the invasion of 4T1 and PANO2 cells, with the inhibition rates of 83.4% and 91.1%, respectively (Supplementary Figs. 23 and 24). Furthermore, we conducted a plate colony formation assay to examine the impact of PAC-SABIs on 4T1 and PANO2 cell proliferative capacity. The results showed that the number of tumorigenic cells significantly decreased, indicating that PAC-SABIs significantly inhibited the colony formation ability of 4T1 and PANO2 cells, with inhibition rates of 79.1% and 80.3%, respectively, which were significantly superior to the inhibitory effect of SAMIs (Supplementary Figs. 23 and 24).

### Promotion on phagocytic clearance of cancer cells via PAC-SABIs treatment in vitro

To evaluate the phagocytic elimination of cancer cells by macrophages, 4T1 and PANO2 cells labeled with the pH-sensitive dye pHrodo Red were co-cultured with RAW264.7 cells pre-exposed to tumor conditioned media (TCM). Over 2 h, PAC-SABIs-treated 4T1 and PANO2 cells exhibited considerably enhanced susceptibility to engulfment and degradation within acidic phagolysosomes compared to IgG controls (Fig. 4a, b and Supplementary Fig. 25). Investigating the therapeutic potential of these findings, we assessed whether PAC-SABIs could amplify the phagocytosis of cancer cells by diverse macrophage populations beyond direct anti-CD24 mAb or SAMIs blockade. 3D reconstructions of CLSM image stacks showed increased vulnerability of 4T1-pHRodo-Red+ cells to phagolysosomal uptake when treated with anti-CD24 mAb or SAMIs. PAC-SABIs further enhanced engulfment of 4T1 cells by TCM-exposed bone marrow-derived macrophages (BMDMs) and RAW264.7 cells (Fig. 4c). Flow cytometry quantitation revealed robustly amplified phagocytosis with PAC-SABIs addition, with approximately 4- and 2-fold greater uptake by BMDMs and RAW264.7 cells respectively compared to anti-CD24 mAb alone (Fig. 4d–g and Supplementary Fig. 26).

In agreement with the results in murine macrophages, human THP-1 monocyte-derived macrophages stimulated by TCM exhibited enhanced clearance of human BC MDA-MB-231 cells treated with PAC-SABIs compared to other groups (Supplementary Fig. 27). Primary human donor-derived macrophages (HDDMs) were also generated by the methodology previously outlined by Barkal et al.[20] to assess PAC-SABIs efficacy (Supplementary Fig. 28a). Subsequently, the HDDMs were conditioned with TCM from MDA-MB-231 to generate TAMs. IF and flow cytometry evidenced upregulation of the M2 marker CD206 in TAMs versus control HDDMs, regardless of their interaction with PAC-SABIs pre-treated MDA-MB-231 cells (Supplementary Figs. 28b, c and 29). As shown in Supplementary Fig. 29c, no significant difference in CD206 expression was noted between G2 and G3. These results provided further proof that the observed heightened phagocytic activity in macrophages from 2D and 3D CLSM imaging was primarily due to the inhibitory effects of PAC-SABIs on the innate ICs CD47 and CD24, rather than a shift in macrophage polarization from M2- to M1-like phenotype (Supplementary Fig. 28d–f).

### Biodistribution and tumor targeting in vivo

In vivo imaging system (IVIS) spectroscopy was employed, as well as subcutaneous BC and PC xenograft models were constructed to monitor the biodistribution of Cy5.5PAC-SABIs after intravenous injection. As shown in Fig. 5, there were notable differences in the distribution of fluorescence among free Cy5.5, Cy5.5SAMIs, and Cy5.5PAC-SABIs. Cy5.5, serving as the representative small molecule probe, exhibited swift distribution and elimination throughout the mice's bodies, without displaying any discernible specific targeting effects on the BC and PC tissues. Due to the absence of anti-CD24 mAb and ALP triggered self-assembly property, Cy5.5SAMIs exhibited limited accumulation of fluorescence signal in the tumor regions. However, Cy5.5PAC-SABIs illuminated the BC and PC, and reached its maximum

intensity at the 48-h point. The EISA and PEG effects significantly improved Cy5.5PAC-SABIs' biodistribution and prolonged their tumor retention time for up to 120 h (Fig. 5a, c). The area under the curve (AUC), which is a crucial parameter for the enrichment of Cy5.5PAC-SABIs and Cy5.5SAMIs in the tumor regions, was determined through fluorescence quantitative analysis. The AUC (0–120 h) of Cy5.5PAC-SABIs was observed to be approximately 3 times greater than that of Cy5.5SAMIs. In addition, our investigation revealed that the elimination of Cy5.5PAC-SABIs in BC and PC exhibited a remarkably slow rate, with only a modest decrease of 16.2% and 25.6% between 48 and 120 h, respectively (Fig. 5b, d).

After the intravenous administration of free Cy5.5, Cy5.5SAMIs, and Cy5.5PAC-SABIs, the mice were euthanized at the 48-h mark, and the heart, liver, spleen, lung, kidney, and tumor were harvested for ex vivo imaging (Fig. 5e, f). Ex vivo near infrared (NIR) fluorescence imaging and quantitative analysis revealed that, consistent with in vivo findings, free Cy5.5 was rapidly cleared from the body with negligible accumulation in the tumors and major organs. Despite the observed retention in metabolic organs such as the liver and kidney, distinct variations in tissue distribution were noted between Cy5.5SAMIs and Cy5.5PAC-SABIs (Fig. 5e, f and Supplementary Fig. 30). Specifically, Cy5.5PAC-SABIs achieved a marked selective enrichment in both BC and PC tumors. In contrast, Cy5.5SAMIs presented a significant reduction in tumor accumulation, displaying fluorescence intensities comparable to that of lung tissues (Supplementary Fig. 30). The discernible dissimilarity between these two molecules could be attributed to the presence of targeted ligands and EISA, which augmented the precise recognition of Cy5.5PAC-SABIs toward cancer cells and facilitated their proficient molecular assembly. To further observe the distribution of Cy5.5PAC-SABIs in the BC and PC tissues, we analyzed the Cy5.5 fluorescence of tumor sections. Interestingly, a notable accumulation of Cy5.5 fluorescence was observed within the cancer cell regions of the 4T1 tumor, whereas the fluorescence intensity was diminished in the fibroblast-rich areas (Fig. 5g). This phenomenon was similar within the interior of PANO2 tumors, where Cy5.5PAC-SABIs exhibited a concentration within the cancer cell area rather than the paracancerous or fibroid regions (Fig. 5g). Moreover, we employed SEM imaging to visualize the in situ self-assembly morphology of PAC-SABIs. In the 4T1 and PANO2 subcutaneous xenografts and clinical BC tissues, SEM images revealed the formation of an intricate nanofiber network scaffold orchestrated by PAC-SABIs, enveloping the surfaces of malignant cells (Supplementary Figs. 31 and 32). These results indicated that the PAC-SABIs' superstructure formation led to their effective accumulation and retention in tumors, which may provide persistent blockade of innate ICs, thereby inducing sustained macrophage-mediated immune responses.

Given the obviously prolonged PAC-SABIs circulation in blood, it is crucial to understand the conditions that influence the interaction between PAC-SABIs and ALP. We first introduced PAC-SABIs into a PBS buffer (pH = 7.4) containing various ALP concentrations. The mixture was incubated at 37 °C for 60 min. Reverse phase HPLC analysis showed that the residual PAC-SABIs decreased consistently with increasing ALP concentrations, with percentages ranging from 96.7 to 6.5% across ALP concentrations of 0.01 to 8.00 U/ml (Supplementary Fig. 33). Notably, our two-piecewise linear regression model analysis indicated an optimal ALP threshold of 1.06 U/ml (Supplementary Fig. 34). Subsequently, we conducted a retrospective study on 22 BC patients from Guangdong Provincial People's Hospital between November and December 2023 (Supplementary Table 2). Most of these patients (95.5%) had preoperative ALP levels within the normal range (50–139 U/l), with a median value (64 U/l) significantly below our in vitro ALP reactivity threshold (Supplementary Table 2 and Supplementary Fig. 35). This suggested that PAC-SABIs could remain stable in the circulatory system of most clinical BC patients. To corroborate this, we analyzed serum samples from four BC patients, co-incubated

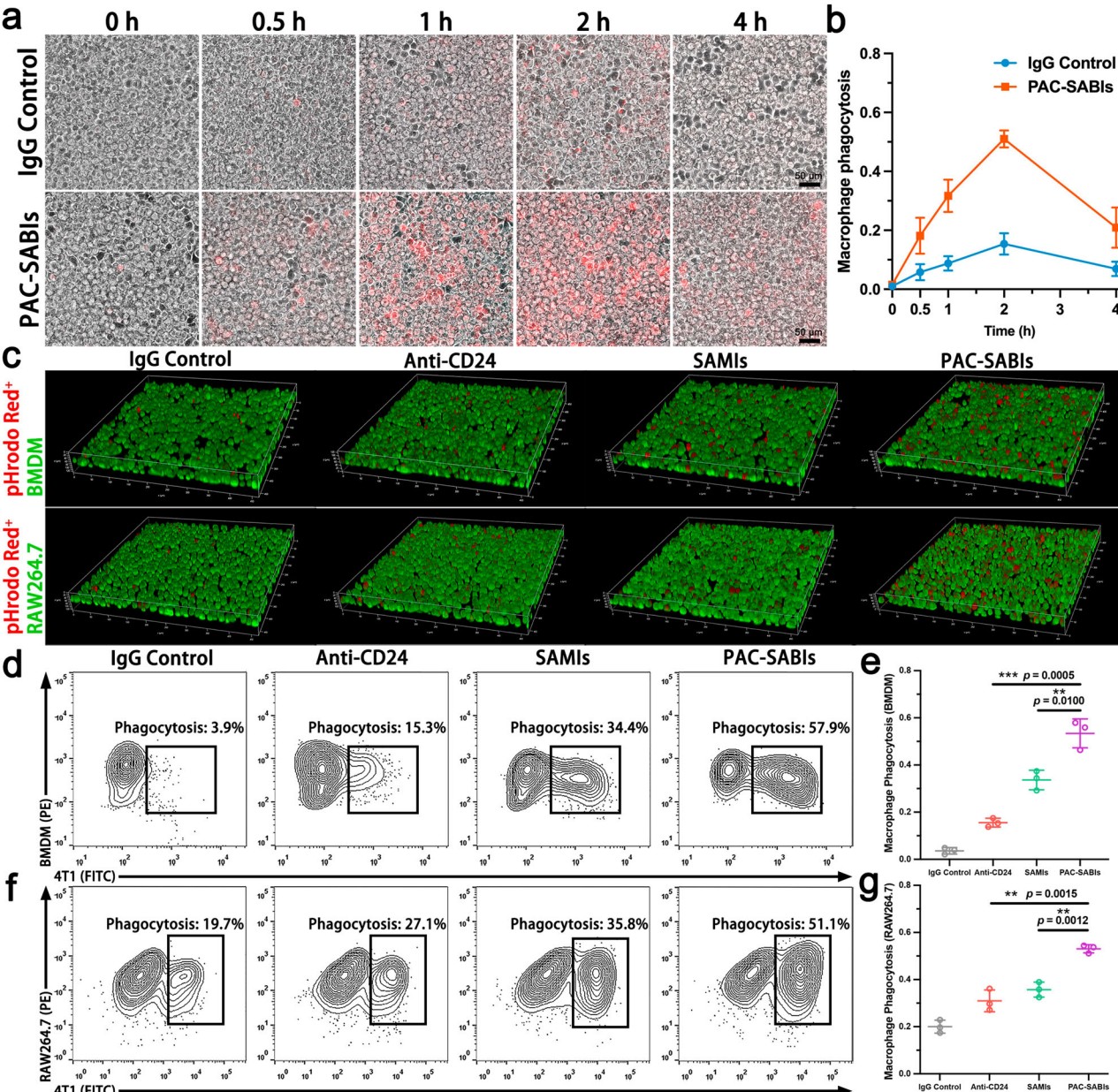

**Fig. 4 | Promotion of phagocytic clearance of cancer cells via PAC-SABIs treatment in vitro. a** Phagocytosis images of pHrodo-red⁺ over time. Scale bar: 50 μm. Three independent experiments were performed. **b** Phagocytosis of 4T1 cells, in the presence of IgG control or PAC-SABIs. The error bars represent the mean ± SD ($n = 3$ independent experiments). **c** Representative 3D CLSM image reconstruction of in vitro phagocytosis of 4T1 cells (pHrodo-red⁺, red) by BMDMs and RAW264.7 cells (Calcein-AM, green). Three independent experiments were performed. **d** Representative flow cytometry plots depicting BMDM phagocytosis of 4T1 cells treated with IgG control, anti-CD24 mAb, SAMIs and PAC-SABIs.

**e** Quantitative analysis of BMDM flow cytometry results. The error bars represent the mean ± SD ($n = 3$ independent experiments; **$p < 0.01$, ***$p < 0.001$; the $p$ value was analyzed by a two-tailed unpaired Student's $t$ test). **f** Representative flow cytometry plots depicting RAW264.7 cell phagocytosis of 4T1 cells treated with IgG control, anti-CD24 mAb, SAMIs and PAC-SABIs. **g** Quantitative analysis of RAW264.7 cell flow cytometry results. The error bars represent the mean ± SD ($n = 3$ independent experiments; **$p < 0.01$; the $p$ value was analyzed by a two-tailed unpaired Student's $t$ test). Source data are provided as a Source Data file.

with PAC-SABIs. Our findings showed a gradual degradation of PAC-SABIs, maintaining over 60% stability after 48 h (Supplementary Fig. 36). These results not only confirmed the long-term stability of PAC-SABIs in a biological environment but also underscored their potential for clinical application.

## Antitumor efficacy and immune response in vivo
Generally, the toxicity of inhibitors, especially the blood toxicity associated with CD47 blockade therapy, is a key criterion affecting clinical transformation. To evaluate the systemic toxicity of PAC-SABIs,

complete blood count was obtained from healthy BALB/c mice after the intravenous injection of PAC-SABIs. The main parameters included red blood cell, hemoglobin, hematocrit, mean corpuscular hemoglobin, mean corpuscular hemoglobin concentration, mean corpuscular volume, mean platelet volume and platelets. All of these parameters were shown to be within normal reference ranges (Supplementary Fig. 37). Considering PAC-SABIs' biodistribution in the liver and kidneys, the key blood biochemical markers such as alanine aminotransferase, aspartate aminotransferase, creatinine, and blood urea nitrogen in healthy BALB/c mice were monitored continuously post

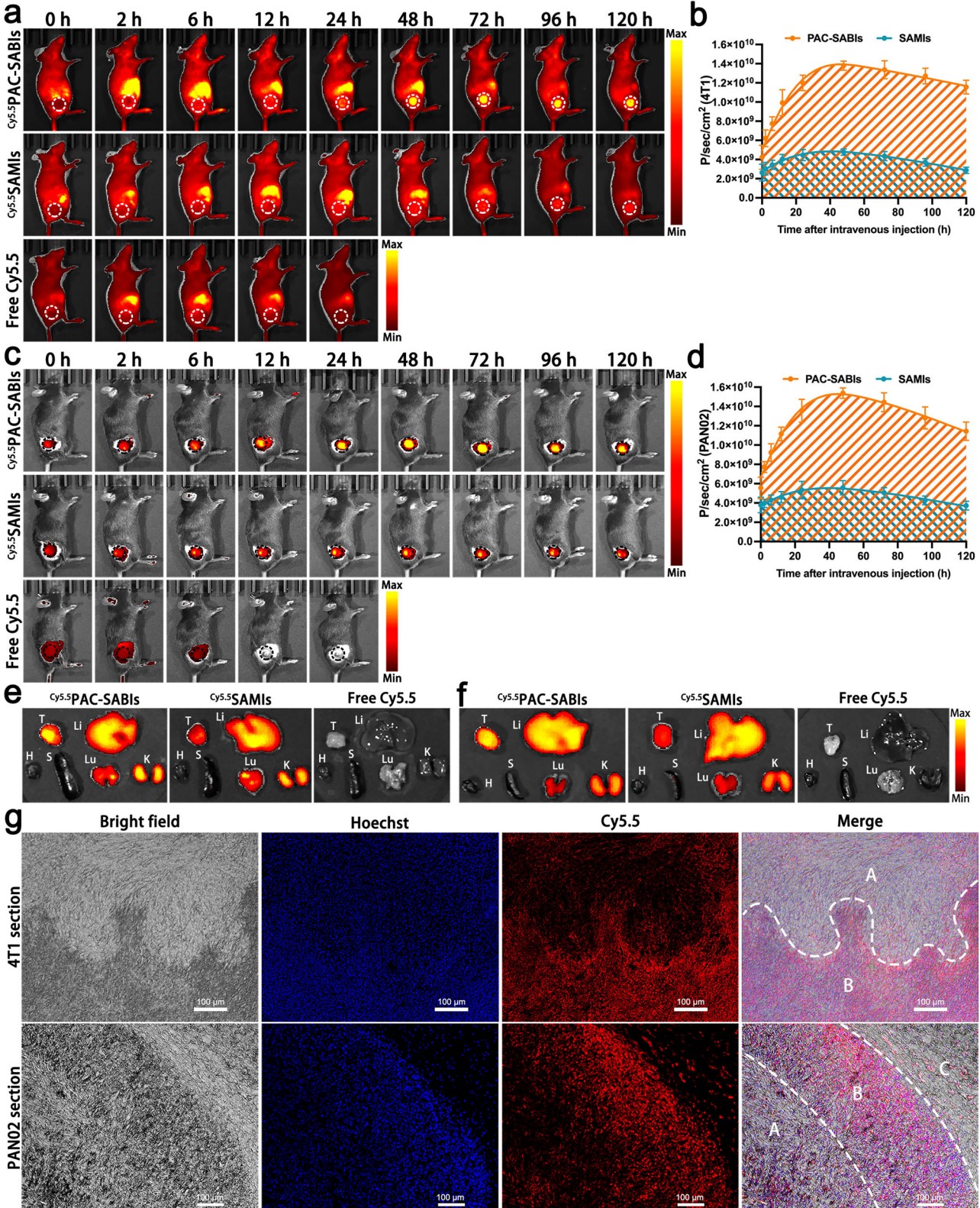

intravenous injection of PAC-SABIs at varying doses. Under the treatment regime using 15 mg/kg PAC-SABIs, as well as at double and triple the initial dose concentrations, all recorded parameters were maintained within the normal reference spectrum (Supplementary Fig. 38). Subsequently, following approximately 1 month of treatment, we collected the main organs of mice in each group, and sliced them for H&E staining. No notable histological alterations or pathological

lesions were observed (Supplementary Fig. 39). These results exhibited high biocompatibility characteristics of PAC-SABIs in mice without significant systemic toxicity.

Encouraged by the aforementioned experimental results, we hypothesized that PAC-SABIs therapy has the potential to instruct macrophages in eliciting proficient in vivo antitumor phagocytic reactions, consequently leading to the inhibition of tumor growth. To

**Fig. 5 | Biodistribution and tumor targeting of $^{Cy5.5}$PAC-SABIs in vivo.**
**a** Representative NIR fluorescence images of free Cy5.5, $^{Cy5.5}$SAMIs and $^{Cy5.5}$PAC-SABIs on 4T1 subcutaneous tumor-bearing mice after intravenous injection. Images were acquired at 0, 2, 6, 12, 24, 48, 72, 96 and 120 h post injection. The white dashed circles refer to the tumor site. **b** Time-dependent quantitative calculation of the average fluorescence intensity in 4T1 tumor area and AUCs of $^{Cy5.5}$SAMIs and $^{Cy5.5}$PAC-SABIs. The error bars represent the mean ± SD ($n$ = 3 mice). **c** Representative NIR fluorescence images of free Cy5.5, $^{Cy5.5}$SAMIs and $^{Cy5.5}$PAC-SABIs on PAN02 subcutaneous tumor-bearing mice after intravenous injection. Images were acquired at 0, 2, 6, 12, 24, 48, 72, 96 and 120 h post injection. The black dashed circles refer to the tumor site. **d** Time-dependent quantitative calculation of

the average fluorescence intensity in PAN02 tumor area and AUCs of $^{Cy5.5}$SAMIs and $^{Cy5.5}$PAC-SABIs. The error bars represent the mean ± SD ($n$ = 3 mice). **e** Ex vivo NIR fluorescence images of 4T1 tumor and major organs (H heart, Li liver, S spleen, Lu lung, K kidney) collected post 48 h injection. Three independent experiments were performed. **f** Ex vivo NIR fluorescence images of PAN02 tumor and major organs (H heart, Li liver, S spleen, Lu lung, K kidney) collected post 48 h injection. Three independent experiments were performed. **g** Fluorescence images of 4T1 and PAN02 subcutaneous tumor sections at 48 h post-injection of $^{Cy5.5}$PAC-SABIs. A refers to the area of fibroids; B refers to the area of cancer cells; C refers to the paracancerous area. Scale bar: 100 μm. Three independent experiments were performed. Source data are provided as a Source Data file.

evidence this, subcutaneous xenograft models of BC and PC were established using 4T1 and PAN02 cells, respectively. Mice were randomly distributed into four groups: IgG Control, Anti-CD24, SAMIs, and PAC-SABIs, and received their respective treatments intravenously every other day over four sessions. We continuously monitored the mouse weight during the treatment. On the 21st day, all mice were humanely euthanized, and the 4T1 and PAN02 tumors were surgically removed and weighed (Supplementary Fig. 40a, e). No significant body weight differences were observed among the groups (Supplementary Fig. 40c, g). Notably, PAC-SABIs demonstrated a significant reduction in tumor growth in both BC and PC models, surpassing the efficacy of anti-CD24 mAb or SAMIs (Supplementary Fig. 40). Furthermore, in the BC xenograft model, PAC-SABIs exhibited a significantly stronger tumor suppression effects than dual antibody combination therapy (Supplementary Fig. 41). Specifically, the average tumor weights were 1.07 g for the antibody combination group and 0.33 g for the PAC-SABIs group, reflecting a unique PAC-SABIs functionality of concurrently inhibiting CD47 and CD24 signaling, coupled with their intrinsic self-assembly ability within the complex TME.

To mimic the natural progression of human BC and PC, the orthotopic BC and PC models were then constructed for in vivo therapeutic efficacy evaluation. According to the depicted illustration (Fig. 6a, e), after the establishment of in situ BC and PC, IgG control, anti-CD24 mAb, SAMIs and PAC-SABIs were administered every other day for a total of 4 doses. The evaluation of tumor regression/progression was conducted through the periodic monitoring of tumor bioluminescence (BLI) at 7-day intervals (Fig. 6b, f). Over time, on the 22nd day post-treatment, the PAC-SABIs group exhibited significantly reduced tumor BLI signal intensities in comparison to the other groups, suggesting a notable deceleration in tumor growth among the BC and PC mice subjected to PAC-SABIs treatment (Fig. 6c, g). Conversely, mice treated with IgG control and anti-CD24 mAb exhibited limited therapeutic efficacy, as the majority of BC and PC mice succumbed within 40 and 50 days, respectively (Fig. 6d, h). The overall survival rate of BC mice treated with PAC-SABIs on the 50th day and that of PC mice on the 60th day were 60% and 40%, respectively, showing a significant survival benefit of PAC-SABIs (Fig. 6d, h). As shown in Fig. 6i–n, macrophage depletion assays were also conducted within an in vivo 4T1 model, and the results reinforced the validation of the above antitumor effect of PAC-SABIs being primarily linked to the heightened phagocytosis of cancer cells by macrophages.

To examine the regulatory effects of PAC-SABIs in vivo, the changes that occurred in the 4T1 TME were analyzed after intravenous administration. H&E staining of xenograft sections in both the IgG control and anti-CD24 groups revealed that the cancer cells were densely arranged in bands and clusters, with disordered arrangement (Fig. 7a). In the PAC-SABIs group, a notable presence of inflammatory cells and tissue necrosis was observed within the tumor interstitium, showing a greater extent of inflammatory cell infiltration and tissue necrosis compared to the SAMIs group (Fig. 7a). Then, we conducted IF staining and observed a noteworthy manifestation of macrophages

engaging in phagocytic and cytotoxic activities against cancer cells in the PAC-SABIs group, characterized by extensive macrophage infiltration and internalization of cancer cell components (Fig. 7a). The nuclear density within this particular region was lower than that in adjacent regions, indicating a reduction in cancer cell density. Conversely, this tumoricidal phenomenon was less frequently observed in the remaining treatment groups. In addition, the IF results demonstrated a minimal presence of CD8+ T cells in the tumor core of the IgG control group, whereas a notable increase in the infiltration depth and number of CD8+ T lymphocytes was observed in the tumor tissue treated with PAC-SABIs (Fig. 7a).

Flow cytometry analyses were performed on GFP+ 4T1 tumors to further explore how PAC-SABIs inhibition altered the immune components in TME. Consistent with the IF results, the highest level of in vivo phagocytosis by infiltrating TAMs was achieved in the PAC-SABIs group, which showed 3-fold higher than that in the IgG control group (Fig. 7b, c and Supplementary Fig. 42). As shown in Fig. 7d, e, anti-CD24 mAb, SAMIs, and PAC-SABIs treatments all potentiated the activation state of TAMs toward a pro-inflammatory antitumor phenotype, indicating a potential mechanism of innate IC blockade strategy. Moreover, flow cytometry analysis of T cells in tumor tissues treated with PAC-SABIs demonstrated a significant increase in the percentage of tumor-infiltrating CD8+ and CD4+ T cells (Fig. 7b, f). We speculated that the phagocytosis of activated macrophages could increase the cross-presentation of tumor antigen to T cells, which enhanced the effectiveness of adaptive antitumor immune response. As the main immunosuppressive cells, the number of FOXP3 regulatory T cells (Tregs) within the tumor following PAC-SABIs treatment was found to be significantly decreased (Fig. 7b, g). This reduction might contribute to the further transition of the macrophage population from a suppressive M2-like phenotype to an M1-like phenotype. Previous studies have shown that reprogramming the TME can impact the cytokine secretion of immune cells, resulting in heightened levels of pro-inflammatory cytokines such as tumor necrosis factor alpha (TNF-α), interferon gamma (IFN-γ), and interleukin-6 (IL-6), and reduced levels of the anti-inflammatory cytokine transforming growth factor beta (TGF-β). As illustrated in Supplementary Fig. 43, the pro-inflammatory cytokines (TNF-α, IFN-γ, and IL-6) in both mouse plasma and tumor tissues showed a significant increase after treatment with PAC-SABIs, while a notable decrease was observed in the anti-inflammatory cytokine TGF-β. These findings strongly indicated that PAC-SABIs held the ability to counteract the anti-inflammatory properties within tumor immune microenvironment (TIME).

Meanwhile, the RNA-seq analyses on PAC-SABIs-treated 4T1 tumors in vivo were performed for an extensive evaluation of its resultant intricate alterations in the TIME. As illustrated in the Fig. 7h, a total of 1276 differential genes were identified, encompassing 486 upregulated genes and 790 downregulated ones. Gene Ontology (GO) assessment elucidated that a predominant fraction of upregulated genes was substantially enriched in immune related pathways such as innate and adaptive immune responses (Fig. 7i). Notably, the three most significant pathways showed intimate connection to activation of

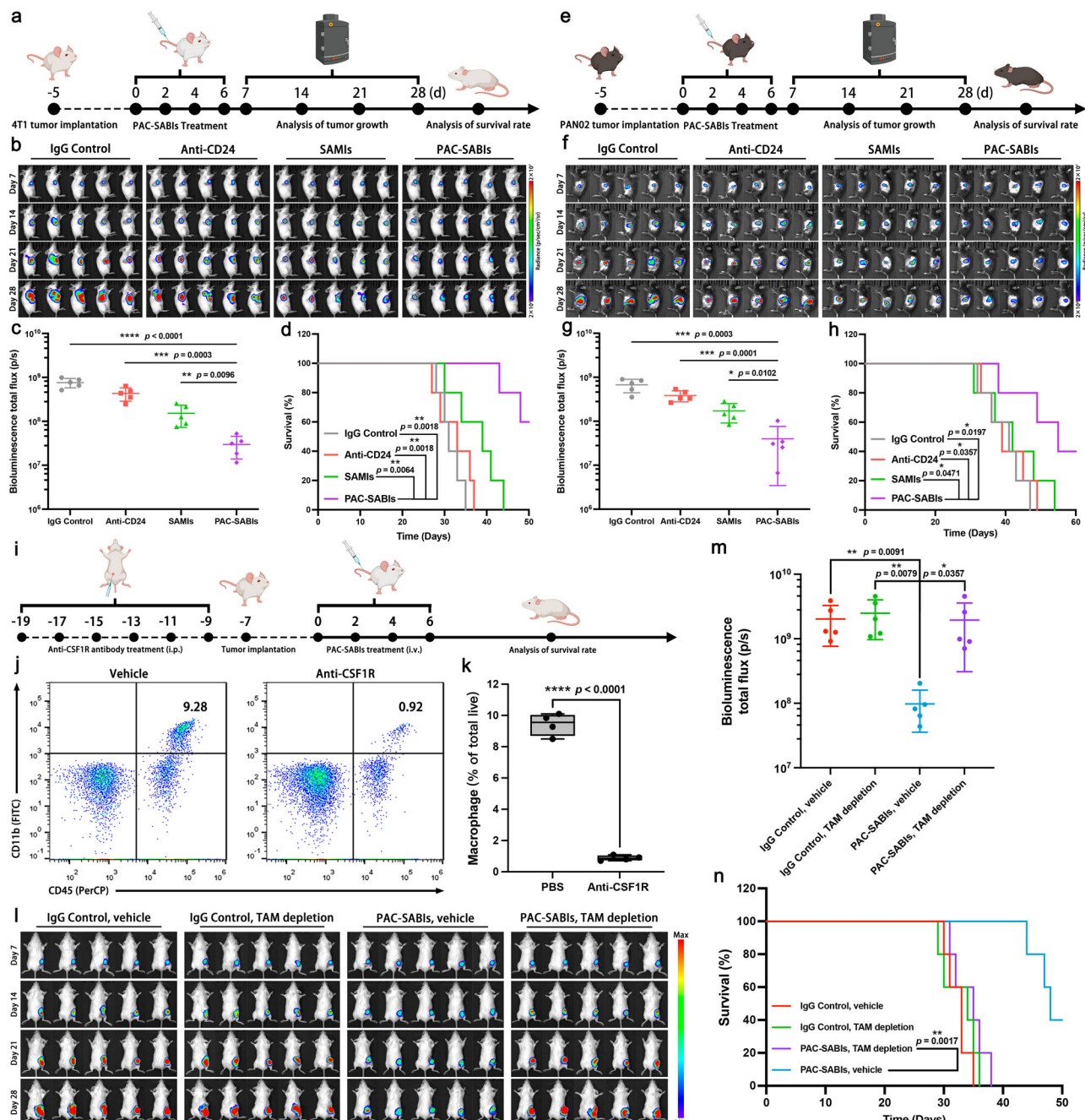

**Fig. 6 | Antitumor efficacy of PAC-SABIs in vivo. a** Scheme of the PAC-SABIs therapeutic strategy for 4T1 orthotopic tumor. **b** BLIs of 4T1 tumor-bearing mice on days 7, 14, 21, and 28 ($n = 5$ mice). **c** Quantification analysis of 4T1 tumor BLI signals in each treatment group. The error bars represent the mean ± SD ($n = 5$ mice; **$p < 0.01$, ***$p < 0.001$, ****$p < 0.0001$; the $p$ value was analyzed by a two-tailed unpaired Student's $t$ test). **d** Kaplan–Meier survival curves of 4T1 tumor-bearing mice ($n = 5$ mice; **$p < 0.01$; the $p$ value was analyzed by log-rank test). **e** Scheme of the PAC-SABIs therapeutic strategy for PAN02 orthotopic tumor. **f** BLIs of PAN02 tumor-bearing mice on days 7, 14, 21, and 28 ($n = 5$ mice). **g** Quantification analysis of PAN02 tumor BLI signals in each treatment group. The error bars represent the mean ± SD ($n = 5$ mice; *$p < 0.05$, ***$p < 0.001$; the $p$ value was analyzed by a two-tailed unpaired Student's $t$ test). **h** Kaplan–Meier survival curves of PAN02 tumor-bearing mice ($n = 5$ mice; *$p < 0.05$; the $p$ value was analyzed by log-rank test).

**i** Scheme of macrophage depletion and therapeutic strategy for 4T1 subcutaneous tumor. **j** Representative flow cytometry plots of tissue-resident macrophages. **k** Quantification analysis of tissue-resident macrophages. Boxplots represent the median and interquartile range, and the whiskers denote minimum and maximum values ($n = 4$ mice; ****$p < 0.0001$; the $p$ value was analyzed by a two-tailed unpaired Student's $t$ test). **l** BLIs of 4T1 tumor-bearing mice on days 7, 14, 21, and 28 ($n = 5$ mice). **m** Quantification analysis of BLI signals of 4T1 tumor in each treatment group. The error bars represent the mean ± SD ($n = 5$ mice; *$p < 0.05$, **$p < 0.01$; the $p$ value was analyzed by a two-tailed unpaired Student's $t$ test). **n** Kaplan–Meier survival curves of 4T1 tumor-bearing mice ($n = 5$ mice; **$p < 0.01$; the $p$ value was analyzed by log-rank test). **a**, **e**, **i** were created with BioRender.com and released under a Creative Commons Attribution-NonCommercial-NoDerivs 4.0 international license. Source data are provided as a Source Data file.

immune response, lymphocyte mediated immunity, and phagocytosis (Fig. 7i, j). Moreover, genes central to regulatory roles in immunity, inflammatory processes, macrophage phagocytotic capacity, and antigenic presentation experienced marked upregulation subsequent

to PAC-SABIs therapy (Fig. 7k). Consequently, the transcriptomic outcomes provided additional evidence to substantiate the effectiveness of PAC-SABIs' innate IC blockade strategy in eliciting potent antitumor immune activity.

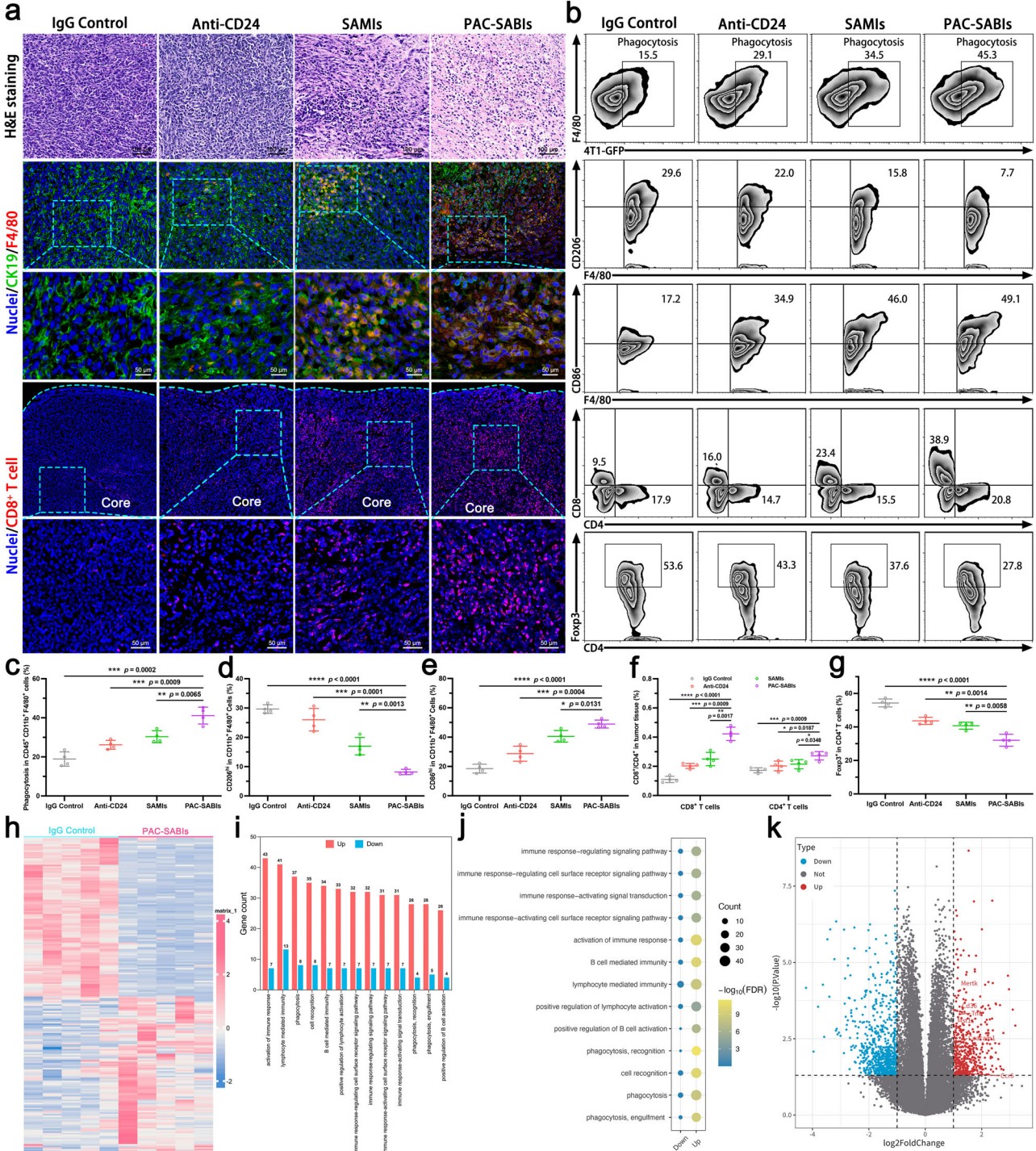

**Fig. 7 | Immune response in vivo. a** Representative H&E and IF staining of cytokeratin 19 (CK19), F4/80+ macrophages and CD8+ T cells for the corresponding 4T1 tumor tissues after different treatments. Scale bars are marked in the figures. **b** Representative flow cytometry plots depicting CD45+CD11b+F4/80+ macrophages phagocytosis, CD11b+F4/80+CD206hi macrophages, CD11b+F4/80+CD86hi macrophages, CD45+CD3+CD4+/CD8+ T cells, and CD45+CD3+CD4+Foxp3+ T cells in 4T1 tumors after different treatments. **c**–**g** Quantification analysis of CD45+CD11b+F4/80+ macrophages phagocytosis, CD11b+F4/80+CD206hi macrophages, CD11b+F4/80+CD86hi macrophages, CD45+CD3+CD4+/CD8+ T cells, and CD45+CD3+CD4+Foxp3+ T cells in 4T1 tumors after different treatments. The error bars represent the mean ± SD ($n = 4$ independent experiments; *$p < 0.05$, **$p < 0.01$, ***$p < 0.001$,

****$p < 0.0001$; the $p$ value was analyzed by a two-tailed unpaired Student's $t$ test). **h** Identification of differentially expressed genes in 4T1 tumors after treating with IgG Control or PAC-SABIs ($n = 5$ mice). **i** Number of differential genes enriched in GO term of 4T1 tumors treated with PAC-SABIs versus IgG Control ($n = 5$ mice). **j** GO enrichment analysis of the differential pathways in 4T1 tumors treated with PAC-SABIs versus IgG Control ($n = 5$ mice). **k** Volcano plot for the transcriptome sequencing of 4T1 tumors treated with PAC-SABIs versus IgG Control. Differentially expressed genes were calculated using a two-sided limma moderated $t$-test with Benjamini−Hochberg correction for multiple testing ($n = 5$ mice). Source data are provided as a Source Data file.

## Inhibition on BC and PC liver metastasis in vivo

To investigate if the ability of PAC-SABIs to inhibit cancer cell migration and invasion in vitro translates to decreased metastasis in vivo, we used an experimental liver metastasis model of BC and PC. 4T1-luc and PAN02-luc cells were injected into the spleens of BALB/c and C57BL/6 mice, respectively, which enabled monitoring and quantification of metastasized cancer cells in the liver with BLIs. Starting from the day before injection of cancer cells into the spleen, mice were given intravenous injections of IgG control, anti-CD24 mAb, SAMIs, and PAC-SABIs every other day for a total of 4 doses. The BLI intensity started to increase soon after injection of cancer cells in the IgG control and anti-CD24 groups, and patchy metastatic lesions formed in the liver region (Supplementary Fig. 44). However, a significant decrease in BLI intensity was observed in the PAC-SABIs group compared to the IgG control group after 2 weeks of injection (Supplementary Figs. 44 and 45). Then, the mice were euthanized, and reductions in the number and size of liver metastases were confirmed macroscopically in the PAC-SABIs group (Supplementary Fig. 44c, g). Additionally, H&E staining of the liver further demonstrated that PAC-SABIs facilitated construction of nanofibrous network barriers on the cancer cell membrane, inhibiting early metastasis and effectively stopping the growth of previously formed metastasis in mouse liver metastasis models (Supplementary Fig. 44d, h).

## Enhanced anti-PD-1 therapy after phagocytosis modulation

Based on the aforementioned in vivo experimental findings, regulating the phagocytosis of cancer cells by macrophages exhibited great potential in adaptive antitumor immune induction. Therefore, it is postulated that the combination of PAC-SABIs and PD-1 pathway blockade could potentially result in an augmented antitumor immune response in vivo, yielding optimal therapeutic outcomes. To investigate this, we constructed a subcutaneous BC xenograft model utilizing 4T1 cells. The mice were administered intravenously with PAC-SABIs or PBS every 2 days for a total of 4 doses, followed by anti-PD-1 or IgG control intraperitoneal injection on days 7, 10, 13, and 16. On the 21st day, the experiment concluded with the euthanasia of the mice, and the 4T1 tumors were surgically removed and the wet weight was recorded (Supplementary Fig. 46a). Throughout the treatment, we continuously monitored mouse weights and observed no significant differences among the groups (Supplementary Fig. 46b). Notably, the sequential PAC-SABIs and anti-PD-1 combination treatment exhibited the most substantial tumor growth inhibition, approximately 94.5% (Supplementary Fig. 46c, d). Then, we inoculated GFP$^+$ 4T1 cells into the third mammary fat pad of female BALB/c mice to construct the orthotopic BC model. The 4T1 tumor growth and survival rate were measured over time to assess the immunotherapeutic efficacy of different treatment regimens (Fig. 8a). The untreated and anti-PD-1 groups showed rapid tumor growth and poor median survival. Compared with these two groups, the PAC-SABIs group showed enhanced tumor growth inhibition and improved survival (Fig. 8b–d). Remarkably, anti-PD-1 therapy after PAC-SABIs activation of macrophages produced the greatest tumor suppression effect and achieved a 60-day optimal survival rate of 57% (Fig. 8d).

At the experimental end point, the lungs were dissected for analysis. Histological examination of representative lung sections using H&E staining showed a lower incidence of pulmonary metastatic nodules in the SAMIs group compared to the untreated and anti-PD-1 groups (Fig. 8e). More importantly, the combination of PAC-SABIs and anti-PD-1 therapy effectively impeded lung metastasis development, with few observable nodules. Given the promising results, flow cytometry was used to investigate macrophage phagocytic activity toward 4T1 cells. The combination treatment led to a substantial increase in phagocytosing CD45$^+$CD11b$^+$F4/80$^+$ macrophages compared to other groups (Fig. 8f). Further analysis using flow cytometry showed that macrophages in the combination treatment group had reduced CD206

expression, similar to the PAC-SABIs group, and a significant increase in CD86 expression compared to other groups (Supplementary Fig. 47). IF analysis of 4T1 tumor sections revealed an increase in CD8$^+$ T cells and NK1.1$^+$ cells, coupled with a decrease in FOXP3$^+$ T cell infiltration following the combination treatment (Fig. 8g and Supplementary Fig. 48). These findings highlighted the potential of PAC-SABIs to effectively activate macrophage phagocytosis, leading to an enhanced adaptive antitumor immune response and offering promising prospects for combined immunotherapy strategies.

## Discussion

In conclusion, we have developed a CD47 and CD24 bi-target inhibitor, referred to as PAC-SABIs, that can amplify an efficacious macrophage phagocytic capacity and antitumor immune response. The incorporation of active targeting and EISA imparts the PAC-SABIs with exceptional spatiotemporal control over bioactivity. The resulting nanostructured network scaffold rapidly forms a dynamic lichen-mimetic cell membrane coating, and thus, stimulates the anti-cancer cell phagocytic clearance by macrophages from both mouse and human sources in vitro. Upon confirming the substantial tumor growth inhibition by PAC-SABIs in mice with BC and PC, a series of immune responses are found. These encompass macrophages heightened antitumor phagocytosis, their reorientation to an M1-like phenotype, increased secretion of inflammatory cytokines, and enhanced infiltration of CD8$^+$ T cells within the typically immunosuppressive environment of the 4T1 tumor. Remarkably, following treatment with PAC-SABIs, we observe an activation of associated pathways of both innate and adaptive immune responses. There is a notable upregulation of genes that serve instrumental roles in immune regulation, inflammatory processes, macrophage phagocytosis, and antigen presentation.

In addition to validating our central hypothesis, we also demonstrate that introducing the anti-CD24 mAbs into the PAC-SABIs system propels the subsequent in situ self-assembly of peptides via ligand-receptor interactions. In comparison with the study by Barkal et al.[20], our regimen involving a lower dosage of solitary antibody therapy doesn't yield a significant tumor growth inhibition or survival benefits. Nevertheless, in vitro and in vivo studies have verified its efficacy in supplying adequate PAC-SABIs to obstruct innate ICs. Besides, with the designed phosphorylated peptide precursors and refined antibody to peptide ratio, PAC-SABIs exhibit long-term stability in the physiological circulatory system and ensure pronounced accumulation and retention in tumor sites. The negligible toxic side effects associated with CD24 and CD47 blockade therapies during PAC-SABIs treatment highlight their prospect for clinical application.

Overall, this work provides a perspective for the innate immune therapy of solid tumor. In our preliminary exploration, the combination of PAC-SABIs with sequential anti-PD-1 therapy has showed promising benefits in synergistic treatment strategies. Future studies might focus on the potential enhancement of this combination with other immunotherapeutic interventions such as varying IC inhibitors and adoptive cell therapy. Leveraging the structural versatility of the PAC-SABIs framework, it becomes viable to incorporate additional functional modules. This adaptability enables the rational design of a multifaceted modular platform, which holds the potential for tailoring therapy to individual immunotherapeutic needs.

## Methods

### Ethical statement

This research complies with all relevant ethical regulations. Experiments were performed in agreement with the Animal Experimentation Ethics Committee of Southern Medical University (LAEC-2022-077 and LAEC-2024-015).

Human peripheral blood from healthy volunteers and human PC specimens were sourced from Nanfang Hospital. Human

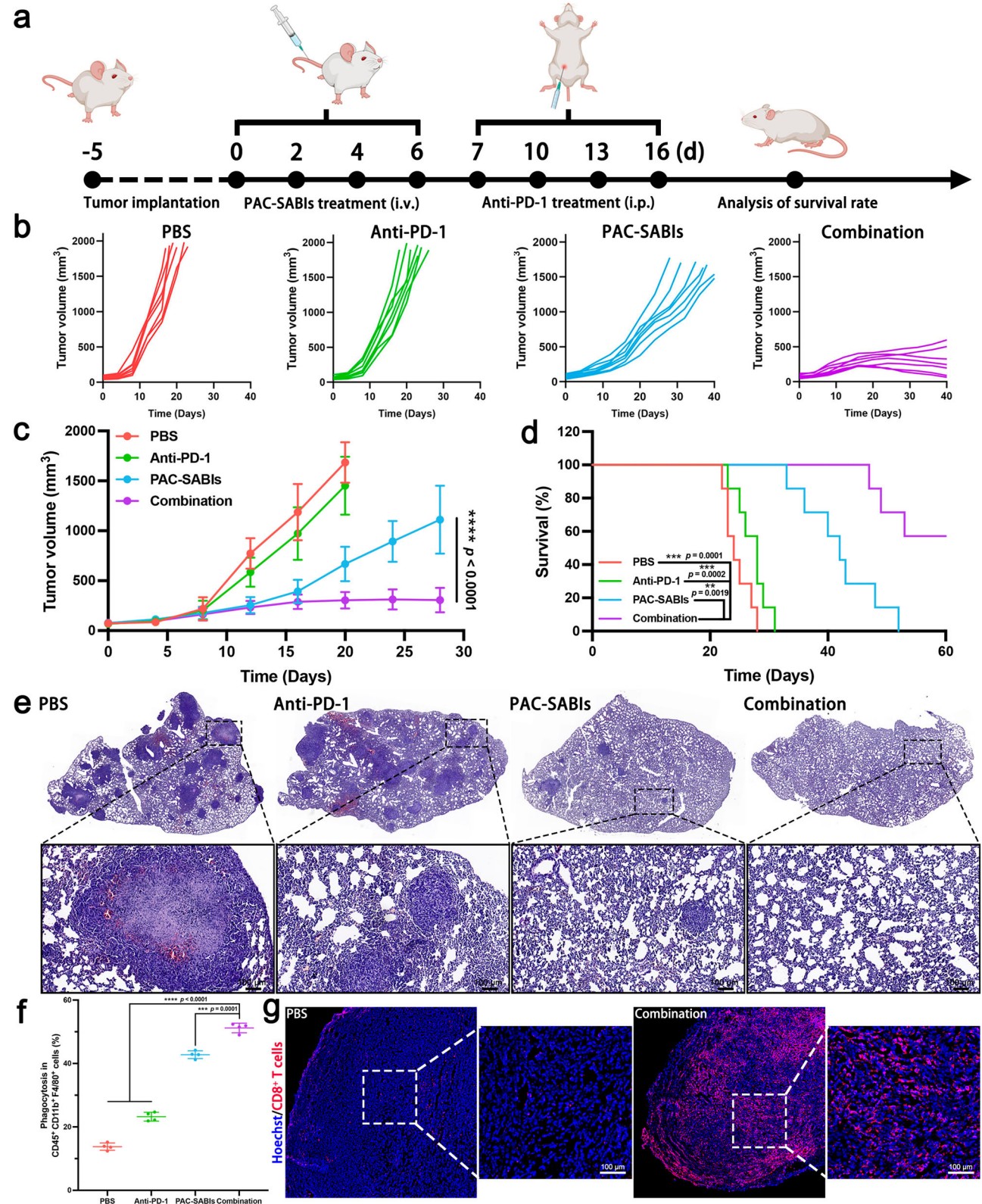

peripheral blood from BC patients and human BC specimens were sourced from Guangdong Provincial People's Hospital. Approvals were obtained from the institutional review board of Nanfang Hospital and Guangdong Provincial People's Hospital of Southern Medical University (NFEC-2023-309, KY2023-179-01 and KY2023-1143-02). The appropriate informed consent was obtained for all sample donors.

**Materials**

All chemicals and reagents were provided by commercial sources. Fmoc-amino acids were purchased from GL Biochem Co. Ltd. (Shanghai, China). 2-chlorotrityl chloride resin was obtained from Nankai Resin Co. Ltd (Tianjin, China). Dimethyl formamide (DMF), dichloride methane (DCM), dimethyl sulfoxide (DMSO), N-Methylmorpholine (NMM), O-(benzotriazol-1-yl)-N, N, N', N'-tetramethyluronium

**Fig. 8 | Enhanced anti-PD-1 therapy after phagocytosis modulation. a** Scheme of the combination therapeutic strategy. Figure was created with BioRender.com and released under a Creative Commons Attribution-NonCommercial-NoDerivs 4.0 international license. **b** Spider plots of individual tumor growth curves for 4T1-tumor-bearing BALB/c mice in each treatment group. **c** Average tumor growth curve of 4T1-tumor-bearing BALB/c mice in each treatment group. The error bars represent the mean ± SD ($n = 7$ mice; ****$p < 0.0001$; the $p$ value was analyzed by a two-tailed unpaired Student's $t$ test). **d** Kaplan–Meier survival curves of 4T1 tumor-bearing BALB/c mice treated with the indicated formulation. ($n = 7$ mice; **$p < 0.01$,

***$p < 0.001$; the $p$ value was analyzed by log-rank test). **e** Representative H&E staining of lungs harvested from the indicated treatment group. Scale bar: 100 μm. Three independent experiments were performed. **f** Quantitative analysis of phagocytosis in CD45$^+$CD11b$^+$F4/80$^+$ macrophages flow cytometry results. The error bars represent the mean ± SD ($n = 4$ independent experiments; ***$p < 0.001$, ****$p < 0.0001$; the $p$ value was analyzed by a two-tailed unpaired Student's $t$ test). **g** Representative IF images of tumor-infiltrated CD8$^+$ T cells. Scale bar: 100 μm. Three independent experiments were performed. Source data are provided as a Source Data file.

hexafluorophosphate (HBTU), trifluoroacetic acid (TFA), 1,2-ethane-dithiol, triisopropylsilane (TIPS), methanol, and acetonitrile were obtained from Aladdin technology Co. Ltd. (Shanghai, China). NHS-PEG$_{2000}$-Mal was acquired from Ruixi Co. Ltd. (Xi'an, China). The pHrodo-Red SE was purchased from Thermo Fisher Scientific (#P36600). Cy5.5-NHS was purchased from APExBIO Technology (#A8103). Recombinant mouse Siglec-G Fc protein (#10103-SL) and recombinant human Siglec-10 Fc protein (#2130-SL) were obtained from R&D Systems Inc. Antibodies used: IF analysis: anti-F4/80 antibody (10 μg/ml, Abcam, #ab6640, clone CI:A3-1), anti-CK19 antibody (2 μg/ml, Abcam, #ab52625, clone EP1580Y), anti-CD8α antibody (1:100, Invitrogen, #MA5-29682, clone 208), anti-CD24 antibody (1:50, Abcam, #ab290730, clone EPR26528-17), anti-CD47 antibody (1:400, Abcam, #ab300124, clone EPR24922-21), anti-NKR-P1C antibody (1:50, Abcam, #ab289542, clone EPR22990-31), anti-Foxp3 antibody (1:100, Affinity Biosciences, #AF6544), goat anti-rat IgG H&L (1:1000, Abcam, #ab150167), goat anti-rabbit IgG H&L (1:1000, Abcam, #ab150077). Treatment: anti-mouse CD24 antibody (BioXcell, #BE0360, clone M1/69), anti-human CD24 antibody (Novus Biologics, #NB100-64861, clone SN3), anti-mouse CD47 antibody (BioXcell, #BE0270, clone MIAP301), anti-mouse PD-1 antibody (BioXcell, #BE0273, clone 29F.1A12™), anti-CSF1R antibody (BioXcell, #BE0213, clone AFS98). Flow cytometry: PerCP/Cy5.5 anti-mouse CD45 antibody (1:200, Elabscience, #E-AB-F1136J, clone 30-F11), PE/Cyanine5 anti-mouse CD11b antibody (1:200, Biolegend, #101209, clone M1/70), APC anti-mouse F4/80 antibody (1:200, Biolegend, #123116, clone BM8), FITC anti-mouse CD206 antibody (1:200, Biolegend, #141703, clone C068C2), FITC anti-mouse CD86 antibody (1:200, Biolegend, #159219, clone A17199A), FITC anti-mouse CD3 antibody (1:200, BDbioscience, #555274, clone 17A2), APC anti-mouse CD4 antibody (1:200, BDbioscience, #553051, clone RM4-5), PE anti-mouse CD8a antibody (1:200, BDbioscience, #553033, clone 53-6.7), PE anti-mouse Foxp3 antibody (1:200, BDbioscience, #563101, clone R16-715). APC anti-human IgG Fc antibody (1:200, Biolegend, #410711, clone M1310G05), APC anti-mouse IgG1 Fc antibody (1:200, Biolegend, #400121, clone MOPC-21).

## Molecular synthesis and characterization

The peptides were synthesized according to the standard solid-phase peptide synthesis (SPPS) principle (Supplementary Figs. 1–10). Firstly, the C-terminus of the first amino acid was coupled to the 2-chlorotrityl chloride resin. Anhydrous DMF and piperidine in a volume ratio of 4:1 was used to remove the Fmoc-protecting group. The growth of the peptide chain was achieved by repeating Fmoc-cleavage and amino acid coupling steps. NMM and HBTU were used to activate the carboxylic group of the subsequent amino acid. Each coupling cycle was run for 120 min and rinsed with DMF for 5 times. The NBD-β-alanine or Cy5.5-NHS was linked to the N-terminal of peptide similar with other amino acid. The Dde protecting group on the side chain of K (Lysine) was removed using DMF with 2% hydrazine to expose NH$_2$, and then coupled with NHS-PEG$_{2000}$-Mal for 24 h. After the resin was washed with DMF, followed by rinsing with DCM for 5 times, the peptide derivatives were cleaved from the resin using a mixed cleavage reagent consisting of a 95% TFA solution with 2.5% water and 2.5% TIPS (v/v/v) for 30 min. The resulting mixture was evaporated through a rotary evaporator and precipitated with anhydrous cold ether. The

precipitate was collected after centrifugation at $1500 \times g$ for 15 min, and the resulting solid product was dissolved in H$_2$O/CH$_3$CN (1:1) for further purification by HPLC. HPLC purification was performed on an Elite P3500 Semi-preparative Integrated Liquid Chromatography System (Elite Analytical Instruments Co. Ltd., China) using a C18 RP column and the eluents CH$_3$CN/H$_2$O (0.1% TFA) with a linear gradient from 10%/90% to 50%/50%. The purified peptide was dissolved in 100 μl of acetonitrile (0.1 mg/ml) for mass spectrometry analysis. The mass spectrometry data were acquired from the QuanTOF I system mass spectrometer (Intelligene Biosystems Co. Ltd., China), and the parameters were set according to Supplementary Table 3. PAC-SABIs were obtained through a Michael addition reaction between the maleimide group on Pep-PEG and the thiol group on the anti-CD24 mAb. Specifically, Pep-PEG was first dissolved in 20 μl of DMSO and quickly injected into 980 μl of PBS (pH = 7.4) while stirring. After Pep-PEG formed a stabilized micellar structure using this rapid precipitation method, we added anti-CD24 mAb to the mixed solution system at various mAb to peptide ratios. The reaction mixtures were stirred at room temperature under light protection for 24 h to allow the coupling reaction to proceed. The resulting PAC-SABIs were stored in a PBS buffer at 4 °C for subsequent experiments.

## Fluorescence assays

The CACs of Pep and Pep-PEG were determined using the 1-phenylnaphthalene-8-sulfonate (ANS) fluorescence assay. ANS was dissolved in DMF to a concentration of 1 mM. Subsequently, 1 μl of ANS was added to 100 μl of varying concentrations of Pep and Pep-PEG. The resulting solution was transferred to a quartz cuvette and analyzed using a Perkin Elmer LS-55 fluorescence spectrophotometer (Perkin Elmer, USA). The CAC was determined from the matched curve, which was obtained by the ANS fluorescent intensities at 475 nm plotted against different concentrations of Pep and Pep-PEG.

The self-assembly process of Pep and Pep-PEG was monitored using NBD fluorescence. The fluorescence intensities of NBD-labeled Pep and Pep-PEG were measured at CAC and half of the CAC concentrations via a fluorescence spectrometer (excitation wavelength: 467 nm). To evaluate the ALP-triggered dephosphorylation and in situ self-assembly process, the NBD fluorescence intensity was detected after incubating Pep and Pep-PEG at CAC concentration with ALP (1 U/ml) at 37 °C for 1 h.

## CD spectroscopy

All the CD spectra analyses were recorded on a CD spectropolarimeter (Applied Photophysics Ltd, UK) in quartz cuvettes with an optical path length of 0.1 cm. Data were collected from 180 to 260 nm with a scanning speed of 100 nm/min at room temperature.

The DichroWeb server (http://dichroweb.cryst.bbk.ac.uk) and the unified ridge regression-based CONTINLL algorithm were utilized for a quantitative analysis of our peptide molecules' secondary structure from CD spectra.

## FTIR spectroscopy

All the FTIR analyses were carried out by a Nicolet 6700 FTIR spectrometer (Thermo Fisher Scientific, USA). A 100 μl solution of different formulations was spun down at $10,000 \times g$ for 1 h, and the pellet was

blown dry with nitrogen. The dry samples were mixed with KBr and pressed into pellets for further analysis.

## TEM characterization

Pep, Pep-PEG, and PAC-SABIs solutions (100 µM) with or without ALP (1 U/ml) incubation were prepared and placed at room temperature for morphology observation at different time points. Copper grids with a porous carbon mesh were used for sample preparation. Ten µl of the Pep, Pep-PEG and PAC-SABIs solution was placed onto a copper grid with a carbon membrane and left for 3 min, followed by removal of excess solutions with filter papers. Then, a small drop of uranyl acetate solution (2% weight/volume in water) was added to the copper grid for negative staining of samples, and the grid was blotted with filter paper after 1 min. Finally, the samples were left on filter paper overnight to facilitate further analysis.

## Cell culture and animal studies

Murine BC 4T1 cell line, murine PC PANO2 cell line, human BC MDA-MB-231 cell line, murine macrophage cell line RAW264.7, and human macrophage cell line THP-1 were purchased from the American Type Culture Collection (ATCC) and cultured according to the supplier's recommendations, supplemented with 10% fetal bovine serum (FBS) and antibiotics. The cell culture supernatant of BC cell lines 4T1 and MDA-MB-231 was collected as TCM, and was used to induce the transformation of RAW264.7 macrophages and phorbol myristate acetate (100 ng/ml)-treated THP-1 macrophages into TAMs, respectively.

C57BL/6 mice (male, 6 weeks old) and BALB/c mice (female, 6 weeks old) were purchased from Sijiajingda Biotechnology Co. The living environment of animals were maintained at a temperature of 22 °C and relative humidity range between 30–70% with a 12 h light/dark cycle, with free access to standard food and water. The tumor volume was calculated with following the formula: Volume = (Length × Width × Width)/2. The humane endpoints included tumor burden exceeding 10% of normal body weight, animal weight loss exceeds 20% of normal animal weight, ulcer at tumor growth point, and sustained self-mutilation in animals.

## BMDMs and HDDMs generation and stimulation

BMDM isolation was performed according to a previously published protocol with minor modification. After sacrificing male C57BL/6 mice and disinfecting the skin with 75% alcohol, the hind legs were cut off and placed in a sterile petri dish containing sterile and ice-cold PBS. The bone marrow was flushed with PBS using a syringe with a 25-gauge needle. The supernatant bone marrow cells were collected and then washed with PBS and resuspended with complete conditioned media for BMDM differentiation (100 ml complete medium consisted of 74 ml Dulbecco's modified Eagle's medium (DMEM) + 15 ml macrophage colony-stimulating factor (25 ng/ml) + 10 ml FBS + 1 ml penicillin-streptomycin solution (PS)), seeded on tissue culture plates, and incubated at 37 °C with 5% $CO_2$. Primary HDDMs were generated from venous blood of three healthy volunteers (male, aged 26, 27, and 30 years old respectively), diluted with PBS (pH = 7.4) and purified through successive density gradients using Ficoll (Sigma-Aldrich) and Percoll (GE Healthcare). Monocytes were then differentiated into macrophages by culture in Iscove's modified Dulbecco's medium (IMDM) + 10% AB human serum (Life Technologies) for 7–10 days. To stimulate macrophages with TCM, we cultured BMDMs and HDDMs with complete medium containing 50% TCM from 4T1 and MDA-MB-231 cells, respectively.

## Characterization of the self-assembly process of PAC-SABIs in 2D and 3D cell culture environments

4T1 or PANO2 cells were inoculated and incubated overnight at a density of $1 \times 10^5$ cells/dish in laser confocal petri dishes. NBD-labeled

PAC-SABIs (100 µM) were co-incubated with 4T1 or PANO2 cells for 30, 60, and 120 min, followed by washing with PBS three times. Then, the NBD fluorescence at different time points was observed by CLSM imaging. To reveal the self-assembly process of PAC-SABIs on cancer cell membranes, 4T1 ($1 \times 10^5$ cells/dish) cells were cultured with NBD-labeled PAC-SABIs (100 µM) for 120 min. After 3 rounds of washing with PBS, the membrane dye CellTracker CM-Dil (1 mM; Invitrogen, USA) was added and incubated for 5 min at 37 °C, and then for an additional 15 min at 4 °C. CLSM imaging was used to detect the spatial distribution of NBD and Dil fluorescence.

Cy5.5-labeled PAC-SABIs (100 µM) was co-incubated with 4T1 3D spheroids for 90 min, followed by washing with PBS three times. Then, the outermost cells and nuclei of the spheroids were stained with Calcein-AM and Hoechst 33342, respectively. Finally, CLSM imaging was used to detect spatial distributions of fluorescence signals with Cy5.5, FITC and DAPI channels in spheroids and different focal plane along the z axis.

## Detection of cell surface distribution using bio-TEM

The 4T1 and PANO2 cells were seeded on cell culture dishes with a diameter of 10 mm at a density of $1 \times 10^5$ cells/dish and cultured overnight, followed by replacement with fresh culture medium containing PAC-SABIs at a concentration of 100 µM for 120 min. Then, the PAC-SABIs-containing medium was removed and the cells were washed three times with PBS. The cells attached to the bottom of the dishes were gently harvested using a cell scraper and collected in a centrifuge tube. After centrifugation, the supernatant was carefully discarded, and then a 4 °C precooled fixative was slowly added along the tube wall and then placed in a 4 °C refrigerator overnight. The samples were then fixed with a 1% osmium acid solution for 1 h followed by dehydration using a gradient concentration of ethanol. The samples were embedded and sliced with a LEICA EM UC7 ultra-thin microtome to obtain 70–90 nm slices. The sections were stained with lead citrate solution, uranyl acetate, and 50% ethanol saturated solution for 5–10 min, respectively, and then observed under TEM imaging.

## SEM analysis

4T1 and PANO2 cells were seeded into 12-well plates containing cover glass slips at a density of $1 \times 10^5$ cells/well and cultured overnight. PAC-SABIs were added into each well at the final concentration of 100 µM and cultured for 120 min. Then, PAC-SABIs-treated 4T1 and PANO2 cells were dehydrated for the time course of 24 h, processing from 30% to 100% ethanol. The samples were then sputter coated with gold (CRC-150 Sputter Coater, USA) and imaged using a Zeiss Ultra 55 SEM (Carl Zeiss, Germany).

For SEM imaging of the PAC-SABIs assembled morphologies within the subcutaneous 4T1 and PANO2 xenografts, the tumor-bearing mice received intravenous injections of PAC-SABIs (15 mg/kg) every other day for a total of four treatments, and the xenografts were harvested on the day after final administration. The xenografts were cut into approximately 200 µm thin slices, then fixed in 2.5% glutaraldehyde and dehydrated through a series of ethanol washes, and examined by SEM. Regarding the freshly acquired clinical BC samples, the tissues were similarly sectioned. The slices were incubated with PAC-SABIs in a 96-well plate at 37 °C for 120 min. Subsequent fixation and dehydration were identical to that of the xenografts, and SEM imaging was performed.

## Anti-CD24 treatment and recombinant Siglec binding assay

In total, 100,000 non-fluorescently labeled 4T1 or PANO2 cells were treated with the anti-CD24 antibody for 1 h at 37 °C in serum-free medium. In parallel, MDA-MB-231 cells underwent identical treatment using the anti-CD24 antibody. All antibodies were added at a concentration of 10 µg/ml. Recombinant Siglec-G (ortholog of human Siglec-10) and Siglec-10 proteins were acquired in the forms of mouse

and human Fc fusion proteins respectively. The Binding of recombinant Siglecs versus isotype IgG1 control was assayed at a concentration of 100,000 cells/1 mg/ml at 37 °C for 1 h. Subsequently, 4T1, PAN02, or MDA-MB-231 cells were stained with fluorescently-conjugated anti-mouse/human Fc antibodies for recombinant Siglec binding analysis via flow cytometry.

## Phagocytosis assay using flow cytometry

The in vitro phagocytosis assays described in this study were performed by co-culture GFP$^+$ 4T1 cells and macrophages at a ratio of 100,000 target cells to 50,000 macrophages for 120 min in a humidified, 5% $CO_2$ incubator at 37 °C in ultra-low-attachment 96-well U-bottom plates (Corning, USA) in serum-free IMDM. 4T1 cells with endogenous fluorescence were harvested from plates using TrypLE Express (Life Technologies, Poland) and treated with PAC-SABIs for 120 min prior to co-culture. After co-culture, phagocytosis assays were stopped by placing plates on ice, centrifuged at $400 \times g$ for 5 min at 4 °C and stained with anti-CD11b to identify macrophages. Assays were analyzed by flow cytometry on a Sony SA3800 Flow Cytometer (Sony Biotechnology, Japan) or a CytoFLEX (Beckman, USA). Phagocytosis was measured as the number of CD11b$^+$, GFP$^+$ macrophages, quantified as a percentage of the total CD11b$^+$ macrophages.

## Phagocytosis assay using live-cell microscopy

Non-fluorescently labeled 4T1, PAN02, or MDA-MB-231 cells were harvested and labeled with pHrodo Red SE as per manufacturer instructions at a concentration of 1: 30,000 in PBS for 1 h at 37 °C, followed by two washes with DMEM + 10 % FBS + 100 U/ml PS. In total, 50,000 macrophages were added to a transparent 96-well plate and allowed to adhere at 37 °C. After macrophage adherence, 100,000 pHrodo-Red-labeled 4T1, PAN02, or MDA-MB-231 cells pretreated with PAC-SABIs for 120 min were added in serum-free IMDM. The plate was centrifuged gently at $50 \times g$ for 2 min in order to promote the timely settlement of 4T1, PAN02, or MDA-MB-231 cells into the same plane as adherent macrophages. Phagocytosis events were calculated as the number of pHrodo red$^+$ events per visual field and the fluorescent signals were captured by a Nikon AX confocal laser microscope (Nikon, Japan).

## In vivo and ex vivo fluorescence imaging

To obtain breast and pancreatic subcutaneous xenografts, $1 \times 10^6$ 4T1 or PAN02 cells were implanted into the right hind limb of BALB/c and C57BL/6 mice, respectively. When the tumor size reached ≈ 50 mm³, mice were randomly sorted into free Cy5.5, $^{Cy5.5}$SAMIs, and $^{Cy5.5}$PAC-SABIs groups. The 4T1 and PAN02 tumor-bearing mice were correspondingly intravenously injected with free Cy5.5, $^{Cy5.5}$SAMIs, and $^{Cy5.5}$PAC-SABIs (the dose of Cy5.5 was 1 mg/kg) for in vivo fluorescence imaging with IVIS system (Perkin Elmer, USA). Then, the mice were sacrificed to collect the major organs (heart, liver, spleen, lung, and kidney) and tumors for ex vivo fluorescence imaging. Finally, the 4T1 and PAN02 tumors were embedded in Tissue-Tek OCT compound (Sakura, Japan), and cryosections of 8 μm thickness were prepared. The Cy5.5 signal was detected using a fluorescence microscope (Nikon, Japan).

## Blood examination and histology

To record the complete blood count data, healthy BALB/c mice were intravenously injected with PAC-SABIs at a dose of 15 mg/kg every other day for a total of 4 administrations. The mice were sacrificed before blood collection (0.5 ml), and complete blood count evaluations at 1, 7, 14 and 21 days postinjection of PAC-SABIs were carried out at the Nanfang Hospital of Southern Medical University. After 1 month following treatment with PAC-SABIs, the mice were sacrificed and the major organs (heart, liver, spleen, lung and kidney) were collected. The organs were immersed in a 4% paraformaldehyde solution for an overnight fixation period, followed by dehydration in a 25% sucrose solution. Subsequently, the fixed tissues were sliced into sections with a thickness of 8 μm, and the sections were stained with H&E (Beyotime Biotech, China) as per the manufacturer's instructions. Finally, a microscope was employed to examine the samples for any histological alterations.

## Antitumor treatment studies in subcutaneous xenograft tumor models

To establish BC and PC subcutaneous xenografts, $1 \times 10^6$ 4T1 and PAN02 cells were implanted into the right hind limb of each BALB/c and C57BL/6 mouse, respectively. Upon the tumor reaching an approximate volume of 50 mm³ by the 7th day post-implantation, the mice were randomized into groups. For evaluation of PAC-SABIs antitumor efficacy, the PAC-SABIs (15 mg/kg) treatment began 7 days after the tumor implantation. Each group received its respective intravenous treatment every other day for a total of four sessions. On the 28th day post-implantation, the mice were euthanized and subsequently dissected for tumor weight analysis.

To compare the antitumor efficacy of PAC-SABIs with the dual antibody combination therapy, three groups of mice received tail vein injections, with the respective treatments being IgG control, a combination of anti-CD47 (20 mg/kg) and anti-CD24 antibodies, and PAC-SABIs. On the 28th day post-implantation, the mice were euthanized and subsequently dissected for tumor weight analysis.

For evaluation of sequential PAC-SABIs and anti-PD-1 combination treatment efficacy, the mice were administered intravenously with PAC-SABIs or PBS every 2 days for a total of 4 doses, followed by anti-PD-1 (10 mg/kg) or IgG control intraperitoneal injection on days 7, 10, 13, and 16. On the 28th day post-implantation, the experiment concluded with the euthanasia of the mice. The 4T1 tumors were surgically removed and the wet weight was recorded.

## Antitumor treatment studies in orthotopic xenograft tumor models

To establish orthotopic xenograft tumor models, 4T1-luc ($1 \times 10^6$) suspended in a 25 μl PBS and Matrigel (Corning, USA) mixture (1:1, v/v) was injected into the right third mammary fat pad of BALB/c mice. PAN02-luc ($1 \times 10^6$) suspended in a 25 μl PBS and Matrigel mixture (1:1, v/v) were injected into the tail region of the pancreatic parenchyma of C57BL/6 mice. The PAC-SABIs treatment began 5 days after the tumor implantation, and the BLIs of orthotopic 4T1-luc and PAN02-luc tumor-bearing mice were detected every 7 days to monitor the tumor growth in each treatment group. For BLI, mice were given D-luciferin potassium salt (150 mg/kg) intraperitoneally and imaged 10 min later in an IVIS system. Survival (in days) of mice in the different treatment groups were monitored throughout the period of study.

In order to study the amplified antitumor immune therapy, 4T1 ($1 \times 10^6$) suspended in a 25 μl PBS and Matrigel mixture (1:1, v/v) was injected into the right third mammary fat pad of BALB/c mice. On the 5th day, 4T1 tumor-bearing mice were randomly divided into 4 groups, and were immunized with PAC-SABIs (15 mg/kg) or PBS on days 5, 7, 9, and 11 through intravenous injection. Then, intraperitoneal injection with or without anti-PD-1 antibody (10 mg/kg) was performed on days 12, 15, 18, and 21. Mouse body weights and tumor sizes (length and width measured by calipers) were measured every other day. Survival (in days) of mice in the different treatment groups were monitored throughout the period of study. To investigate the role of PAC-SABIs in prevention of lung metastasis, the histological examination of representative lung sections was conducted using H&E staining at the endpoints of different treatment procedures.

## In vivo macrophage depletion treatment study

BALB/c mice were intraperitoneally treated with 400 μg anti-CSF1R antibody per mouse or PBS (vehicle) thrice a week for 2 weeks.

Successful tissue-resident macrophage depletion was confirmed prior to tumor engraftment by peritoneal lavage and flow cytometry analysis. The macrophage-depleted and vehicle treated mice were then randomized and subsequently engrafted with 4T1-Luc cells, followed by administration of PAC-SABIs as previously described. BLIs were detected every 7 days to monitor the tumor growth in each treatment group. Survival (in days) of mice in the different treatment groups were monitored throughout the period of study.

### Analysis of immune response in tumor

The 4T1 tumor-bearing mice were euthanized, and tumor tissues were collected and frozen in optimal cutting temperature medium on dry ice. Tumor sections were cut using a cryotome, mounted on slides and stained with different primary antibodies overnight at 4 °C according to the manufacturer's instructions. Following the addition of fluorescently labeled goat anti-rat IgG H&L and goat anti-rabbit IgG H&L, the slides were analyzed with a confocal microscope.

For flow cytometry analysis, the 4T1 tumors were cut into small pieces and homogenized in cold staining buffer to form single-cell suspensions in the presence of digestive enzyme. Cells were stained with different fluorescence-labeled antibodies following the manufacturer's instructions. The stained cells were measured on a Sony SA3800 Flow Cytometer or a CytoFLEX. The numbers presented in the flow cytometry analysis images were percentage based.

For transcriptomics study, the 4T1 tumors after different treatments were collected and stored quickly in the liquid nitrogen. RNA-seq libraries were prepared according to the manufacturer's protocol. Briefly, a total amount of 1 µg RNA per sample was used as input material for the RNA sample preparations. Sequencing libraries were generated using NEBNext®UltraTM RNA Library Prep Kit for Illumina® (NEB, USA) and index codes were added to attribute sequences to each sample. The clustering of the index-coded samples was performed on a cBot Cluster Generation System using TruSeq PE Cluster Kit v3-cBot-HS (Illumina) according to the manufacturer's instructions. After cluster generation, the library preparations were sequenced on an Illumina Novaseq platform and 150 bp paired-end reads were generated. Reference genome GRCm39 (Genome Reference Consortium Mouse Reference 39) and gene model annotation files were downloaded from genome website. Index of the reference genome was built using Hisat2 v2.0.5 and paired-end clean reads were aligned to the reference genome. In this study, all the analyses were performed based on the Bioinforcloud platform (http://www.bioinforcloud.org.cn).

### Statistical analysis

Data are presented as mean ± standard deviation (SD). Unpaired two-tailed Student's $t$ test was used for two-group comparisons, and Kaplan–Meier curves were used to analyze the survival study. Statistical significance was defined as $*p < 0.05$, $**p < 0.01$, $***p < 0.001$ and $****p < 0.0001$. Statistical analysis was performed using GraphPad Prism version 10.2.1 and SPSS version 22.0. The number of replicates performed is indicated in each figure legend, where applicable.

### Reporting summary

Further information on research design is available in the Nature Portfolio Reporting Summary linked to this article.

## Data availability

RNA-seq data that support the findings of this study have been deposited in the NCBI Sequence Read Archive under the BioProject ID PRJNA1124390. RNA-seq data based on the TCGA and Genotype-Tissue Expression (GTEx) projects, obtained from the Gene Expression Profiling Interactive Analysis 2 (GEPIA2, http://gepia2.cancer-pku.cn) database, were used to investigate the expression of CD47 and CD24 in tumor and paired normal tissue, and to conduct survival analysis. The remaining data generated in this study are available within the Article, Supplementary Information or Source Data file. Source data are provided with this paper.

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

## Acknowledgements

We acknowledge the financial support from National Natural Science Foundation of China (51973090 to Y.C., 32271372 to Y.C., 31900952 to Y.C., 32101058 to J.Z., 82273256 to N.L.), Guangdong Basic and Applied Basic Research Foundation (2023A1515012734 to J.Z., 2023A1515110674 to W.Z.), Science and Technology Projects of Guangzhou (2024A04J5154 to J.Z., 2024A04J4490 to W.Z.). China Postdoctoral Science Foundation (2023M740768 to W.Z.). Postdoctoral Fellowship Program of CPSF to W.Z.

## Author contributions

W.Z., N.L. and Y.C. conceived the study and designed the experiments. W.Z., Q.X. and J.Z. carried out the synthesis and characterization of the PAC-SABIs. W.Z. and Y.L. collected clinical data and samples. W.Z., H.W. and N.L. performed single-cell RNA sequencing and analysis. W.Z., Y.Z., Y.W., J.L. and Y.L. performed all the in vitro and in vivo experiments. W.Z., Y.Z. and Y.C. discussed the results, interpreted the data, wrote and revised the paper. J.Z., N.L. and Y.C. supervised the research. All authors read and approved the final version of the manuscript.

## Competing interests

The authors declare no competing interests.
