## [Peer Review File · Nature Communications]

An in-situ peptide-antibody self-assembly to block CD47 and CD24 signaling enhances macrophage-mediated phagocytosis and anti-tumor immune responsesREVIEWER COMMENTS

Reviewer #1 (Remarks to the Author):

In this work, the authors have presented an ALP-responsive peptide-antibody combi-supramolecule (PAC-SABI) to achieve in situ assemble CD47 and CD24 bi-target inhibitors. The design of this work is reasonable and experiment data is abundant, while there are some important issues should be clarified as following:

- 1) What's the different between this work and other works on enzyme-instructed peptides for cancer immunotherapy? There are many papers have published similar works, including J Am Chem Soc 2023, 145, 4366, J Am Chem Soc 2016, 138, 3813...
- 2) What about the secondary structure calculation of Pep, Pep-PEG, and PAC-SABIs, the details need to be offered.
- 3) In Fig. 1I, the authors claimed that "nearly all precursors transformed into nanofibers with a diameter of 13.2 ± 2.9 nm". However, PAC-SABIs treated with ALP for 120 min exhibited reticular polymer structure, and the nanofibers were less conspicuous than 60 min. Did the nanofibers from PAC-SABIs aggregate with each other and form a larger polymer framework?
- 4) The pictures in this manuscript were blurry, the authors need to offer higher-definition version.
- 5) In Fig. 5B and 5C, the authors claimed that the TCM-stimulated HDDMs exhibited upregulation of the M2 marker CD206. Therefore, did the treatment with PAC-SABI of MDA-MB-231 cells result in repolarization of macrophage due to the blockage of CD24 and CD47? More evidence should be offered.
- 6) In Fig. 7, the authors should offer the physical images of tumor tissue and the data on body weight change in mice et al.
- 7) In Fig. 9, the enhanced anti-PD-1 therapy exhibited strong repression of tumor growth. But, why PAC-SABIs group shows more slight tumor growth inhibition? Therefore, the physical images of tumor tissue should be supplemented. And did the combination therapy possess a higher repolarization ratio of macrophage? How about other tumor-infiltrating lymphocytes?
- 8) In this manuscript, the authors designed an ALP-responsive PAC-SABI, and the

introduction of the PEG2000 chain obviously prolonged circulation in blood. How did the PAC-SABI be stable in the blood because of the presence of ALP in human serum? And what about the threshold of ALP of PAC-SABI?

9) The manuscript still contains spelling and formatting errors, the authors should check it carefully.

Reviewer #2 (Remarks to the Author):

In this interesting study by Zhang et al, the authors have presented an innovative peptide-antibody combo-supramolecular in situ assembled CD47 and CD24 bi-target inhibitor (PAC-SABI). PAC-SABIs demonstrated robust effects in stimulating tumour cell phagocytosis. When tested in in vivo tumour models, PAC-SABIs demonstrated impressive tumour inhibitory effects on tumour growth and stimulated T cell response, and a combination of PAC-SABIs with PD1 therapy significantly improved the survival of mice with 4T1 tumors.

Major concerns:

1. Information regarding anti-mouse CD24 (for 4T1 and PAN02) and anti-human CD24 (for MDA-MB-231) used in the study is missing. Whether these antibodies effectively block CD24-Siglec10 interaction/signaling should be verified.
2. CD47 and CD24 blockade with blocking antibodies has been reported in a number of previous studies in inhibiting tumor growth. It's unclear whether PAC-SABIs would exhibit superior antitumour effects compared to a combination of CD47 and CD24 blocking antibodies. An in vivo experiment should be performed to directly compare the anticancer effects of PAC-SABIs with that of a combination of CD47 and CD24 blocking antibodies.
3. In situ self-assembly of PAC-SABIs was examined in an in vitro context, as depicted in Fig3. However, it is important to extend this analysis to the tumour microenvironment to confirm the validity of this process in TME.
4. The authors have showed PAC-SABIs induced strong phagocytosis of cancer cells in vitro.

A macrophage-depletion assay (by anti-CSF1R antibody or clodronate) should be performed in the in vivo models to verify the anticancer effects of PAC-SABIs are attributable to enhanced tumor cells phagocytosis by macrophages.

Minor:

Fig2H, the expression of Sirpa and Siglec10 should be investigated.

Statistical methods used for each figure should be indicated.

FigS19, BLI imaging should be quantified.

Reviewer #3 (Remarks to the Author):

The authors' goal is to make use of a complex combination of peptides, a polymer (PEG) and two antibodies against CD24 and CD47 or IAP as a combined-supramolecular inhibitor structure to tackle cancer immunotherapy. The idea is interesting and the biological data including the in vivo studies seem to be promising.

When focusing solely on the molecular design and self-assembly behavior, it is quite challenging to follow the assumption of the authors how the bi-target inhibitor (PAC-SABI) is assembling to its final proposed structure by a molecular self-assembly process. Many questions would need to be answered to trust that the proposed inhibitor structure is indeed the suggested structure, following the proposed mechanism as depicted in Scheme 1.

The authors believe that the beta-amyloid derived peptide KLVFF is the driving force behind the molecular assembly of PAC-SABI. In fact, when comparing the size of the 5-mer peptide with the overall size and structure of the monomer, it rather looks that the amyloid-derived peptide is somehow buried within the monomer structure and might not be easily accessible to drive the self-assembly process. To convincingly prove this assumption, the amyloid motif KLVFF should be exchanged with another hydrophobic glycine or alanine 5-mer sequence leading to an almost identical micelle structure. Also, the typical cross-beta structure of amyloids should be expected to be found and an easy monitoring with fluorescent dye thioflavine T showing an intrinsic fluorescence should give an answer to the importance of the amyloid beta motif for the self-assembly of the inhibitor structure.

Was the known fact of amyloids' self-assembly features the rationale beta-amyloid peptide motif KLVFF was used?

In case, the amyloid motif is the peptide backbone that drives the assembly of the suggested micelle structure how can it be understood that the second 12-mer peptide sequence motif AWSATWSNpYWRH (Pep-20) and the attached PEG 2000 motif is not interfering with the self-assembly process.

Assuming the Co-assemblies of the antibodies with the micelles are occurring as proposed, the final step of alkaline phosphatase (ALP)-driven formation of PAC-SABIs and Pep fibers remain a mystery. The suggestion is made that phosphorylated tyrosine pTyr residues promote the nanofiber assemblies. Both structures Pep and PAC-SABIs are significant different in their molecular structure and composition, but the nanofiber formation interestingly shows similarities. Can this be explained?

It is mentioned that a Michael addition is happening. A more profound explanation of the proposed chemical reaction would be desired.

TEM micrographs are shown, but SEM micrographs should give an additional important information about potential scaffold formation.

The CD spectroscopy analysis of secondary structural motifs, see Figure 1, shows a mixture of all kinds of secondary motifs such as beta sheet, beta turn, alpha helix, and random structure. It is not surprising that such a mixture of secondary structures in a single molecule would give rise to a globular structure. The question is then, if the proposed micelle is nothing else than an aggregate of globular molecule structures and not an aligned self-assembled micelle. Why is there no significant difference in the secondary motif composition when comparing Pep and Pep + ALP, or PAC-SABIs and PAC-SABIs + ALP? Supposedly, Pep and PAC-SABIs is showing globular features, whereas the presence of ALP is introducing nanofiber structures which are by fundamentally different to globular structures, i. e. fibers being anisotropic whereas globular structures are isotropic. This kind of different structural appearance would not allow a similar CD spectrum with similar secondary motif compositions.

In summary, the assembled complex inhibitor structures might support the clinical therapeutic goal of a cancer immunotherapy, but there are substantial doubts on the proposed combo structures. Overall, much more meticulous characterization and proof-of-concept is needed to believe in the suggested mechanism of action of the agglomerate or aggregate.

Title: “Enzyme-Instructed Peptide Self-Assembly as A Cell Membrane Lichen Activating Macrophage-Mediated Cancer Immunotherapy” (**Manuscript ID: NCOMMS-23-40650**)

In answer to the comments raised by Reviewer #1

Comment 1: What’s the different between this work and other works on enzyme-instructed peptides for cancer immunotherapy? There are many papers have published similar works, including J Am Chem Soc 2023, 145, 4366, J Am Chem Soc 2016, 138, 3813...

Response to comment 1: Thank you for your beneficial comments. As you pointed out, the field of enzyme-instructed self-assembly (EISA) of peptides has experienced rapid growth in recent years. Fundamental principles have been established in key studies, demonstrating potential applications and laying the groundwork for the PAC-SABIs described in our manuscript. Initial efforts in EISA systems focused on developing basic concepts and components, along with seminal studies confirming the feasibility of directing in situ assembly through EISA¹⁻⁵. Later, research efforts expanded to explore ways of enhancing control over the self-assembly process and the resulting supramolecular structures⁶⁻⁸. Furthermore, combining EISA with targeting ligands, disease-penetrating peptides, or immunomodulatory domains enabled the development of advanced material designs^{9,10}. Integrating molecular recognition, enzymatic conversion, and controllable hierarchical self-assembly, these strategies showed promise in fulfilling EISA’s potential for personalized therapeutics⁶.

A significant advance in this field has been the development of EISA systems interfacing with biological entities like lipids and proteins. It has been shown that hybrid peptide precursors undergo EISA at membrane surfaces¹¹. This discovery has opened avenues for cell membrane-targeted applications, underscoring the potential of EISA in precision engineering at biointerfaces¹². Despite these advancements, substantial room exist to build upon these initial proofs-of-concept and realize enhanced therapeutic efficacy in biomedical settings.

Our current work represents an innovative extension of preceding studies,

incorporating several key considerations: the interaction between CD47 and SIRP α transmits a “don’t eat me” signal, while the CD24-Siglec-10 axis similarly mitigates macrophage phagocytosis. The PAC-SABIs integrated complementary functional units into a modular platform. Within this framework, the anti-CD24 antibody targeted its receptor with precision, while the peptide concurrently orchestrated CD47 inhibition. This synergistic combination of distinct active components represented a dual inhibition against two discrete immunosuppressive pathways modulated by CD47 and CD24. Contrasting with most existing approaches that focused solely on blocking CD47, the simultaneous obstruction of these inhibitory axes via PAC-SABIs empowered macrophages to identify and eradicate neoplastic cells more efficiently, thereby intensifying the tumor’s immunogenicity and broadening the spectrum for enhancing adaptive immune responses. Additionally, EISA imparted the PAC-SABIs with exceptional spatiotemporal control over bioactivity. The phosphorylated peptide precursor remained inert until triggered by alkaline phosphatase (ALP) overexpression at the tumor site. This programmability ensured therapeutic accuracy and avoided widespread systemic effects, which was critical in reducing the serious side effects of CD47 blockade therapy in clinical practice. The resulting nanostructured network scaffold formed a dynamic lichen-mimetic cell membrane coating, augmenting immunomodulatory processes at target loci in a localized manner. In summary, our rationally designed modular system establishes supramolecular peptide engineering as a facile yet powerful approach to merge multitargeting, programmability, and spatial control within a singular nanoscale platform for more efficacious next-generation immunotherapies.

We sincerely appreciate the insightful comments offered by you and Reviewer #2 and Reviewer #3. These insights have significantly deepened our comprehension of the innovative dimensions of our work. Informed by this enriched perspective, the modifications for the Discussion section have been made in the resubmitted manuscript.

Reference

1. Gao, Y., Shi, J., Yuan, D. & Xu, B. Imaging enzyme-triggered self-assembly of small molecules inside live cells. *Nat. Commun.* **3**, 1033 (2012).

2. Petkau-Milroy, K. & Brunsveld, L. Supramolecular chemical biology; bioactive synthetic self-assemblies. *Org Biomol Chem* **11**, 219–232 (2013).
3. Tanaka, A. *et al.* Cancer Cell Death Induced by the Intracellular Self-Assembly of an Enzyme-Responsive Supramolecular Gelator. *J. Am. Chem. Soc.* **137**, 770–775 (2015).
4. Gao, Y., Zhao, F., Wang, Q., Zhang, Y. & Xu, B. Small peptide nanofibers as the matrices of molecular hydrogels for mimicking enzymes and enhancing the activity of enzymes. *Chem. Soc. Rev.* **39**, 3425 (2010).
5. Callmann, C. E. *et al.* Therapeutic Enzyme-Responsive Nanoparticles for Targeted Delivery and Accumulation in Tumors. *Adv. Mater.* **27**, 4611–4615 (2015).
6. Gao, J. *et al.* Enzyme Promotes the Hydrogelation from a Hydrophobic Small Molecule. *J. Am. Chem. Soc.* **131**, 11286–11287 (2009).
7. Wang, Z. *et al.* Narrowing the diversification of supramolecular assemblies by preorganization. *Chem. Commun.* **54**, 2751–2754 (2018).
8. Shi, J., Fichman, G. & Schneider, J. P. Enzymatic Control of the Conformational Landscape of Self-Assembling Peptides. *Angew. Chem. Int. Ed.* **57**, 11188–11192 (2018).
9. Wang, H. *et al.* Integrating Enzymatic Self-Assembly and Mitochondria Targeting for Selectively Killing Cancer Cells without Acquired Drug Resistance. *J. Am. Chem. Soc.* **138**, 16046–16055 (2016).
10. Jeena, M. T. *et al.* Mitochondria localization induced self-assembly of peptide amphiphiles for cellular dysfunction. *Nat. Commun.* **8**, 26 (2017).
11. Ding, Y. *et al.* Enzyme-Instructed Peptide Assembly Favored by Preorganization for Cancer Cell Membrane Engineering. *J. Am. Chem. Soc.* **145**, 4366–4371 (2023).
12. Gao, J., Zhan, J. & Yang, Z. Enzyme-Instructed Self-Assembly (EISA) and Hydrogelation of Peptides. *Adv. Mater.* **32**, 1805798 (2020).

Changes in the manuscript on page 32 and 33 (Discussion section)

In conclusion, our designed PAC-SABIs integrated complementary functional units into a modular platform. Within this framework, the anti-CD24 mAb targeted its

receptor with precision, while the peptide concurrently orchestrated CD47 inhibition. This synergistic combination of distinct active components represented a dual inhibition against two discrete immunosuppressive pathways modulated by CD47 and CD24. Contrasting with most existing approaches that focused solely on blocking CD47, the simultaneous obstruction of these inhibitory axes via PAC-SABIs empowered macrophages to identify and eradicate neoplastic cells more efficiently, thereby intensifying the tumor's immunogenicity and broadening the spectrum for enhancing adaptive immune responses. Additionally, EISA imparted the PAC-SABIs with exceptional spatiotemporal control over bioactivity. The phosphorylated peptide precursor remained inert until triggered by ALP overexpression at the tumor site. This programmability ensured therapeutic accuracy and avoided widespread systemic effects, which was critical in reducing the serious side effects of CD47 blockade therapy in clinical practice. The resulting nanostructured network scaffold formed a dynamic lichen-mimetic cell membrane coating, augmenting immunomodulatory processes at target loci in a localized manner. Thus, the *in vivo* construction of PAC-SABIs-based nanoarchitectonics epitomizes a pioneering immunotherapy approach in bridging innate and adaptive immunity to maximize therapeutic potency.

Comment 2: What about the secondary structure calculation of Pep, Pep-PEG, and PAC-SABIs, the details need to be offered.

Response to comment 2: Thank you for your rigorous comments. We utilized the DichroWeb server to determine the secondary structure of peptide molecules based on circular dichroism (CD) spectra. This server has been used in a wide range of structural biology studies, with more than one million analyses undertaken thus far, rendering it the most widely-cited tool in use for analyzing CD spectra¹. Specifically, we employed the unified ridge regression-based CONTINLL algorithm to quantitatively estimate the secondary structure of our molecules. The method incorporates variable selection techniques that filter the datasets so that only the closest matching spectra to the test spectrum are used in the final analysis. The explanation for the secondary structure calculation has been added to the Methods section in the resubmitted manuscript.

Reference

1. Miles, A. J., Ramalli, S. G. & Wallace, B. A. DichroWeb, a website for calculating protein secondary structure from circular dichroism spectroscopic data. *Protein Sci.* **31**, 37–46 (2022).

Changes in the manuscript on page 33 and 34 (Methods section)

CD Spectroscopy

All the CD spectra analyses were recorded on a CD spectropolarimeter (Applied Photophysics Ltd, UK) in quartz cuvettes with an optical path length of 0.1 cm. Data were collected from 180 to 260 nm with a scanning speed of 100 nm / min at room temperature.

The DichroWeb server (<http://dichroweb.cryst.bbk.ac.uk>) and the unified ridge regression-based CONTINLL algorithm were utilized for a quantitative analysis of our peptide molecules' secondary structure from CD spectra.

Comment 3: In Fig. 1I, the authors claimed that “nearly all precursors transformed into nanofibers with a diameter of 13.2 ± 2.9 nm”. However, PAC-SABIs treated with ALP for 120 min exhibited reticular polymer structure, and the nanofibers were less conspicuous than 60 min. Did the nanofibers from PAC-SABIs aggregate with each other and form a larger polymer framework?

Response to comment 3: Thank you for your incisive comments. Reflecting upon both your and Reviewer #3's insights, we acknowledge that our initial description of “At 120 min, nearly all precursors transformed into nanofibers with a diameter of 13.2 ± 2.9 nm” in the original manuscript is not entirely accurate. The transformation of PAC-SABIs from irregular aggregates to fiber networks following alkaline phosphatase (ALP) treatment was indeed significant. However, when considering the molecular diversity inherent to PAC-SABIs and the evidence provided by scanning electron microscopy (SEM) images in various environments, it became evident that PAC-SABIs manifested as a larger network scaffold with interlaced nanofibers presenting a diameter

measuring 13.2 ± 2.9 nm under incubation with ALP for 120 min. In the resubmitted manuscript, the modifications have been made to the morphology description of PAC-SABIs after ALP treatment for 120 min through transmission electron microscopy (TEM) imaging.

Changes in the manuscript on page 8 and 9 (Results section)

Subsequently, we conducted an extensive assessment of the micromorphology of Pep-PEG and their intermediates utilizing transmission electron microscopy (TEM). The TEM imaging results, as illustrated in Supplementary Fig. 13, revealed that the modular design of Pep-PEG manifested distinct variations in self-assembly dynamics and microstructure corresponding to changes in module components. Notably, both Pep and PAC-SABIs displayed time-dependent morphological transformations under ALP co-incubation (Fig. 2i). Pep exhibited a quasi-circular aggregate, wherein these peptide assemblies subsequently aggregated and manifested a spherical structure after a 10-min incubation period with ALP. Then, the advent of a short fiber structure was observed at 30 min. By the 60-min mark, the majority of the spherical structure was dissipated, giving way to a substantial presence of fibrous structures. At the 120-min interval, the entire assembly system was predominantly occupied by characteristic nanofibers with a diameter of 6.2 ± 0.7 nm (Fig. 2i). However, PAC-SABIs were observed as irregular aggregates. During the initial 30 min of enzyme catalyzed reactions, these aggregates rapidly coalesced into entangled nanofibers. Subsequently, these nanofibers underwent further extension and bundling, resulting in the formation of a three-dimensional (3D) network after 60 min. At 120 min, nearly all PAC-SABIs transformed into a larger network scaffold with interlaced nanofibers presenting a diameter measuring 13.2 ± 2.9 nm (Fig. 2i). TEM analysis showed that the final morphology of PAC-SABIs was closely related to the proportion of peptide molecules in the system. At a mAb: peptide ratio of 1: 1, the TEM image depicted large, irregular aggregates. Progressing to a ratio of 1: 5, these aggregates appeared denser and more branched, while at a ratio of 1: 10, they evolved into a fibrous network structure, distinguished by the high degree of cross-

linking and framework complexity (Supplementary Fig. 14). ThT assays further confirmed the critical role of primary peptide components in forming β -sheet structures within the desired PAC-SABIs system (Supplementary Fig. 15). The notable increase in β -sheet content, characterized by parallel or antiparallel amino acid chains stabilized by hydrogen bonds, facilitated stacking conducive to fiber morphology assembly. Moreover, SEM imaging offered a more comprehensive view and surface feature information of PAC-SABIs, implying their potential for 3D network scaffold formation via in situ self-assembly (Supplementary Fig. 16).

Comment 4: The pictures in this manuscript were blurry, the authors need to offer higher-definition version.

Response to comment 4: Thank you for your helpful comments. The adjustments to the figures have been made to enhance their resolution and clarity in the resubmitted manuscript. Additionally, these figures have been uploaded as separate high-quality image files to the system. Should any issues with blurriness arise in the manuscript or response letter due to file size constraints, we kindly request that you refer to the image files for optimal viewing.

Comment 5: In Fig. 5B and 5C, the authors claimed that the TCM-stimulated HDDMs exhibited upregulation of the M2 marker CD206. Therefore, did the treatment with PAC-SABI of MDA-MB-231 cells result in repolarization of macrophage due to the blockage of CD24 and CD47? More evidence should be offered.

Response to comment 5: Thank you for your insightful comments. Given that inhibiting CD47 and CD24 signaling pathways via PAC-SABIs promoted a shift from M2 to M1 phenotype in tumor-associated macrophages (TAMs) within the 4T1 breast cancer (BC) model, we deem it imperative to investigate the interactions between MDA-MB-231 cells treated with PAC-SABIs and human donor-derived macrophages (HDDMs), as it may influence and potentially recalibrate the polarization state of these macrophages.

Initially, we obtained HDDMs utilizing the methodology as delineated in the

original manuscript and subsequently induced TAM differentiation under same conditions. Subsequently, 100,000 TAMs were seeded onto a transparent plate and maintained at 37°C to facilitate adherence. After confirming macrophage adherence, 200,000 MDA-MB-231 cells, pre-treated with PAC-SABIs for 120 min, were introduced into a serum-free IMDM medium (Gibco, Life Technologies). Following a 120-min co-incubation period, the samples underwent rigorous washing procedures, involving three sequential washes with PBS, to ensure the removal of any non-phagocytosed cancer cells. The expression of the M2 macrophage marker CD206 was evaluated using confocal laser scanning microscopy (CLSM) imaging and flow cytometry. Our findings revealed that, in comparison to the unstimulated group, the expression of CD206 in HDDMs treated with tumor conditioned media (TCM) exhibited a notable increase, regardless of their interaction with PAC-SABIs pre-treated MDA-MB-231 cells (Supplementary Fig. 27). As shown in Supplementary Fig. 27c, no significant difference in CD206 expression was noted between G2 and G3. These results suggest that the observed heightened phagocytic activity in macrophages is primarily due to the inhibitory effects of PAC-SABIs on the innate immune checkpoints CD47 and CD24, rather than a shift in macrophage polarization from M2 to M1 phenotype.

Changes in the manuscript on page 18 (Results section)

In agreement with the results in murine macrophages, human THP-1 monocyte-derived macrophages stimulated by TCM exhibited enhanced clearance of human breast cancer MDA-MB-231 cells treated with PAC-SABIs compared to other groups (Supplementary Fig. 26). Primary human donor-derived macrophages (HDDMs) were also generated by the methodology previously outlined by Barkal et al.²⁰ to assess PAC-SABIs efficacy (Fig. 6a). Subsequently, the HDDMs were conditioned with TCM from MDA-MB-231 to generate TAMs. IF and flow cytometry evidenced upregulation of the M2 marker CD206 in TAMs versus control HDDMs, regardless of their interaction with PAC-SABIs pre-treated MDA-MB-231 cells (Fig. 6b, c, Supplementary Fig. 27). As shown in Supplementary Fig. 27c, no significant difference in CD206 expression was

noted between G2 and G3. These results provided further proof that the observed heightened phagocytic activity in macrophages from 2D and 3D CLSM imaging was primarily due to the inhibitory effects of PAC-SABIs on the innate immune checkpoints CD47 and CD24, rather than a shift in macrophage polarization from M2 to M1 phenotype (Fig. 6d-f).

Changes in the Supporting Information on page 32

Fig. 27 Transformation in TCM-stimulated HDDMs polarization. **a** IF staining of CD11b and CD206 in unstimulated HDDMs, TCM-stimulated HDDMs (No interaction with MDA-MB-231 cells), and TCM-stimulated HDDMs (Interaction with MDA-MB-231 cells). Scale bar: 50 μ m. **b** Flow cytometry-based measurement of CD206 expression by unstimulated HDDMs (grey), TCM-stimulated HDDMs (No interaction with MDA-MB-231 cells, red), and TCM-stimulated HDDMs (Interaction with MDA-MB-231 cells, blue). **c** Quantification analysis of CD206^{hi} macrophages after different treatments. G1 refers to unstimulated HDDMs; G2 refers to TCM-stimulated HDDMs

(No interaction with MDA-MB-231 cells); G3 refers to TCM-stimulated HDDMs (Interaction with MDA-MB-231 cells). The error bars represent the mean \pm SD (n = 3 independent experiments; n.s. refers to no significance; the *p* value was analyzed by a two-tailed unpaired Student's *t*-test).

Comment 6: In Fig. 7, the authors should offer the physical images of tumor tissue and the data on body weight change in mice et al.

Response to comment 6: We appreciate your valuable comments. In the original manuscript, Fig. 8 was primarily focused on recording and analyzing the survival durations of mice across different treatment groups, which did not include a uniform collection of tumor samples at a consistent time point. Following your suggestion, we have conducted an additional experiment to more clearly demonstrate the efficacy of PAC-SABIs in treating breast cancer (BC) and pancreatic cancer (PC).

We first established subcutaneous xenograft models of BC and PC using 4T1 and PAN02 cells, respectively. Mice were randomly distributed into four groups: IgG Control, Anti-CD24, SAMIs, and PAC-SABIs. Each group underwent intravenous administration of their respective treatments every other day, for a total of four sessions. On the 21st day, all the mice were humanely euthanized, and the tumors (both 4T1 and PAN02) were surgically excised and weighed (Supplementary Fig. 35a, e). Our observations indicated no significant differences in the body weights of the mice across the treatment groups (Supplementary Fig. 35c, g). Importantly, the physical tumor assessments and wet weight measurements showed that PAC-SABIs significantly inhibited tumor growth in both BC and PC models, surpassing the efficacy of either anti-CD24 antibodies or SAMIs when used independently (Supplementary Fig. 35). These results further elucidate the potent antitumor capabilities of PAC-SABIs.

Changes in the manuscript on page 23 and 24 (Results section)

Encouraged by the aforementioned experimental results, we hypothesized that PAC-SABIs therapy has the potential to instruct macrophages in eliciting proficient *in vivo* antitumor phagocytic reactions, consequently leading to the inhibition of tumor growth. To evidence this, subcutaneous xenograft models of BC and PC were

established using 4T1 and PAN02 cells, respectively. Mice were randomly distributed into four groups: IgG Control, Anti-CD24, SAMIs, and PAC-SABIs, and received their respective treatments intravenously every other day over four sessions. We continuously monitored the mouse weight during the treatment. On the 21st day, all mice were humanely euthanized, and the 4T1 and PAN02 tumors were surgically removed and weighed (Supplementary Fig. 35a, e). No significant weight differences were observed among the groups (Supplementary Fig. 35c, g). Notably, PAC-SABIs demonstrated a significant reduction in tumor growth in both BC and PC models, surpassing the efficacy of anti-CD24 antibodies or SAMIs (Supplementary Fig. 35). Furthermore, in the BC xenograft model, PAC-SABIs exhibited a significantly stronger tumor suppression effects than dual antibody combination therapy (Supplementary Fig. 36). Specifically, the average tumor weights were 1.07 g for the antibody combination group and 0.33 g for the PAC-SABIs group, reflecting a unique PAC-SABIs functionality of concurrently inhibiting CD47 and CD24 signaling, coupled with their intrinsic self-assembly ability within the complex TME.

Changes in the manuscript on page 39 and 40 (Methods section)

Antitumor Treatment Studies in subcutaneous xenograft tumor models

To establish BC and PC subcutaneous xenografts, 1×10^6 4T1 and PAN02 cells were implanted into the right hind limb of each BALB/c and C57BL/6 mouse, respectively. Upon the tumor reaching an approximate volume of 50 mm^3 by the 7th day post-implantation, the mice were randomized into groups. For evaluation of PAC-SABIs antitumor efficacy, the PAC-SABIs (15 mg / kg) treatment began 7 days after the tumor implantation. Each group received its respective intravenous treatment every other day for a total of four sessions. On the 28th day post-implantation, the mice were euthanized and subsequently dissected for tumor weight analysis.

To compare the antitumor efficacy of PAC-SABIs with the dual antibody combination therapy, three groups of mice received tail vein injections, with the respective treatments being IgG control, a combination of Anti-CD47 (20 mg / kg, miap301, BioXcell) and Anti-CD24 antibodies, and PAC-SABIs. On the 28th day post-

implantation, the mice were euthanized and subsequently dissected for tumor weight analysis.

For evaluation of sequential PAC-SABIs and anti-PD-1 combination treatment efficacy, the mice were administered intravenously with PAC-SABIs or PBS every 2 days for a total of 4 doses, followed by anti-PD-1 (10 mg / kg) or IgG control intraperitoneal injection on days 7, 10, 13, and 16. On the 28th day post-implantation, the experiment concluded with the euthanasia of the mice. The 4T1 tumors were surgically removed and the wet weight was recorded.

Changes in the Supporting Information on page 41

Fig. 35 Antitumor efficacy of PAC-SABIs in vivo. **a** Scheme of the PAC-SABIs therapeutic strategy for 4T1 subcutaneous tumor (Created with BioRender.com). **b** Bright field images of excised 4T1 tumors. **c** Changes in 4T1 tumor-bearing mouse body weight during treatment. The error bars represent the mean \pm SD (n = 5). **d** 4T1 tumor weights of each group after treatment. The

error bars represent the mean \pm SD (n = 5; ** $p < 0.01$, *** $p < 0.001$; the p value was analyzed by a two-tailed unpaired Student's t-test). **e** Scheme of the PAC-SABIs therapeutic strategy for PAN02 subcutaneous tumor (Created with BioRender.com). **f** Bright field images of excised PAN02 tumors. **g** Changes in PAN02 tumor-bearing mouse body weight during treatment. The error bars represent the mean \pm SD (n = 5). **h** PAN02 tumor weights of each group after treatment. The error bars represent the mean \pm SD (n = 5; ** $p < 0.01$, *** $p < 0.001$; the p value was analyzed by a two-tailed unpaired Student's t-test).

Comment 7: In Fig. 9, the enhanced anti-PD-1 therapy exhibited strong repression of tumor growth. But, why PAC-SABIs group shows more slight tumor growth inhibition? Therefore, the physical images of tumor tissue should be supplemented. And did the combination therapy possess a higher repolarization ratio of macrophage? How about other tumor-infiltrating lymphocytes?

Response to comment 7: Thank you for your constructive comments. Prior to conducting sequential PAC-SABIs and anti-PD-1 combination treatment, we undertook a meticulous analysis of PAC-SABIs impact on the tumor immune microenvironment (TIME). Our investigations unveiled that PAC-SABIs significantly enhanced macrophage phagocytosis in the 4T1 model. This was coupled with an effective polarization of macrophages towards the M1 phenotype and the release of pro-inflammatory cytokines. Additionally, macrophage modulation exerted a profound influence on adaptive antitumor immunity, as evidenced by increased quantity and depth of T lymphocyte infiltration within TIME, particularly CD8⁺ T cells. The surge in T cell abundance thus illuminates the rationale for the amplified in vivo antitumor immune response following the combination treatment. These results align with our anticipated therapeutic efficacy, affirming that the combination regimen confers a more potent inhibitory effect on tumor growth when compared to the solitary administration of PAC-SABIs.

In the original manuscript, we analyzed the survival duration of 4T1 tumor-bearing mice across various treatment groups. To further validate the oncological efficacy of the combination therapy, we conducted a new experiment using a subcutaneous BC

xenograft model with 4T1 cells. The mice were administered intravenously with PAC-SABIs or PBS every 2 days for a total of 4 doses, followed by anti-PD-1 or IgG control intraperitoneal injection on days 7, 10, 13, and 16. On the 21st day, the experiment concluded with the euthanasia of the mice. The 4T1 tumors were surgically removed and the wet weight was recorded (Supplementary Fig. 39a). Moreover, in response to your recommendations, flow cytometry and immunofluorescence (IF) were employed to investigate whether the combination therapy resulted in a heightened macrophage repolarization towards the M1 phenotype as well as enhanced lymphocyte infiltration within the TIME.

Throughout the treatment, we continuously monitored mouse weights and observed no significant differences among the groups (Supplementary Fig. 39b). Notably, the sequential PAC-SABIs and anti-PD-1 combination treatment exhibited the most substantial tumor growth inhibition, approximately 94.5% (Supplementary Fig. 39). Further analysis using flow cytometry showed that macrophages in the combination treatment group had reduced CD206 expression, similar to the PAC-SABIs group, and a significant increase in CD86 expression compared to other groups (Supplementary Fig. 40). IF analysis of 4T1 tumor sections revealed an increase in NK1.1⁺ cells and a decrease in FOXP3⁺ T cell infiltration following the combination treatment (Supplementary Fig. 41). These findings highlight the potential of PAC-SABIs to effectively activate macrophage phagocytosis, leading to an enhanced adaptive antitumor immune response and offering promising prospects for combined immunotherapy strategies.

Changes in the manuscript on page 29-31 (Results section)

Based on the aforementioned *in vivo* experimental findings, regulating the phagocytosis of cancer cells by macrophages exhibited great potential in adaptive **antitumor** immune induction. Therefore, it is postulated that the combination of PAC-SABIs and PD-1 pathway blockade could potentially result in an augmented **antitumor** immune response *in vivo*, yielding optimal therapeutic outcomes. **To investigate this, we constructed a subcutaneous BC xenograft model utilizing 4T1 cells. The mice were**

administered intravenously with PAC-SABIs or PBS every 2 days for a total of 4 doses, followed by anti-PD-1 or IgG control intraperitoneal injection on days 7, 10, 13, and 16. On the 21st day, the experiment concluded with the euthanasia of the mice, and the 4T1 tumors were surgically removed and the wet weight was recorded (Supplementary Fig. 39a). Throughout the treatment, we continuously monitored mouse weights and observed no significant differences among the groups (Supplementary Fig. 39b). Notably, the sequential PAC-SABIs and anti-PD-1 combination treatment exhibited the most substantial tumor growth inhibition, approximately 94.5% (Supplementary Fig. 39c, d). Then, we inoculated GFP⁺ 4T1 cells into the third mammary fat pad of female BALB/c mice to construct the orthotopic BC model. The 4T1 tumor growth and survival rate were measured over time to assess the immunotherapeutic efficacy of different treatment regimens (Fig. 10a). The untreated and anti-PD-1 groups showed rapid tumor growth and poor median survival. Compared with these two groups, the PAC-SABIs group showed enhanced tumor growth inhibition and improved survival (Fig. 10b-d). Remarkably, anti-PD-1 therapy after PAC-SABIs activation of macrophages produced the greatest tumor suppression effect and achieved a 60-day optimal survival rate of 57% (Fig. 10d).

At the experimental end point, the lungs were dissected for analysis. Histological examination of representative lung sections using H&E staining showed a lower incidence of pulmonary metastatic nodules in the SAMIs group compared to the untreated and anti-PD-1 groups (Fig. 10e). More importantly, the combination of PAC-SABIs and anti-PD-1 therapy effectively impeded lung metastasis development, with few observable nodules. Given the promising results, flow cytometry was used to investigate macrophage phagocytic activity towards 4T1 cells. The combination treatment led to a substantial increase in phagocytosing CD45⁺CD11b⁺F4/80⁺ macrophages compared to other groups (Fig. 10f). Further analysis using flow cytometry showed that macrophages in the combination treatment group had reduced CD206 expression, similar to the PAC-SABIs group, and a significant increase in CD86 expression compared to other groups (Supplementary Fig. 40). IF analysis of 4T1 tumor sections revealed an increase in CD8⁺ T cells and NK1.1⁺ cells, coupled with a decrease

in FOXP3⁺ T cell infiltration following the combination treatment (Fig. 10g, Supplementary Fig. 41). These findings highlighted the potential of PAC-SABIs to effectively activate macrophage phagocytosis, leading to an enhanced adaptive antitumor immune response and offering promising prospects for combined immunotherapy strategies.

Changes in the manuscript on page 39 and 40 (Methods section)

Antitumor Treatment Studies in subcutaneous xenograft tumor models

To establish BC and PC subcutaneous xenografts, 1×10^6 4T1 and PAN02 cells were implanted into the right hind limb of each BALB/c and C57BL/6 mouse, respectively. Upon the tumor reaching an approximate volume of 50 mm³ by the 7th day post-implantation, the mice were randomized into groups. For evaluation of PAC-SABIs antitumor efficacy, the PAC-SABIs (15 mg / kg) treatment began 7 days after the tumor implantation. Each group received its respective intravenous treatment every other day for a total of four sessions. On the 28th day post-implantation, the mice were euthanized and subsequently dissected for tumor weight analysis.

To compare the antitumor efficacy of PAC-SABIs with the dual antibody combination therapy, three groups of mice received tail vein injections, with the respective treatments being IgG control, a combination of Anti-CD47 (20 mg / kg, miap301, BioXcell) and Anti-CD24 antibodies, and PAC-SABIs. On the 28th day post-implantation, the mice were euthanized and subsequently dissected for tumor weight analysis.

For evaluation of sequential PAC-SABIs and anti-PD-1 combination treatment efficacy, the mice were administered intravenously with PAC-SABIs or PBS every 2 days for a total of 4 doses, followed by anti-PD-1 (10 mg / kg) or IgG control intraperitoneal injection on days 7, 10, 13, and 16. On the 28th day post-implantation, the experiment concluded with the euthanasia of the mice. The 4T1 tumors were surgically removed and the wet weight was recorded.

Changes in the manuscript on page 41 (Methods section)

Analysis of Immune Response in Tumor

The 4T1 tumor-bearing mice were euthanized, and tumor tissues were collected and frozen in optimal cutting temperature medium on dry ice. Tumor sections were cut using a cryotome, mounted on slides and stained with different primary antibodies: Anti-F4/80 antibody (Abcam, cat. no. ab6640), Anti-CK19 antibody (Abcam, cat. no. ab52625), Anti-CD8 antibody (Invitrogen, cat. no. MA5-29682), Anti-NK1.1 antibody (Abcam, cat. no. ab289542), and Anti-Foxp3 antibody (Affinity Biosciences, cat. no. AF6544) overnight at 4 °C according to the manufacturer's instructions. Following the addition of fluorescently labelled goat anti-rat IgG H&L (Abcam, cat. no. ab150167) and goat anti-rabbit IgG H&L (Abcam, cat. no. ab150077), the slides were analyzed with a confocal microscope (Nikon, Japan).

Changes in the Supporting Information on page 45

Fig. 39 Enhanced anti-PD-1 therapy after phagocytosis modulation. a Scheme of the combination therapeutic strategy (Created with BioRender.com). **b** Changes in 4T1 tumor-bearing mouse body weight during treatment. The error bars represent the mean \pm SD (n = 7). **c** Bright field images of excised 4T1 tumors. **d** 4T1 tumor weights of each group after treatment. The error bars represent the mean \pm SD (n = 7; *** p < 0.001; the p value was analyzed by a two-tailed unpaired Student's t-test).

Changes in the Supporting Information on page 46

Fig. 40 Macrophage polarization in vivo. **a** Representative flow cytometry plots and quantification analysis of CD11b⁺F4/80⁺CD206^{hi} macrophages in 4T1 tumors after different treatments. The error bars represent the mean \pm SD ($n = 4$ independent experiments; n.s. refers to no significance, $^{***}p < 0.001$; the p value was analyzed by a two-tailed unpaired Student's t-test). **b** Representative flow cytometry plots and quantification analysis of CD11b⁺F4/80⁺CD86^{hi} macrophages in 4T1 tumors after different treatments. The error bars represent the mean \pm SD ($n = 4$ independent experiments; $^*p < 0.05$, $^{***}p < 0.001$; the p value was analyzed by a two-tailed unpaired Student's t-test).

Changes in the Supporting Information on page 47

Fig. 41 Representative IF staining of NK1.1⁺ cells and Foxp3⁺ T cells for the corresponding 4T1 tumor tissues after different treatments. Scale bar: 200 μ m.

Comment 8: In this manuscript, the authors designed an ALP-responsive PAC-SABI, and the introduction of the PEG2000 chain obviously prolonged circulation in blood. How did the PAC-SABI be stable in the blood because of the presence of ALP in human serum? And what about the threshold of ALP of PAC-SABI?

Response to comment 8: Thank you for your insightful comments. Given the obviously prolonged PAC-SABIs circulation in blood, it is crucial to understand the conditions that influence the interaction between PAC-SABIs and ALP.

To begin our investigation into the dephosphorylation and in situ self-assembly processes induced by ALP, we introduced PAC-SABIs into a PBS buffer (pH = 7.4) containing various ALP concentrations. The mixture was incubated at 37 °C for 60 min. The remaining PAC-SABIs were then quantified using reverse phase high-performance liquid chromatography (RP-HPLC). Integrated peak area calculations showed that the residual PAC-SABIs decreased consistently with increasing ALP concentrations, with percentages ranging from 96.7 % to 6.5 % across ALP concentrations of 0.01 to 8.00 U / mL (Supplementary Fig. 30). Notably, our two-piecewise linear regression model analysis indicated an optimal ALP threshold of 1.06 U / mL (Supplementary Fig. 31).

Recognizing the influence of factors like cancer type, liver function, bone metabolism, and treatment on serum ALP levels in cancer patients, we conducted a retrospective study involving a consecutive cohort of 22 breast cancer (BC) patients diagnosed by pathological examination at Department of Breast Cancer from Guangdong Provincial People's Hospital between November and December 2023 (Supplementary Table 2). Most of these patients (95.5 %) had preoperative ALP levels within the normal range (50 - 139 U / L), with a median value (64 U / L) significantly below our in vitro ALP reactivity threshold (Supplementary Table 2, Fig. 32). This suggests that PAC-SABIs could remain stable in the circulatory system of most clinical BC patients. To corroborate this, we analyzed serum samples from four BC patients, co-incubated with PAC-SABIs. Our findings showed a gradual degradation of PAC-

SABIs, maintaining over 60% stability after 48 h (Supplementary Fig. 33). These results not only confirm the long-term stability of PAC-SABIs in a biological environment but also underscore their potential for clinical application.

Changes in the manuscript on page 21 and 22 (Results section)

Given the obviously prolonged PAC-SABIs circulation in blood, it is crucial to understand the conditions that influence the interaction between PAC-SABIs and ALP. We first introduced PAC-SABIs into a PBS buffer (pH = 7.4) containing various ALP concentrations. The mixture was incubated at 37 °C for 60 min. Reverse phase HPLC analysis showed that the residual PAC-SABIs decreased consistently with increasing ALP concentrations, with percentages ranging from 96.7 % to 6.5 % across ALP concentrations of 0.01 to 8.00 U / mL (Supplementary Fig. 30). Notably, our two-piecewise linear regression model analysis indicated an optimal ALP threshold of 1.06 U / mL (Supplementary Fig. 31). Subsequently, we conducted a retrospective study on 22 BC patients from Guangdong Provincial People's Hospital between November and December 2023 (Supplementary Table 2). Most of these patients (95.5 %) had preoperative ALP levels within the normal range (50 - 139 U / L), with a median value (64 U / L) significantly below our in vitro ALP reactivity threshold (Supplementary Table 2, Fig. 32). This suggested that PAC-SABIs could remain stable in the circulatory system of most clinical BC patients. To corroborate this, we analyzed serum samples from four BC patients, co-incubated with PAC-SABIs. Our findings showed a gradual degradation of PAC-SABIs, maintaining over 60% stability after 48 h (Supplementary Fig. 33). These results not only confirmed the long-term stability of PAC-SABIs in a biological environment but also underscored their potential for clinical application.

Changes in the Supporting Information on page 35

Fig. 30 Stability of PAC-SABIs under different concentrations of ALP (U / mL) incubation conditions.

Changes in the Supporting Information on page 36

Fig. 31 Two-piecewise linear regression model analysis for threshold value of ALP (U / mL).

Changes in the Supporting Information on page 37

Table 2 Baseline clinical characteristics of the BC patients.

Characteristic	
Age, median (min, max), y	57 (31, 72)
Gender, n (%)	
Male	0 (0.0)
Female	22 (100.0)
Tumor type, n (%)	
Infiltrating ductal	18 (81.8)
Infiltrating lobular	3 (13.7)
Other	1 (4.5)
Tumor localization, n (%)	
Unilateral	21 (95.5)
Bilateral	1 (4.5)
ALP, median (min, max), U / L	64 (40, 146)

y, year; n, number; ALP, alkaline phosphatase; Normal range of ALP: 50 - 139 U / L.

Changes in the Supporting Information on page 38

Fig. 32 Serum ALP levels in clinical BC patients (U / L). The upper and lower red dashed lines refer to the upper and lower limits of the normal ALP range (139 and 50 U / L, respectively).

Changes in the Supporting Information on page 39

Fig. 33 Stability of PAC-SABIs in serum of BC patients over time. The error bars represent the mean \pm SD (n = 4 independent experiments).

Comment 9: The manuscript still contains spelling and formatting errors, the authors should check it carefully.

Response to comment 9: Thank you for your meticulous review. We have carefully examined the spelling and formatting errors in the original manuscript, and all the modifications have been made and highlighted in yellow within the resubmitted manuscript.

In answer to the comments raised by Reviewer #2

Comment 1: Information regarding anti-mouse CD24 (for 4T1 and PAN02) and anti-human CD24 (for MDA-MB-231) used in the study is missing. Whether these antibodies effectively block CD24-Siglec10 interaction/signaling should be verified.

Response to comment 1: Thank you for your valuable comments. Aligning with your suggestion, we acknowledge the imperative for a rigorous validation of the efficacy of anti-CD24 antibodies in obstructing the CD24-Siglecs interaction. Following the methodology reported by Barkal et al.¹, we aimed to supplement the missing results in the original manuscript.

In our experiment, 100,000 non-fluorescently labeled 4T1 or PAN02 cells were treated with the anti-CD24 antibody (clone M1/69, BioXcell) for 1 h at 37 °C in serum-free medium. In parallel, MDA-MB-231 cells underwent identical treatment using the anti-CD24 antibody (clone SN3, Novus Biologics). All antibodies were added at a concentration of 10 µg / mL. Recombinant Siglec-G (ortholog of human Siglec-10) and Siglec-10 proteins, acquired in the forms of mouse and human Fc fusion proteins respectively, were sourced from R&D Systems Inc. The Binding of recombinant Siglecs versus isotype IgG1 control was assayed at a concentration of 100,000 cells / 1 mg / mL at 37 °C for 1 h. Subsequently, 4T1, PAN02, or MDA-MB-231 cells were stained with fluorescently-conjugated anti-mouse/human Fc antibodies (Biolegend) for recombinant Siglec binding analysis via flow cytometry. The results demonstrated that anti-CD24 antibody treatment significantly reduced Siglec-10 binding to MDA-MB-231 cells and Siglec-G binding to 4T1 and PAN02 cells (Supplementary Fig. 19). The consistent pattern across different cell types indicates that CD24 antibodies disrupt CD24-Siglecs interactions, highlighting their role in modulating key interactions between cancer cells and macrophages.

Reference

1. Barkal, A. A. *et al.* CD24 signalling through macrophage Siglec-10 is a target for cancer immunotherapy. *Nature* **572**, 392–396 (2019).

Changes in the manuscript on page 12 and 13 (Results section)

Having determined the effective assembly of PAC-SABIs under static external conditions, we proceeded to investigate the self-assembly behavior of PAC-SABIs within BC and PC cellular environments using the NBD fluorescence. Firstly, the widespread expression of ALP in 4T1 and PAN02 subcutaneous tumor, as well as clinical BC and PC samples, was verified using the BCIP/NBT color development kit (Supplementary Fig. 17). Then, 4T1 and PAN02 cells were co-incubated with NBD-labeled PAC-SABIs at 37 °C, and changes in NBD fluorescence were directly monitored using confocal laser scanning microscopy (CLSM) imaging at different time points (Fig. 4a, b). As anticipated, the treated 4T1 and PAN02 cells exhibited a time-dependent augmentation in NBD fluorescence emission on the cellular surface. Following a 30-min co-incubation period, the fluorescence appeared as a diffuse signal surrounding the cancer cells as PAC-SABIs bound to the membrane's outer surface. Then, a prompt aggregation of fluorescent clusters on the membrane was observed at the 60-min mark. By the 120-min interval, fluorescence became uniform and encompassed the cell surface, signifying the in situ self-assembly of PAC-SABIs on cancer cell membrane (Fig. 4a, b). This observation was further supported by the findings from differential interference contrast (DIC) and CLSM imaging (Fig. 4c). As shown in Fig. 4d-f, the green fluorescence from NBD exhibited a strong co-localization with the red fluorescence from Dil dye, suggesting that a majority of PAC-SABI molecules were assembled and localized on the 4T1 cell membrane. Under identical conditions, 4T1 cells treated with SAMIs exhibited robust intracellular green fluorescence, reflecting the internalization of SAMIs into the cells, a phenomenon commonly observed when other nanostructures are incubated with live cells (Supplementary Fig. 18). After confirming that anti-CD24 mAbs effectively disrupted the CD24-Siglecs interaction (Supplementary Fig. 19), we conducted a blocking experiment utilizing these anti-CD24 mAbs. The surface fluorescence signals of pretreated 4T1 cells were significantly reduced during dynamic incubation with PAC-SABIs, indicating that the specificity of interaction between CD24 and PAC-SABIs promoted the membrane in situ self-assembly process (Supplementary Fig. 20). Additionally, as shown in Supplementary Fig. 21, TEM revealed the morphologies of

PAC-SABIs assembled on the top surface of cell membranes. We also observed the formation of superstructure networks over time on the 4T1 and PAN02 cell membranes treated with PAC-SABIs through SEM imaging. During the co-incubation process from 30 to 120 min, PAC-SABIs, like the proliferation process of lichens, first attached to the cell's adherent edge, then migrated along the cell surface, and finally formed a 3D network scaffold covering the membrane (Fig. 4g, h).

Changes in the manuscript on page 37 (Methods section)

Anti-CD24 treatment and recombinant Siglec binding assay

100,000 non-fluorescently labeled 4T1 or PAN02 cells were treated with the anti-CD24 antibody (clone M1/69, BioXcell) for 1 h at 37 °C in serum-free medium. In parallel, MDA-MB-231 cells underwent identical treatment using the anti-CD24 antibody (clone SN3, Novus Biologics). All antibodies were added at a concentration of 10 µg / mL. Recombinant Siglec-G (ortholog of human Siglec-10) and Siglec-10 proteins, acquired in the forms of mouse and human Fc fusion proteins respectively, were sourced from R&D Systems Inc. The Binding of recombinant Siglecs versus isotype IgG1 control was assayed at a concentration of 100,000 cells / 1 mg / mL at 37 °C for 1 h. Subsequently, 4T1, PAN02, or MDA-MB-231 cells were stained with fluorescently-conjugated anti-mouse/human Fc antibodies (Biolegend) for recombinant Siglec binding analysis via flow cytometry.

Changes in the Supporting Information on page 24

Fig. 19 Flow cytometry histogram measuring binding of Siglec-10 to cancer cells. a Flow

cytometry histogram measuring binding of Siglec-10 to MDA-MB-231 cells after different treatments. **b** Flow cytometry histogram measuring binding of Siglec-G to 4T1 cells after different treatments. **c** Flow cytometry histogram measuring binding of Siglec-G to PAN02 cells after different treatments.

Comment 2: CD47 and CD24 blockade with blocking antibodies has been reported in a number of previous studies in inhibiting tumor growth. It's unclear whether PAC-SABIs would exhibit superior antitumor effects compared to a combination of CD47 and CD24 blocking antibodies. An in vivo experiment should be performed to directly compare the antitumor effects of PAC-SABIs with that of a combination of CD47 and CD24 blocking antibodies.

Response to comment 2: We appreciate your insightful comments. In the original manuscript, PAC-SABIs exhibited the distinct capability of simultaneously inhibiting CD47 and CD24 signaling. This is complemented by their intrinsic ability to self-assemble in situ within the complex tumor microenvironment (TME). Given these characteristics, we conducted an in vivo experiment to compare the antitumor efficacy of PAC-SABIs with the established dual antibody combination therapy.

For establishing subcutaneous breast cancer (BC) xenografts, we inoculated 1×10^6 4T1 cells into the right hind limb of each BALB/c mouse. Upon the tumor reaching an approximate volume of 50 mm^3 by the 7th day post-implantation, the mice were randomized into three groups, each comprising five individuals. In alignment with methodologies documented in prior research¹, we administered a 20 mg / kg dose of anti-mouse CD47 antibody (miap301, BioXcell). The groups received tail vein injections, with the respective treatments being IgG control, a combination of Anti-CD47 and Anti-CD24 antibodies, and PAC-SABIs, on a bi-daily schedule, totaling four treatments. On the 21st day, the mice were euthanized and subsequently dissected for tumor weight analysis (Supplementary Fig. 36a). The results indicated that PAC-SABIs had a significantly stronger tumor-suppressive effect compared to the group treated with the antibody combination. Specifically, the average tumor weights were 1.07 g for the antibody combination group and 0.33 g for the PAC-SABIs group (Supplementary

Fig. 36), demonstrating a substantial antitumor benefit of PAC-SABIs.

Reference

1. Wang, H. *et al.* CD47/SIRP α blocking peptide identification and synergistic effect with irradiation for cancer immunotherapy. *J. Immunother. Cancer* **8**, e000905 (2020).

Changes in the manuscript on page 23 and 24 (Results section)

Encouraged by the aforementioned experimental results, we hypothesized that PAC-SABIs therapy has the potential to instruct macrophages in eliciting proficient *in vivo* antitumor phagocytic reactions, consequently leading to the inhibition of tumor growth. To evidence this, subcutaneous xenograft models of BC and PC were established using 4T1 and PAN02 cells, respectively. Mice were randomly distributed into four groups: IgG Control, Anti-CD24, SAMIs, and PAC-SABIs, and received their respective treatments intravenously every other day over four sessions. We continuously monitored the mouse weight during the treatment. On the 21st day, all mice were humanely euthanized, and the 4T1 and PAN02 tumors were surgically removed and weighed (Supplementary Fig. 35a, e). No significant weight differences were observed among the groups (Supplementary Fig. 35c, g). Notably, PAC-SABIs demonstrated a significant reduction in tumor growth in both BC and PC models, surpassing the efficacy of anti-CD24 antibodies or SAMIs (Supplementary Fig. 35). Furthermore, in the BC xenograft model, PAC-SABIs exhibited a significantly stronger tumor suppression effects than dual antibody combination therapy (Supplementary Fig. 36). Specifically, the average tumor weights were 1.07 g for the antibody combination group and 0.33 g for the PAC-SABIs group, reflecting a unique PAC-SABIs functionality of concurrently inhibiting CD47 and CD24 signaling, coupled with their intrinsic self-assembly ability within the complex TME.

Changes in the manuscript on page 39 and 40 (Methods section)

Antitumor Treatment Studies in subcutaneous xenograft tumor models

To establish BC and PC subcutaneous xenografts, 1×10^6 4T1 and PAN02 cells were implanted into the right hind limb of each BALB/c and C57BL/6 mouse, respectively.

Upon the tumor reaching an approximate volume of 50 mm³ by the 7th day post-implantation, the mice were randomized into groups. For evaluation of PAC-SABIs antitumor efficacy, the PAC-SABIs (15 mg / kg) treatment began 7 days after the tumor implantation. Each group received its respective intravenous treatment every other day for a total of four sessions. On the 28th day post-implantation, the mice were euthanized and subsequently dissected for tumor weight analysis.

To compare the antitumor efficacy of PAC-SABIs with the dual antibody combination therapy, three groups of mice received tail vein injections, with the respective treatments being IgG control, a combination of Anti-CD47 (20 mg / kg, miap301, BioXcell) and Anti-CD24 antibodies, and PAC-SABIs. On the 28th day post-implantation, the mice were euthanized and subsequently dissected for tumor weight analysis.

For evaluation of sequential PAC-SABIs and anti-PD-1 combination treatment efficacy, the mice were administered intravenously with PAC-SABIs or PBS every 2 days for a total of 4 doses, followed by anti-PD-1 (10 mg / kg) or IgG control intraperitoneal injection on days 7, 10, 13, and 16. On the 28th day post-implantation, the experiment concluded with the euthanasia of the mice. The 4T1 tumors were surgically removed and the wet weight was recorded.

Changes in the Supporting Information on page 42

Fig. 36 Comparison of in vivo antitumor effects of PAC-SABIs and dual antibody combination therapy. **a** Scheme of the therapeutic strategy for 4T1 subcutaneous tumor (Created with BioRender.com). **b** Changes in 4T1 tumor-bearing mouse body weight during treatment. The error bars represent the mean \pm SD (n = 5). **c** Bright field images of excised 4T1 tumors. **d** 4T1 tumor weights of each group after treatment. The error bars represent the mean \pm SD (n = 5; *** $p < 0.001$; the p value was analyzed by a two-tailed unpaired Student's t-test).

Comment 3: In situ self-assembly of PAC-SABIs was examined in an in vitro context, as depicted in Fig3. However, it is important to extend this analysis to the tumour microenvironment to confirm the validity of this process in TME.

Response to comment 3: Thank you for your constructive comments. Following your suggestion, we conducted scanning electron microscope (SEM) experiments to evaluate the assembly morphology of PAC-SABIs within subcutaneous 4T1 or PAN02 xenografts, as well as clinical breast cancer (BC) specimens. In brief, mice with subcutaneous 4T1 and PAN02 xenografts received intravenous injections of PAC-SABIs (15 mg / kg) every other day for a total of four treatments, and the xenografts were harvested on the day after final administration. Then, we sectioned the xenografts into slices of approximately 200 μ m, followed by fixation in 2.5 % glutaraldehyde and

a series of ethanol dehydration washes. For the freshly acquired clinical BC samples, the tissues were similarly sectioned, and the slices were incubated with PAC-SABIs in a 96-well plate at 37 °C for 120 min. Subsequent fixation and dehydration were identical to that of the xenografts. SEM imaging at various magnifications was employed to examine the in situ self-assembly morphology of PAC-SABIs. In the 4T1 and PAN02 subcutaneous xenografts and clinical BC tissues, SEM images revealed the formation of an intricate three-dimensional (3D) network scaffold orchestrated by PAC-SABIs, enveloping the surfaces of malignant cells (Supplementary Fig. 28, 29). These findings indicate that the phenomena observed are due to the enrichment and in situ self-assembly of PAC-SABIs within varied tumor microenvironments (TMEs). In future research, we aspire to further investigate the assembly mechanisms and biodistribution of PAC-SABIs in various in vivo tumor models to deepen our understanding of this complex process.

Changes in the manuscript on page 20 and 21 (Results section)

After the intravenous administration of free Cy5.5, ^{Cy5.5}SAMIs, and ^{Cy5.5}PAC-SABIs, the mice were euthanized at the 48-h mark, and the heart, liver, spleen, lung, kidney, and tumor were harvested for ex vivo imaging (Fig. 7e, f). Similar to the in vivo results, free Cy5.5 cleared rapidly from the body with negligible accumulation in the tumor and major organs. IVIS imaging showed notable differences in tissue distribution between ^{Cy5.5}SAMIs and ^{Cy5.5}PAC-SABIs. Specifically, ^{Cy5.5}PAC-SABIs exhibited distinct selectivity within tumor tissues of BC and PC, with partial retention observed in metabolic organs such as the liver and kidney. By comparison, no significant variance was observed in the biodistribution of ^{Cy5.5}SAMIs among the lungs, kidneys, and tumors, with the majority of molecules localized in the liver. The discernible dissimilarity between these two molecules could be attributed to the presence of targeted ligands and EISA, which augmented the precise recognition of ^{Cy5.5}PAC-SABIs towards cancer cells and facilitated their proficient molecular assembly. To further observe the distribution of ^{Cy5.5}PAC-SABIs in the BC and PC tissues, we analyzed the Cy5.5 fluorescence of frozen tumor sections. Interestingly, a notable

accumulation of Cy5.5 fluorescence was observed within the cancer cell regions of the 4T1 tumor, whereas the fluorescence intensity was diminished in the fibroblast-rich areas (Fig. 7g). This phenomenon was similar within the interior of PAN02 tumors, where ^{Cy5.5}PAC-SABIs exhibited a concentration within the cancer cell area rather than the paracancerous or fibroid regions (Fig. 7g). Moreover, we employed SEM imaging at various magnifications to visualize the in situ self-assembly morphology of PAC-SABIs. In the 4T1 and PAN02 subcutaneous xenografts and clinical BC tissues, SEM images revealed the formation of an intricate nanofiber network scaffold orchestrated by PAC-SABIs, enveloping the surfaces of malignant cells (Supplementary Fig. 28, 29). These results indicated that the PAC-SABIs' superstructure formation led to their effective accumulation and retention in tumors, which may provide persistent blockade of innate immune checkpoints, thereby inducing sustained macrophage-mediated immune responses.

Changes in the manuscript on page 36 and 37 (Methods section)

SEM Analysis

4T1 and PAN02 cells were seeded into 12-well plates containing cover glass slips at a density of 1×10^5 cells / well and cultured overnight. PAC-SABIs were added into each well at the final concentration of 100 μ M and cultured for 120 min. Then, PAC-SABIs-treated 4T1 and PAN02 cells were dehydrated for the time course of 24 h, processing from 30 % to 100 % ethanol. The samples were then sputter coated with gold (CRC-150 Sputter Coater, USA) and imaged using a Zeiss Ultra 55 SEM (Carl Zeiss, Germany).

For SEM imaging of the PAC-SABIs assembled morphologies within the subcutaneous 4T1 and PAN02 xenografts, the tumor-bearing mice received intravenous injections of PAC-SABIs (15 mg / kg) every other day for a total of four treatments, and the xenografts were harvested on the day after final administration. The xenografts were cut into approximately 200 μ m thin slices, then fixed in 2.5 % glutaraldehyde and dehydrated through a series of ethanol washes, and examined by SEM (Carl Zeiss, Germany). Regarding the freshly acquired clinical BC samples, the tissues were

similarly sectioned. The slices were incubated with PAC-SABIs in a 96-well plate at 37 °C for 120 min. Subsequent fixation and dehydration were identical to that of the xenografts, and SEM (Carl Zeiss, Germany) imaging was performed.

Changes in the Supporting Information on page 33

Fig. 28 SEM images of PAC-SABIs accumulation within subcutaneous 4T1 and PAN02 xenograft. The red arrows point to the PAC-SABI nanofiber network scaffold. Scale bar: 4 μm.

Changes in the Supporting Information on page 34

Fig. 29 SEM images of PAC-SABIs accumulation within clinical BC samples. The red arrows point to the PAC-SABI nanofiber network scaffold. Scale bar: 2 μm .

Comment 4: The authors have showed PAC-SABIs induced strong phagocytosis of cancer cells in vitro. A macrophage-depletion assay (by anti-CSF1R antibody or clodronate) should be performed in the in vivo models to verify the antitumor effects of PAC-SABIs are attributable to enhanced tumor cells phagocytosis by macrophages.

Response to comment 4: Thank you for your valuable comments. In the original manuscript, it was demonstrated through flow cytometry and confocal laser scanning microscopy (CLSM) experiments that PAC-SABIs effectively trigger robust phagocytic responses in macrophages against cancer cells, both in vitro and in vivo. We agree that incorporating macrophage depletion assays in an in vivo model is essential to further validate that the antitumor effects of PAC-SABIs are predominantly due to enhanced phagocytosis of cancer cells by macrophages.

For the macrophage depletion experiments, we treated BALB/c mice intraperitoneally with either 400 μg of anti-CSF1R antibody (clone AFS98, BioXcell) per mouse or PBS (as a vehicle control) three times weekly for two weeks. Successful tissue resident macrophage depletion was confirmed prior to tumor engraftment by peritoneal lavage and flow cytometry analysis. The macrophage-depleted and vehicle treated mice were then randomized and subsequently engrafted with 4T1-Luc cells,

followed by administration of PAC-SABIs as previously described. Based on the bioluminescence (BLI) findings, TAM depletion did not significantly alter the burden of 4T1-Luc tumors, while loss of TAMs largely abrogated the reduction of tumor proliferation observed in the PAC-SABIs group. Furthermore, it is worth noting that vehicle treated mice receiving PAC-SABIs demonstrated a significant survival benefit in comparison to other treatment groups (Fig. 8i-n). These results strongly suggest that the observed reduction in tumor burden is attributable to the enhanced phagocytosis of cancer cells by TAMs stimulated by PAC-SABIs.

Changes in the manuscript on page 24 and 25 (Results section)

To mimic the natural progression of human BC and PC, the orthotopic BC and PC models were then constructed for in vivo therapeutic efficacy evaluation. According to the depicted illustration (Fig. 8a, e), after the establishment of in situ BC and PC, control IgG, anti-CD24, SAMIs and PAC-SABIs were administered every other day for a total of 4 doses. The evaluation of tumor regression/progression was conducted through the periodic monitoring of tumor bioluminescence (BLI) at 7-day intervals. Over time, on the 22nd day post-treatment, the PAC-SABIs group exhibited significantly reduced tumor BLI signal intensities in comparison to the other groups, suggesting a notable deceleration in tumor growth among the BC and PC mice subjected to PAC-SABIs treatment (Fig. 8). Conversely, mice treated with Control IgG and anti-CD24 mAb exhibited limited therapeutic efficacy, as the majority of BC and PC mice succumbed within 40 and 50 days, respectively (Fig. 8d, h). The overall survival rate of BC mice treated with PAC-SABIs on the 50th day and that of PC mice on the 60th day were 60 % and 40 %, respectively, showing a significant survival benefit of PAC-SABIs (Fig. 8d, h). As shown in Fig. 8i-n, macrophage depletion assays were also conducted within an in vivo 4T1 model, and the results reinforced the validation of the above antitumor effect of PAC-SABIs being primarily linked to the heightened phagocytosis of cancer cells by macrophages.

Changes in the manuscript on page 41 (Methods section)

In vivo macrophage depletion treatment study

BALB/c mice were intraperitoneally treated with 400 µg CSF1R antibody (clone AFS98, BioXcell) per mouse or PBS (vehicle) thrice a week for two weeks. Successful tissue resident macrophage depletion was confirmed prior to tumor engraftment by peritoneal lavage and flow cytometry analysis. The macrophage-depleted and vehicle treated mice were then randomized and subsequently engrafted with 4T1-Luc cells, followed by administration of PAC-SABIs as previously described. BLIs were detected every 7 days to monitor the tumor growth in each treatment group. Survival (in days) of mice in the different treatment groups were monitored throughout the period of study.

Changes in the manuscript on page 25 and 26 (Results section)

Fig. 8 Antitumor efficacy of PAC-SABIs in vivo. **a** Scheme of the PAC-SABIs therapeutic strategy for 4T1 orthotopic tumor (Created with BioRender.com). **b** BLIs of 4T1 orthotopic tumor-bearing mice on days 7, 14, 21, and 28. **c** Quantification analysis of 4T1 orthotopic tumor BLI signals in each treatment group. The error bars represent the mean \pm SD ($n = 5$; $** p < 0.01$, $*** p < 0.001$; the p value was analyzed by a two-tailed unpaired Student's t-test). **d** Kaplan–Meier survival curves of 4T1 orthotopic tumor-bearing mice treated with the indicated formulation ($n = 5$; $** p < 0.01$; the p value was analyzed by log-rank test). **e** Scheme of the PAC-SABIs therapeutic strategy for PAN02

orthotopic tumor (Created with BioRender.com). **f** BLIs of PAN02 orthotopic tumor-bearing mice on days 7, 14, 21, and 28. **g** Quantification analysis of PAN02 orthotopic tumor BLI signals in each treatment group. The error bars represent the mean \pm SD (n = 5; ** $p < 0.01$, *** $p < 0.001$; the p value was analyzed by a two-tailed unpaired Student's t-test). **h** Kaplan–Meier survival curves of PAN02 orthotopic tumor-bearing mice treated with the indicated formulation (n= 5; * $p < 0.05$; the p value was analyzed by log-rank test). **i** Scheme of the macrophage depletion and therapeutic strategy for 4T1 subcutaneous tumor (Created with BioRender.com). **j** Representative flow cytometry plots of tissue-resident macrophages. **k** Quantification analysis of tissue-resident macrophages. Boxplots depict mean and range (n = 4 independent experiments; *** $p < 0.001$; the p value was analyzed by a two-tailed unpaired Student's t-test). **l** In vivo BLIs of 4T1 subcutaneous tumor-bearing mice on days 7, 14, 21, and 28. **m** Quantification analysis of BLI signal of 4T1 subcutaneous tumor in each treatment group. The error bars represent the mean \pm SD (n = 5; * $p < 0.05$, ** $p < 0.01$; the p value was analyzed by a two-tailed unpaired Student's t-test). **n** Kaplan–Meier survival curves of 4T1 subcutaneous tumor-bearing mice treated with the indicated formulation (n= 5; ** $p < 0.01$; the p value was analyzed by log-rank test).

Comment 5: Fig.2H, the expression of Sirpa and Siglec10 should be investigated.

Response to comment 5: We appreciate your valuable comments and concur with the imperative of delving deeper into the expression profiles of SIRP α and Siglec-10. We have meticulously re-evaluated the single-cell RNA sequencing data derived from the primary breast cancer (BC) specimen, with a refined focus on the distinct expression signatures of CD47, CD24, CD274, SIRP α and Siglec-10 at the individual cell level (Fig. 3g, h).

The resubmitted manuscript has included expanded findings, wherein it unveiled a pronounced overexpression of CD47 and CD24 within the cancerous cellular populations, surpassing that of CD274 (Fig. 3h). Importantly, SIRP α and Siglec-10 were predominantly concentrated within the non-cancerous cellular cohorts, with a marked expression discernible in clusters of tumor-associated macrophages (TAMs) (Fig. 3h). This discovery suggests a potential immunomodulatory interaction between cancer cells and TAMs, possibly mediated through these molecules.

Changes in the manuscript on page 10 and 11 (Results section)

To assess the impact of CD47 and CD24 signaling on the regulation of macrophage-mediated immune response against cancer, we investigated the expression of CD47 and CD24 in different tumor types. Analysis of RNA sequencing data obtained from The Cancer Genome Atlas (TCGA) revealed a significant overexpression of CD47 and CD24 in nearly all examined tumors (Fig. 3a). Furthermore, when compared to the well-established adaptive immune checkpoint PD-L1, CD47 and CD24 exhibited a consistent up-regulation in BC and PC (Fig. 3a). The levels of CD47 and CD24 expression in PC and BC were notably elevated compared to their corresponding normal tissues, respectively, while no significant variation was detected in PD-L1 expression (Fig. 3b-d). Stratification of patients based on their CD47 and CD24 expression levels demonstrated a significant correlation with improved overall survival in individuals with lower CD47 expression among PC patients and lower CD24 expression among BC patients (Fig. 3e, f). Single-cell RNA sequencing of a primary BC sample, focusing on examining the expression patterns of CD47, CD24, CD274, SIRP α and Siglec-10 at the individual cell level, revealed a prominent expression of CD47 and CD24 in cancer cells, notably higher than that of CD274 (Fig. 3g, h). Importantly, SIRP α and Siglec-10 were predominantly concentrated within the non-cancerous cellular cohorts, with a marked expression discernible in clusters of tumor associated macrophages (TAMs) (Fig. 3h). These results suggested that CD47 and CD24 may serve as potential markers specific to cancer cells. Moreover, immunofluorescence (IF) analysis of two cancer nodule sections obtained from patient 1 with BC indicated the presence of CD47 and CD24, which had a broad distribution within the cytoplasm and membrane of malignant cells (Fig. 3i, j). IF results acquired from consecutive sections of BC tissue in patient 2 showed robust CD47 and CD24 protein expression by cancer cells, indicating redundancy of the two membrane-bound “don’t-eat-me” signals (Fig. 3k, l). As illustrated in Fig. 3m, CD47 and CD24 function as innate ICs, diminishing the phagocytic action of macrophages on cancer cells in a synergistic manner. By utilizing the interface receptor ligand interaction and ALP

catalysis, PAC-SABIs can precisely identify the cancer cell membrane and establish stable assemblies. This process efficiently inhibits CD47 and CD24 signals, regulating TAM phagocytic activity and, eventually, boosting **antitumor** immune responses.

Changes in the manuscript on page 11 and 12 (Results section)

Fig. 3 Expression of CD47 and CD24 in BC and PC. **a** Heatmap showing the normalized expression of CD47, CD24, and PD-L1 in a pan-cancer cohort. **b-d** Expression of CD47, CD24 and PD-L1 in BC and PC compared to matched normal tissue by analyzing GEPIA³⁸ ($p < 0.05$; the p value was analyzed by a two-tailed unpaired Student's t-test). **e** Overall survival for PC patients ($n = 178$) with high versus low CD47 expression as defined by median (Kaplan-Meier survival analysis

was utilized and the *p* value was analyzed by log-rank test). **f** Overall survival for BC patients (n = 1070) with high versus low CD24 expression as defined by median (Kaplan-Meier survival analysis was utilized and the *p* value was analyzed by log-rank test). **g** UMAP dimension 1 and 2 plots displaying cells from a primary sample of BC, and cells colored by cluster identity (n = 20000 single cells). **h** CD44, CD47, CD24, CD274, SIRP α , and Siglec-10 expression overlaid onto UMAP space. **i, j** IF analysis of CD47 and CD24 in Patient 1. **k, l** IF analysis of CD47 and CD24 in Patient 2. **m** Schematic illustration of immune quiescent microenvironment in BC, in situ assembly of PAC-SABIs on the BC cell surface, and blockage of CD47 and CD24 phagocytic checkpoints by PAC-SABIs (Created with BioRender.com).

Comment 6: Statistical methods used for each figure should be indicated.

Response to comment 6: Thank you for your beneficial comments. The statistical methods used for each figure have been indicated in the resubmitted manuscript.

Comment 7: FigS19, BLI imaging should be quantified.

Response to comment 7: Thank you for your helpful comments. The quantitative analysis of bioluminescence (BLI) imaging in Supplementary Fig. 37 has been added to the resubmitted manuscript. The quantitative data indicated a significant reduction in BLI intensity within the PAC-SABIs group compared to other treatment groups (Supplementary Fig. 38). The results demonstrate that PAC-SABIs administration effectively suppresses the progression of liver metastasis in both breast and pancreatic cancers models.

Changes in the manuscript on page 29 (Results section)

To investigate if the ability of PAC-SABIs to inhibit cancer cell migration and invasion in vitro translates to decreased metastasis in vivo, we used an experimental liver metastasis model of BC and PC. 4T1-luc and PAN02-luc cells were injected into the spleens of BALB/c and C57BL/6 mice, respectively, which enabled monitoring and quantification of metastasized cancer cells in the liver with BLIs. Starting from the day before injection of cancer cells into the spleen, mice were given intravenous injections

of IgG control, anti-CD24 mAb, SAMIs, and PAC-SABIs every other day for a total of 4 doses. The BLI intensity started to increase soon after injection of cancer cells in the IgG control and anti-CD24 mAb groups, and patchy metastatic lesions formed in the liver region (Supplementary Fig. 37). However, a significant decrease in BLI intensity was observed in the PAC-SABIs group compared to the IgG control group after two weeks of injection (Supplementary Fig. 37, 38). Then, the mice were euthanized, and reductions in the number and size of liver metastases were confirmed macroscopically in the PAC-SABIs group (Supplementary Fig. 37c, g). Additionally, H&E staining of the liver further demonstrated that PAC-SABIs facilitated construction of nanofibrous network barriers on the cancer cell membrane, inhibiting early metastasis and effectively stopping the growth of previously formed metastasis in mouse liver metastasis models (Supplementary Fig. 37d, h).

Changes in the Supporting Information on page 44

Fig. 38 Quantification analysis of the in vivo BLI signals. a Quantification analysis of the BLI signals in BC liver metastasis after different treatments. The error bars represent the mean \pm SD (n = 3; * $p < 0.05$; the p value was analyzed by a two-tailed unpaired Student's t-test). **b** Quantification analysis of the BLI signals in PC liver metastasis after different treatments. The error bars represent the mean \pm SD (n = 3; * $p < 0.05$; the p value was analyzed by a two-tailed unpaired Student's t-test).

In answer to the comments raised by Reviewer #3

Comment 1: The authors believe that the beta-amyloid derived peptide KLVFF is the driving force behind the molecular assembly of PAC-SABI. In fact, when comparing

the size of the 5-mer peptide with the overall size and structure of the monomer, it rather looks that the amyloid-derived peptide is somehow buried within the monomer structure and might not be easily accessible to drive the self-assembly process. To convincingly prove this assumption, the amyloid motif KLVFF should be exchanged with another hydrophobic glycine or alanine 5-mer sequence leading to an almost identical micelle structure. Also, the typical cross-beta structure of amyloids should be expected to be found and an easy monitoring with fluorescent dye thioflavine T showing an intrinsic fluorescence should give an answer to the importance of the amyloid beta motif for the self-assembly of the inhibitor structure.

Response to comment 1: Thank you for your constructive comments regarding our utilization of the β -amyloid-derived peptide KLVFF (Lys-Leu-Val-Phe-Phe) in constructing PAC-SABIs. Your thoughtful analysis of the potential burial of the KLVFF peptides within the monomer structure has prompted us to provide a more detailed explanation and additional experiments to address your concerns.

In alignment with your suggestion, we substituted the KLVFF sequence with two distinct hydrophobic 5-mer peptides, namely alanine (Ala) and glycine (Gly). To circumvent the fluorescence crosstalk issues associated with Thioflavine T, we synthesized peptide molecules capped with the near-infrared dye Cy5.5, as delineated in Supplementary Fig. 1: Pep (Ala, Cy5.5-AAAAA-AWSATWSNpYWRH) and Pep (Gly, Cy5.5-GGGGG-AWSATWSNpYWRH). The molecular identification of the resultant products was confirmed in Fig. R1 (Response Letter, below).

Using transmission electron microscopy (TEM), we examined the microstructure of Pep (Cy5.5-FFVLK-AWSATWSNpYWRH), Pep (Ala), and Pep (Gly). All three peptides formed irregular aggregates without alkaline phosphatase (ALP) treatment (Fig. R2, Response Letter, below). When co-incubated with ALP for the first 30 min, the Pep aggregates underwent a notable morphological transformation, involving entanglement and elongation phases, eventually reorganizing into nanofiber structures after 60 min (Fig. R2, Response Letter, below). The other peptides, however, despite undergoing further aggregation upon exposure to ALP, did not form distinct fibrous structures in this period (Fig. R2, Response Letter, below). The results suggest that the

introduction of hydrophilic phosphate groups initiates the formation of stable micelle structures in an aqueous milieu, facilitated by amphiphilic interactions. ALP-induced dephosphorylation shifts this balance towards hydrophobicity, reducing amphiphilic interactions and prompting molecular rearrangement. Within this rearrangement, KLVFF motifs are more prone to aggregation, forming stable hydrogen bonds and β -sheet structures. The repetitive sequences of AAAAA and GGGGG, in contrast, may impede the formation of stable β -sheets and fibrous structures due to their linear uniformity.

Subsequently, to observe peptide self-assembly, we used UV-visible absorption spectroscopy to monitor turbidity changes over time. Pep demonstrated a significant turbidity increase compared to Pep (Ala) and Pep (Gly) when co-incubated with ALP over a 60-min period (PBS, pH = 7.4), indicating a rapid Pep reassembly into the desired superstructure (Fig. R3, Response Letter, below). Thioflavin T (ThT) assays confirmed the amyloid-like nature of the Pep fibrous structures noted in the TEM images. ThT fluorescence quantum yield markedly increased in Pep plus ALP group, underscoring the importance of KLVFF sequence in β -sheet formation (Fig. R4, Response Letter, below). Furthermore, wide-angle X-ray scattering (WAXS) was performed to study the molecular packing mode in the β -sheet fibrous structure. There were two distinct peaks in WAXS, in which 4.6 Å spacing corresponded to the distance between two adjacent strands in the β -sheet, and 12.7 Å spacing was relevant to two sheets in the bilayers (Fig. R5, Response Letter, below). These findings collectively affirm the specificity of KLVFF in the self-assembly process, enhancing our understanding of its function in peptide molecular architecture.

In addition, we employed TEM to visualize Pep-PEG monomers, which revealed the formation of nanoparticle clusters. A significant observation was the augmented aggregation of Pep-PEG following a 60-min exposure to ALP, leading to the emergence of well-structured and short fiber-like formations (Fig. R6a, Response Letter, below). Supporting these morphological changes, Thioflavin T (ThT) assays demonstrated an enhanced ThT fluorescence in the Pep-PEG plus ALP group, suggesting the KLVFF motif within the monomer maintained high accessibility (Fig. R6b, Response Letter,

below). This feature is crucial for promoting β -sheet folding, a key driving force in achieving our desired molecular structure.

Fig. R1 MALDI-TOF mass spectrometry. a MALDI-TOF mass spectrometry of Pep (Ala). **b** MALDI-TOF mass spectrometry of Pep (Gly).

Fig. R2 Time-dependent TEM images of Pep (Ala), Pep (Gly), and Pep. Scale bar: 200 nm.

Fig. R3 Time-dependent turbidity changes of Pep (Ala), Pep (Gly), and Pep incubated with ALP (1 U / mL) in PBS (pH = 7.4).

Fig. R4 ThT fluorescence spectrum of Pep (Ala), Pep (Gly), and Pep incubated with ALP (1 U / mL). Insert: confocal laser scanning microscopy (CLSM) images of ThT-stained Pep (Ala), Pep (Gly), and Pep, green: ThT fluorescence; scale bar: 10 μm .

Fig. R5 WAXS of Pep after incubation with ALP (1 U / mL). **a** D-spacings of 4.6 and 12.7 \AA are attributed to the spacing of the adjacent strands and laminates, respectively. **b** Proposed antiparallel β -sheet molecular arrangement of Pep (Created with BioRender.com).

Fig. R6 Characterization of Pep-PEG. **a** TEM images of Pep-PEG and Pep-PEG plus ALP. Scale bar: 200 nm. **b** ThT fluorescence spectrum of Pep-PEG and Pep-PEG plus ALP (1 U / mL). Insert: confocal laser scanning microscopy (CLSM) images of ThT-stained Pep-PEG, green: ThT fluorescence; scale bar: 10 μ m.

Comment 2: Was the known fact of amyloids' self-assembly features the rationale beta-amyloid peptide motif KLVFF was used?

Response to comment 2: We appreciate your valuable comments and acknowledge the selection of this particular motif is indeed informed by its intrinsic properties related to amyloid self-assembly, as elucidated through a series of scientific discoveries and research milestones. Alzheimer's disease (AD) is characterized pathologically by the progressive intracerebral accumulation of β -amyloid ($A\beta$) peptides¹. These peptides appear to be unstructured in their monomer state but aggregate to form fibrils with an ordered cross- β -sheet pattern²⁻⁵. Because $A\beta$ is self-assembling, one possible strategy to prevent this process is to use short peptide fragments homologous to the full-length wild-type protein as inhibitors⁶⁻⁸. In 1996, Tjernberg et al. identified $A\beta_{16-20}$ (KLVFF), which binds to full-length $A\beta$ and prevents assembly into fibrils⁶. The intrinsic proclivity of KLVFF sequence to form β -sheet structures has been established through various biophysical techniques, including circular dichroism (CD), Fourier-transform infrared (FTIR) spectroscopy, and fiber X-ray diffraction⁹⁻¹². These researches revealed that KLVFF readily adopts β -sheet conformations and assembles into fibrils, demonstrating features characteristic of amyloid aggregates.

The intrinsic amyloidogenic characteristics of KLVFF, resembling those observed in full-length $A\beta$ peptides, render it an exemplary model for developing innovative

peptide-based therapeutics with broad applications. Current researches on KLVFF-containing peptides predominantly concentrated on the utilization in cancer treatment, collectively highlighting their versatility and efficacy in targeting various aspects of tumor progression. These peptides enabled the formation of well-defined and stable nanofibers or nanoparticles through hydrogen bonding and hydrophobic forces, offering benefits in terms of drug delivery, biocompatibility, and controlled release¹³⁻¹⁸. Moreover, the distinctive morphological transformation of KLVFF-containing peptides into nanofibrillar networks upon administration enhanced drug retention and penetration within the tumor microenvironment (TME), addressing key challenges in targeted cancer therapy¹⁹⁻²⁴. On another front, KLVFF-containing peptides exhibited considerable effectiveness in the realm of tumor imaging²⁵. Their intrinsic properties contributed to a broader imaging window that provided insights into tumor progression dynamics^{26,27}. The integration of an active targeting mechanism with an assembly-induced retention effect significantly improved imaging specificity, allowing for a precise visualization of the tumor site^{26,28}. Furthermore, recent biomedical advancements emphasized the potential applications of KLVFF-containing peptides in addressing critical challenges related to neovascularization and coagulation^{29,30}. As demonstrated, the versatility of KLVFF-containing peptides makes them promising candidates for novel therapeutic interventions in diverse medical fields, marking a significant advancement in peptide-based therapeutics.

Reference

1. Viet, M. H., Ngo, S. T., Lam, N. S. & Li, M. S. Inhibition of aggregation of amyloid peptides by beta-sheet breaker peptides and their binding affinity. *J. Phys. Chem. B* **115**, 7433–7446 (2011).
2. Eanes, E. D. & Glenner, G. G. X-ray diffraction studies on amyloid filaments. *J. Histochem. Cytochem.* **16**, 673–677 (1968).
3. Kirschner, D. A., Abraham, C. & Selkoe, D. J. X-ray diffraction from intraneuronal paired helical filaments and extraneuronal amyloid fibers in Alzheimer disease indicates cross-beta conformation. *Proc. Natl. Acad. Sci. U. S. A.* **83**, 503–507 (1986).
4. Petkova, A. T. *et al.* A structural model for Alzheimer's beta -amyloid fibrils based

- on experimental constraints from solid state NMR. *Proc. Natl. Acad. Sci. U. S. A.* **99**, 16742–16747 (2002).
5. Hardy, J. & Selkoe, D. J. The amyloid hypothesis of Alzheimer's disease: progress and problems on the road to therapeutics. *Science* **297**, 353–356 (2002).
 6. Tjernberg, L. O. *et al.* Arrest of beta-amyloid fibril formation by a pentapeptide ligand. *J. Biol. Chem.* **271**, 8545–8548 (1996).
 7. Soto, C., Kindy, M. S., Baumann, M. & Frangione, B. Inhibition of Alzheimer's amyloidosis by peptides that prevent beta-sheet conformation. *Biochem. Biophys. Res. Commun.* **226**, 672–680 (1996).
 8. Li, H. *et al.* Mechanistic investigation of the inhibition of Abeta42 assembly and neurotoxicity by Abeta42 C-terminal fragments. *Biochemistry* **49**, 6358–6364 (2010).
 9. Castelletto, V. *et al.* Self-Assembly and Anti-Amyloid Cytotoxicity Activity of Amyloid beta Peptide Derivatives. *Sci. Rep.* **7**, 43637 (2017).
 10. Qu, A. *et al.* The synergistic effect between KLVFF and self-assembly chaperones on both disaggregation of beta-amyloid fibrils and reducing consequent toxicity. *Chem. Commun. Camb. Engl.* **53**, 1289–1292 (2017).
 11. Enache, T. A., Chiorcea-Paquim, A.-M. & Oliveira-Brett, A. M. Amyloid Beta Peptide VHHQ, KLVFF, and IIGLMVGGVV Domains Involved in Fibrilization: AFM and Electrochemical Characterization. *Anal. Chem.* **90**, 2285–2292 (2018).
 12. Pizzi, A., Dichiarante, V., Terraneo, G. & Metrangolo, P. Crystallographic insights into the self-assembly of KLVFF amyloid-beta peptides. *Biopolymers* (2017) doi:10.1002/bip.23088.
 13. Hu, X.-J. *et al.* An In Vivo Self-Assembled Bispecific Nanoblocker for Enhancing Tumor Immunotherapy. *Adv. Mater.* **35**, e2303831 (2023).
 14. Xu, A.-P. *et al.* Bio-inspired metal ions regulate the structure evolution of self-assembled peptide-based nanoparticles. *Nanoscale* **8**, 14078–14083 (2016).
 15. Luo, Q. *et al.* A self-destructive nanosweeper that captures and clears amyloid β -peptides. *Nat. Commun.* **9**, 1802 (2018).
 16. Li, Y.-J. *et al.* An adhesive peptide specifically induces microtubule condensation. *Mater. Horiz.* **10**, 5298–5306 (2023).

17. Wang, Z. *et al.* Addressable Peptide Self-Assembly on the Cancer Cell Membrane for Sensitizing Chemotherapy of Renal Cell Carcinoma. *Adv. Mater. Deerfield Beach Fla* **31**, e1807175 (2019).
18. An, H.-W. *et al.* A bispecific glycopeptide spatiotemporally regulates tumor microenvironment for inhibiting bladder cancer recurrence. *Sci. Adv.* **9**, eabq8225 (2023).
19. Sun, C. *et al.* Polyamine-Responsive Morphological Transformation of a Supramolecular Peptide for Specific Drug Accumulation and Retention in Cancer Cells. *Small* **17**, e2101139 (2021).
20. Wang, L. *et al.* Transformable Dual-Inhibition System Effectively Suppresses Renal Cancer Metastasis through Blocking Endothelial Cells and Cancer Stem Cells. *Small* **16**, e2004548 (2020).
21. Zhang, K. *et al.* Peptide-Based Nanoparticles Mimic Fibrillogenesis of Laminin in Tumor Vessels for Precise Embolization. *ACS Nano* **14**, 7170–7180 (2020).
22. Hu, X.-X. *et al.* Transformable Nanomaterials as an Artificial Extracellular Matrix for Inhibiting Tumor Invasion and Metastasis. *ACS Nano* **11**, 4086–4096 (2017).
23. Yang, P.-P. *et al.* Host Materials Transformable in Tumor Microenvironment for Homing Theranostics. *Adv. Mater.* **29**, (2017).
24. Fan, J.-Q. *et al.* Binding-Induced Fibrillogenesis Peptides Recognize and Block Intracellular Vimentin Skeletonization against Breast Cancer. *Nano Lett.* **21**, 6202–6210 (2021).
25. Luo, Q. *et al.* In Vivo Anchoring Bis-Pyrene Probe for Molecular Imaging of Early Gastric Cancer by Endoscopic Techniques. *Adv. Sci.* **10**, e2203918 (2023).
26. Ren, H. *et al.* A bioactivated in vivo assembly nanotechnology fabricated NIR probe for small pancreatic tumor intraoperative imaging. *Nat. Commun.* **13**, 418 (2022).
27. Zhao, X.-X. *et al.* In Situ Self-Assembled Nanofibers Precisely Target Cancer-Associated Fibroblasts for Improved Tumor Imaging. *Angew. Chem. Int. Ed Engl.* **58**, 15287–15294 (2019).
28. Hou, D.-Y. *et al.* An activated excretion-retarded tumor imaging strategy towards metabolic organs. *Bioact. Mater.* **14**, 110–119 (2022).

29. Zhang, K. *et al.* A Monotargeting Peptidic Network Antibody Inhibits More Receptors for Anti-Angiogenesis. *ACS Nano* **15**, 13065–13076 (2021).

30. Yang, P.-P. *et al.* A biomimetic platelet based on assembling peptides initiates artificial coagulation. *Sci. Adv.* **6**, eaaz4107 (2020).

Comment 3: In case, the amyloid motif is the peptide backbone that drives the assembly of the suggested micelle structure how can it be understood that the second 12-mer peptide sequence motif AWSATWSNpYWRH (Pep-20) and the attached PEG 2000 motif is not interfering with the self-assembly process.

Response to comment 3: Thank you for your valuable comments. The intricate assembly process of peptide molecules, governed by intermolecular interactions, spatial configurations, and temporal dynamics, is predominantly determined by their intrinsic properties. Consequently, in our modular design of Pep-PEG, the integration of 12-mer peptide sequence motif and PEG₂₀₀₀ is anticipated to alter its molecule structures, potentially leading to novel effects on the assembly mechanism.

In an effort to gain deeper insights into the assembly process of Pep-PEG and its intermediates, we synthesized new peptide molecules, specifically NBD-FFVLK and NBD-FFVLK-AWSATW. The molecular identification of the resultant products was confirmed in Supplementary Fig. 11, 12. We then conducted an assessment of the microscopic assembly morphology for a comprehensive array of components, including NBD-FFVLK, NBD-FFVLK-AWSATW, Pep, and Pep-PEG through transmission electron microscopy (TEM) imaging (Supplementary Fig. 13). NBD-FFVLK manifested as a fibrous network, characterized by densely interwoven fibers with a morphology similar to typical amyloid fibrils. Upon incorporating the AWSATW sequences, NBD-FFVLK-AWSATW retained its fibrous morphology, albeit forming a more loosely structured network. The introduction of additional hydrophilic segments containing phosphate groups in Pep led to quasi-circular aggregates, suggesting a new equilibrium between the β -sheet formation driven by KLVFF and amphiphilic interactions. Pep-PEG's structure resembled that of Pep, but the aggregates were enveloped in a sparser layer of material, likely due to PEGylation. These observations

illustrate that the modular design of Pep-PEG manifests changes in self-assembly forces and microstructures with varying module components. Moreover, the presence of KLVFF, ALP responsive motif and PEG₂₀₀₀, which promote β -sheet formation and amphiphilic interactions, underscore the significance of controlled self-assembly of Pep-PEG.

Changes in the manuscript on page 5 and 6 (Results section)

To test the feasibility of our concepts and investigate the assembly behavior, we synthesized the peptide molecules as depicted in Fig. 1a. The multifunctional peptide-based inhibitor employed a modular design. In the molecular structure, the fluorophores nitrobenzoxadiazole (NBD) and cyanine 5.5 (Cy5.5) were incorporated as capping groups at the ends of the peptides, and enabled real-time tracking of molecular assemblies in vitro and in vivo through fluorescence imaging. The typical self-assembled peptide, Lys-Leu-Val-Phe-Phe, with a sequence derived from Alzheimer's disease-associated β -amyloid, was the peptide backbone that accomplished molecular assembly. The hydrophilic PEG₂₀₀₀ was introduced and implemented to modify the prodrug assemblies for prolonging circulation in blood and increasing accumulation in tumor. Pep-20 (Ala-Trp-Ser-Ala-Thr-Trp-Ser-Asn-Tyr-Trp-Arg-His) incorporated in our molecular design was identified to specifically bind to both human and murine CD47 for blockage of CD47/SIRP α interaction³⁷. The peptide monomers were assembled into micelles in the aqueous solution, which were subsequently connected to the anti-CD24 monoclonal antibody (mAb). The linkage allowed for active targeting of CD24 overexpressing cancer cells, thereby exerting a synergistic macrophage immunomodulation (Fig. 1b). Moreover, the tyrosine of Pep-20 enabled the synthesis of phosphorylated analogs (pTyr) to promote in situ rearrangement and growth of molecular assemblies, leading to the formation of final PAC-SABIs upon ALP mediated dephosphorylation (Fig. 1c, d). Expanding upon our supramolecular platform, we engineered additional supramolecular assembled CD47 mono-target inhibitors (SAMIs) to validate the efficacy of PAC-SABIs. Solid-phase peptide synthesis and liquid synthesis were employed, and the detailed procedures were described in Supplementary

Fig. 1, 2. The molecular identification of the intermediates and final products were confirmed in Supplementary Fig. 3-12.

Changes in the manuscript on page 7 and 8 (Results section)

Subsequently, we conducted an extensive assessment of the micromorphology of Pep-PEG and their intermediates utilizing transmission electron microscopy (TEM). The TEM imaging results, as illustrated in Supplementary Fig. 13, revealed that the modular design of Pep-PEG manifested distinct variations in self-assembly dynamics and microstructure corresponding to changes in module components. Notably, both Pep and PAC-SABIs displayed time-dependent morphological transformations under ALP co-incubation (Fig. 2i). Pep exhibited a quasi-circular aggregate, wherein these peptide assemblies subsequently aggregated and manifested a spherical structure after a 10-min incubation period with ALP. Then, the advent of a short fiber structure was observed at 30 min. By the 60-min mark, the majority of the spherical structure was dissipated, giving way to a substantial presence of fibrous structures. At the 120-min interval, the entire assembly system was predominantly occupied by characteristic nanofibers with a diameter of 6.2 ± 0.7 nm (Fig. 2i). However, PAC-SABIs were observed as irregular aggregates. During the initial 30 min of enzyme catalyzed reactions, these aggregates rapidly coalesced into entangled nanofibers. Subsequently, these nanofibers underwent further extension and bundling, resulting in the formation of a three-dimensional (3D) network after 60 min. At 120 min, nearly all PAC-SABIs transformed into a larger network scaffold with interlaced nanofibers presenting a diameter measuring 13.2 ± 2.9 nm (Fig. 2i). TEM analysis showed that the final morphology of PAC-SABIs was closely related to the proportion of peptide molecules in the system. At a mAb: peptide ratio of 1: 1, the TEM image depicted large, irregular aggregates. Progressing to a ratio of 1: 5, these aggregates appeared denser and more branched, while at a ratio of 1: 10, they evolved into a fibrous network structure, distinguished by the high degree of cross-linking and framework complexity (Supplementary Fig. 14). ThT assays further confirmed the critical role of primary peptide components in forming β -sheet structures

within the desired PAC-SABIs system (Supplementary Fig. 15). The notable increase in β -sheet content, characterized by parallel or antiparallel amino acid chains stabilized by hydrogen bonds, facilitated stacking conducive to fiber morphology assembly. Moreover, SEM imaging offered a more comprehensive view and surface feature information of PAC-SABIs, implying their potential for 3D network scaffold formation via in situ self-assembly (Supplementary Fig. 16).

Changes in the Supporting Information on page 15

Fig. 11 MALDI-TOF mass spectrometry of NBD-FFVLK.

Changes in the Supporting Information on page 16

Fig. 12 MALDI-TOF mass spectrometry of NBD-FFVLK-AWSATW.

Changes in the Supporting Information on page 18

Fig. 13 TEM images of NBD-FFVLK, NBD-FFVLK-AWSATW, Pep, Pep-PEG. Scale bar: 200 nm.

Comment 4: Assuming the Co-assemblies of the antibodies with the micelles are occurring as proposed, the final step of alkaline phosphatase (ALP)-driven formation of PAC-SABIs and Pep fibers remain a mystery. The suggestion is made that phosphorylated tyrosine pTyr residues promote the nanofiber assemblies. Both structures Pep and PAC-SABIs are significant different in their molecular structure and

composition, but the nanofiber formation interestingly shows similarities. Can this be explained?

Response to comment 4: Thank you for your insightful comments. As detailed in the original manuscript, our initial investigation delved into the alterations of Zeta potential in peptide-antibody combinations at varied ratios in vitro, leading to a notable observation: the Zeta potential progressively diminished in tandem with the increased binding of the negatively charged anti-CD24 antibody proteins. A critical point was observed when the ratio of monoclonal antibody (mAb) to peptide molecules achieved a 1:10 scale. At this point, the PAC-SABIs displayed a Zeta potential of -11.7 ± 0.8 mV (Fig. 2d). Remarkably, this potential remained consistent, unaffected by further increments in peptide concentration (Supplementary Table 1). Given the substantially lower molecular weight of monomeric peptides relative to that of antibody proteins, this ratio resulted in a PAC-SABIs system enriched with peptide molecules. These molecules are presumed to be instrumental in fostering the development of a fibrous network framework throughout the system, a phenomenon propelled by the extensive self-assembly forces of the peptides.

To prove this hypothesis, we analyzed the co-assembly structures of PAC-SABIs at various peptide-to-antibody ratios post a 60-min co-incubation with ALP utilizing transmission electron microscopy (TEM) imaging (Supplementary Fig. 14). The antibody proteins predominantly exhibited a dispersed morphology, typified by nanoparticles or larger aggregates. Intriguingly, as the peptide-to-antibody ratio escalated, an increase in connections between nanoparticles was observed, giving rise to more expansive and complex structures. At a mAb: peptide ratio of 1: 1, the TEM image depicted large, irregular aggregates. Progressing to a ratio of 1: 5, these aggregates appeared denser and more branched, while at a ratio of 1: 10, they evolved into a fibrous network structure, distinguished by its high degree of cross-linking and framework complexity (Supplementary Fig. 14). These TEM images provide an intuitive illustration of the impact of varying peptide molecule proportions on the nanostructural formation of PAC-SABIs under ALP action. Furthermore, Thioflavin T (ThT) assays were conducted on these varied systems. The results showed that a

significant uptick in fluorescence quantum yield was observed exclusively in the Pep plus ALP and PAC-SABIs plus ALP (mAb: peptide = 1: 10) groups (Supplementary Fig. 15). This underscores the critical role of the primary peptide components in the self-assembly and formation processes within the final PAC-SABIs system. Additionally, it suggests a considerable degree of compositional similarity between PAC-SABIs and Pep, validating the observed similarity in fiber morphology as a logical outcome.

Changes in the manuscript on page 8 and 9 (Results section)

Subsequently, we conducted an extensive assessment of the micromorphology of Pep-PEG and their intermediates utilizing transmission electron microscopy (TEM). The TEM imaging results, as illustrated in Supplementary Fig. 13, revealed that the modular design of Pep-PEG manifested distinct variations in self-assembly dynamics and microstructure corresponding to changes in module components. Notably, both Pep and PAC-SABIs displayed time-dependent morphological transformations under ALP co-incubation (Fig. 2i). Pep exhibited a quasi-circular aggregate, wherein these peptide assemblies subsequently aggregated and manifested a spherical structure after a 10-min incubation period with ALP. Then, the advent of a short fiber structure was observed at 30 min. By the 60-min mark, the majority of the spherical structure was dissipated, giving way to a substantial presence of fibrous structures. At the 120-min interval, the entire assembly system was predominantly occupied by characteristic nanofibers with a diameter of 6.2 ± 0.7 nm (Fig. 2i). However, PAC-SABIs were observed as irregular aggregates. During the initial 30 min of enzyme catalyzed reactions, these aggregates rapidly coalesced into entangled nanofibers. Subsequently, these nanofibers underwent further extension and bundling, resulting in the formation of a three-dimensional (3D) network after 60 min. At 120 min, nearly all PAC-SABIs transformed into a larger network scaffold with interlaced nanofibers presenting a diameter measuring 13.2 ± 2.9 nm (Fig. 2i). TEM analysis showed that the final morphology of PAC-SABIs was closely related to the proportion of peptide molecules in the system. At a mAb: peptide

ratio of 1: 1, the TEM image depicted large, irregular aggregates. Progressing to a ratio of 1: 5, these aggregates appeared denser and more branched, while at a ratio of 1: 10, they evolved into a fibrous network structure, distinguished by the high degree of cross-linking and framework complexity (Supplementary Fig. 14). ThT assays further confirmed the critical role of primary peptide components in forming β -sheet structures within the desired PAC-SABIs system (Supplementary Fig. 15). The notable increase in β -sheet content, characterized by parallel or antiparallel amino acid chains stabilized by hydrogen bonds, facilitated stacking conducive to fiber morphology assembly. Moreover, SEM imaging offered a more comprehensive view and surface feature information of PAC-SABIs, implying their potential for 3D network scaffold formation via in situ self-assembly (Supplementary Fig. 16).

Changes in the Supporting Information on page 19

Fig. 14 TEM images of anti-CD24 mAb, anti-CD24 mAb plus ALP, PAC-SABIs plus ALP (mAb: peptide = 1: 1), PAC-SABIs plus ALP (mAb: peptide = 1: 5), PAC-SABIs plus ALP

(mAb: peptide = 1: 10). Scale bar: 200 nm.

Changes in the Supporting Information on page 20

Fig. 15 ThT fluorescence spectrum of anti-CD24 mAb, Pep, PAC-SABIs, Pep plus ALP, PAC-SABIs plus ALP (mAb: peptide = 1: 1), PAC-SABIs plus ALP (mAb: peptide = 1:10).

Comment 5: It is mentioned that a Michael addition is happening. A more profound explanation of the proposed chemical reaction would be desired.

Response to comment 5: Thank you for your beneficial comments. The Michael addition is a fundamental reaction in organic chemistry, characterized by the nucleophilic addition of a carbanion to an α , β -unsaturated carbonyl compound. This reaction is significant for its ability to form carbon-carbon bonds in a selective and controlled manner, essential for antibody modification. The textual explanations and diagrams illustrating the reaction mechanism have been included in the resubmitted manuscript (Supplementary Fig. 2).

As depicted in Supplementary Fig. 2, thiol groups are typically situated in specific areas of antibody molecule, particularly within the Fc region. The sulfur atom in the thiol group, endowed with a lone pair of electrons, acted as a nucleophilic agent. Its nucleophilicity was directed towards maleimides, characterized by their electron-deficient carbon-carbon double bonds, serving as electrophilic centers. The reaction was initiated with the sulfur atom of the thiol group attacking one of the carbon atoms

of the maleimide's carbon-carbon double bond. Concurrently, the electrons from the double bond were shifted towards the other carbon atom. This attack resulted in the formation of an enolate intermediate, accompanied by the creation of a new Ab-S-C (Ab, Antibody) bond. However, the enolate intermediates thus formed were often not inherently stable and underwent a series of internal electronic rearrangements. Following the stabilization of the enolate intermediate, chain transfer occurred. At this point, the original double bond was completely opened, forming a new carbon-carbon single bond. The final stage involved the formation of the stable addition product post chain transfer and electron rearrangement. This product was characterized by the newly formed carbon-sulfur bond and a new carbon-carbon single bond, marking the completion of the Michael addition process and the formation of the stable compound.

Changes in the Supporting Information on page 5 and 6

Fig. 2 The synthetic route for PAC-SABIs by Michael addition reaction.

Reaction mechanism: thiol groups are typically situated in specific areas of antibody molecule, particularly within the Fc region. The sulfur atom in the thiol group, endowed with a lone pair of electrons, acted as a nucleophilic agent. Its nucleophilicity was directed towards maleimides,

characterized by their electron-deficient carbon-carbon double bonds, serving as electrophilic centers. The reaction was initiated with the sulfur atom of the thiol group attacking one of the carbon atoms of the maleimide's carbon-carbon double bond. Concurrently, the electrons from the double bond were shifted towards the other carbon atom. This attack resulted in the formation of an enolate intermediate, accompanied by the creation of a new Ab-S-C (Ab, Antibody) bond. The enolate intermediates thus formed were often not inherently stable and underwent a series of internal electronic rearrangements. Following the stabilization of the enolate intermediate, chain transfer occurred. At this point, the original double bond was completely opened, forming a new carbon-carbon single bond. The final stage involved the formation of the stable addition product post chain transfer and electron rearrangement. This product was characterized by the newly formed carbon-sulfur bond and a new carbon-carbon single bond, marking the completion of the Michael addition process and the formation of the stable compound.

Comment 6: TEM micrographs are shown, but SEM micrographs should give an additional important information about potential scaffold formation.

Response to comment 6: Thank you for your valuable comments. As you pointed out, scanning electron microscopy (SEM) plays a crucial role in offering a more comprehensive view of the PAC-SABIs structure, enhancing the surface feature information obtained from the previously presented transmission electron microscopy (TEM) images. We employed high-resolution SEM imaging at an elevated magnification to examine the microstructure of PAC-SABIs. The findings confirmed their porous and densely packed fiber network scaffold structure, providing support for the potential in situ self-assembly of PAC-SABIs (Supplementary Fig. 16).

Changes in the manuscript on page 8 and 9 (Results section)

Subsequently, we conducted an extensive assessment of the micromorphology of Pep-PEG and their intermediates utilizing transmission electron microscopy (TEM). The TME imaging results, as illustrated in Supplementary Fig. 13, revealed that the modular design of Pep-PEG manifested distinct variations in self-assembly dynamics and microstructure corresponding to changes in module components. Notably, both Pep

and PAC-SABIs displayed time-dependent morphological transformations under ALP co-incubation (Fig. 2i). Pep exhibited a quasi-circular aggregate, wherein these peptide assemblies subsequently aggregated and manifested a spherical structure after a 10-min incubation period with ALP. Then, the advent of a short fiber structure was observed at 30 min. By the 60-min mark, the majority of the spherical structure was dissipated, giving way to a substantial presence of fibrous structures. At the 120-min interval, the entire assembly system was predominantly occupied by characteristic nanofibers with a diameter of 6.2 ± 0.7 nm (Fig. 2i). However, PAC-SABIs were observed as irregular aggregates. During the initial 30 min of enzyme catalyzed reactions, these aggregates rapidly coalesced into entangled nanofibers. Subsequently, these nanofibers underwent further extension and bundling, resulting in the formation of a three-dimensional (3D) network after 60 min. At 120 min, nearly all PAC-SABIs transformed into a larger network scaffold with interlaced nanofibers presenting a diameter measuring 13.2 ± 2.9 nm (Fig. 2i). TEM analysis showed that the final morphology of PAC-SABIs was closely related to the proportion of peptide molecules in the system. At a mAb: peptide ratio of 1: 1, the TEM image depicted large, irregular aggregates. Progressing to a ratio of 1: 5, these aggregates appeared denser and more branched, while at a ratio of 1: 10, they evolved into a fibrous network structure, distinguished by the high degree of cross-linking and framework complexity (Supplementary Fig. 14). ThT assays further confirmed the critical role of primary peptide components in forming β -sheet structures within the desired PAC-SABIs system (Supplementary Fig. 15). The notable increase in β -sheet content, characterized by parallel or antiparallel amino acid chains stabilized by hydrogen bonds, facilitated stacking conducive to fiber morphology assembly. Moreover, SEM imaging offered a more comprehensive view and surface feature information of PAC-SABIs, implying their potential for 3D network scaffold formation via in situ self-assembly (Supplementary Fig. 16).

Changes in the Supporting Information on page 21

Fig. 16 Representative SEM images of PAC-SABIs' microscopic structure. Scale bars are marked in the figures.

Comment 7: The CD spectroscopy analysis of secondary structural motifs, see Figure 1, shows a mixture of all kinds of secondary motifs such as beta sheet, beta turn, alpha helix, and random structure. It is not surprising that such a mixture of secondary structures in a single molecule would give rise to a globular structure. The question is then, if the proposed micelle is nothing else than an aggregate of globular molecule structures and not an aligned self-assembled micelle. Why is there no significant difference in the secondary motif composition when comparing Pep and Pep + ALP, or PAC-SABIs and PAC-SABIs + ALP? Supposedly, Pep and PAC-SABIs is showing globular features, whereas the presence of ALP is introducing nanofiber structures which are by fundamentally different to globular structures, i.e., fibers being anisotropic whereas globular structures are isotropic. This kind of different structural appearance would not allow a similar CD spectrum with similar secondary motif compositions.

Response to comment 7: We appreciate your thoughtful comments regarding the circular dichroism (CD) spectroscopy analysis and the interpretation of the secondary structure. Transmission electron microscopy (TEM) investigations revealed that, in the absence of alkaline phosphatase (ALP) treatment, Pep and PAC-SABIs preferentially formed molecular aggregates of quasi-circular and irregular morphologies, respectively. Remarkably, upon co-incubation with ALP, both Pep and PAC-SABIs underwent a morphological shift towards nanofibers. This notable transition from isotropic spherical structures to anisotropic fibrous conformations implied substantial alterations at the

secondary structural level, which were not reflected in the CD spectral results of our original manuscript. In light of these circumstances, following an exhaustive review and meticulous reassessment of our data, we deem it imperative to repeat CD spectroscopy and Fourier transform infrared spectroscopy (FTIR) experiments to validate the secondary structure analysis more accurately and reliably.

In the resubmitted manuscript, the CD spectrum analysis revealed prominent negative peaks at 217 nm for both Pep and PAC-SABIs under ALP co-incubation, with Pep plus ALP group additionally exhibiting a strong positive peak around 198 nm (Fig. 2e). These spectral characteristics were indicative of β -sheet formations, suggesting that ALP exposure triggered conformational transitions within Pep and PAC-SABIs, leading to an increased proportion of β -sheet content. Re-conducting the quantification using the DichroWeb server and employing the unified ridge regression-based CONTINLL algorithm, we observed a substantial increase in β -sheet content: from 11.3% to 59.3% in Pep, and from 7.1% to 47.5% in PAC-SABIs post-ALP treatment (Fig. 2f). Moreover, the FTIR results in the resubmitted manuscript showed strong absorption bands around 1629 cm^{-1} for Pep and PAC-SABIs in interaction with ALP (Fig. 2g). These bands, attributable to the stretching vibrations of C=O and C-N groups, indicated the occurrence of a high β -sheet structure content. The notable increase in β -sheet content, characterized by parallel or antiparallel amino acid chains stabilized by hydrogen bonds, facilitates stacking conducive to fiber morphology assembly. The CD spectra and FTIR results corroborate the link between secondary structural changes and the morphological transformation observed in Pep and PAC-SABIs under ALP's catalytic influence. Thank you again for your incisive comments, which have not only enriched our comprehension of the molecular assembly process in PAC-SABIs but also refined the precision of our research findings.

Changes in the manuscript on page 7 and 8 (Results section)

To accurately monitor the secondary structure transformation of the peptide molecules, we conducted circular dichroism (CD) spectroscopy and Fourier transform infrared spectroscopy (FTIR) analysis. **The CD spectrum analysis revealed prominent**

negative peaks at 217 nm for both Pep and PAC-SABIs under ALP co-incubation, with Pep plus ALP group additionally exhibiting a strong positive peak around 198 nm (Fig. 2e). These spectral characteristics were indicative of β -sheet formations, suggesting that ALP exposure triggered conformational transitions within Pep and PAC-SABIs, leading to an increased proportion of β -sheet content. Subsequent quantitative analysis of the secondary structures underscored this transformation in the ALP-dependent secondary structure of Pep and PAC-SABIs. Specifically, we observed a substantial increase in β -sheet content: from 11.3% to 59.3% in Pep, and from 7.1% to 47.5% in PAC-SABIs post-ALP treatment (Fig. 2f). The FTIR results showed strong absorption bands around 1629 cm^{-1} for Pep and PAC-SABIs in interaction with ALP (Fig. 2g). These bands, attributable to the stretching vibrations of C=O and C-N groups, indicated the occurrence of a high β -sheet structure content. Then, high-performance liquid chromatograph (HPLC) analysis was carried out to monitor the kinetics of EISA. Although the PEG₂₀₀₀ and mAb modification might produce steric hindrance for molecular recognition and ALP reaction, Pep, Pep-PEG, and PAC-SABIs all displayed high conversion efficiency, with a conversion rate of approximately 90% at 60 min (Fig. 2h). These results demonstrated that ALP mediated enzymatic reactions acted as a trigger for initiating the rearrangement of peptide molecules, promoting the rapid formation of ordered superstructures.

Changes in the manuscript on page 9 and 10 (Results section)

Fig. 2 Characterization of PAC-SABIs. **a** CACs of Pep and Pep-PEG. **b** Fluorescence spectrum of NBD-labeled Pep. Insert: fluorescence emission image of NBD-labeled Pep and Pep plus ALP (1 U / mL). **c** Fluorescence spectrum of NBD-labeled Pep-PEG. Insert: fluorescence emission image of NBD-labeled Pep-PEG and Pep-PEG plus ALP (1 U / mL). **d** Zeta potentials of Pep, Pep-PEG and PAC-SABIs detected by dynamic light scattering. **e** CD spectrum of Pep and PAC-SABIs in the presence or absence of ALP (1 U / mL). **f** Secondary structure calculation of Pep and PAC-SABIs in the presence or absence of ALP (1 U / mL). **g** FTIR spectra of Pep and PAC-SABIs in the presence or absence of ALP (1 U / mL). **h** Conversion rates of Pep, Pep-PEG and PAC-SABIs incubated with ALP (1 U / mL) over time. **i** Time-dependent TEM images of Pep and PAC-SABIs. The red arrows point to the formation of nanofibers. Scale bar: 200 nm.

REVIEWER COMMENTS

Reviewer #1 (Remarks to the Author):

The paper “Enzyme-Instructed Peptide Self-Assembly as A Cell Membrane Lichen Activating Macrophage-Mediated Cancer Immunotherapy” by Zhang et al. report an assembled CD47 and CD24 bi-target inhibitors (PAC-SABIs) that simultaneously blocking CD47 and CD24 signaling, and demonstrated promising macrophage-dependent inhibition of tumor growth and metastasis in vivo. Although there have been several similar studies on the use of enzyme-instructed peptides for cancer immunotherapy, the scientific views and data support in this article are very abundant, and the medical problem that the author wants to surmount is also representative. Although the article has a unique idea and sufficient data, there have been many similar studies and the quality of the data presented was not satisfactory. I think that this paper cannot meet the requirements of “Nature communication” journals, so further improvement is needed.

1. In all the presented data, the SAMIs treatment was more effective than anti-CD24 or PD-1 group (Fig. 5d-5f, 6d; Fig. S35, S38, S39, S40). We can even notice that there was no difference between the anti-CD24 treatment and the control group, especially in terms of survival benefits in the mice (Fig. 8a-8f). Since the phagocytosis effect mediated by the CD24-Siglec10 axis is also very important and the affinity of peptides (SAMIs) is not as good as that of monoclonal antibody (anti-CD24). More importantly, Barkal et al. (Nature 2019, 572, 392) had proved that anti-CD24 therapy has a certain tumor therapeutic effect, so the authors need to explain why this experimental phenomenon occurs and explained it in detail in the discussion section.
2. Fig. 7e and 7f lack quantitative analysis results.
3. The Living image results of mice in Fig. 8 need to show all tumor-bearing mice.
4. The cytokine levels in the plasma of mice cannot fully reflect the changes in the tumor immune microenvironment after treatment, it is suggested to supplement the levels of cytokines in tumors after different treatment.
5. In Fig. 9a, the CD8 positive staining does not look significant, and the tiny positive staining seems to be an edge effect. The authors are advised to repeat the experiment to provide more representative data.

6. Since the PAC-SABIs could block the CD47 and CD24 signaling and are composed of multiple components, the authors cannot reveal changes in the immune microenvironment simply by detecting the changes of macrophages and CD8+ T cells in tumor. It is recommended to do some RNA-seq to reveal the changes in the immune transcriptomic after drug treatment of tumor cells in vivo (Adv. Mater. 2023, 35, 2306281).
7. Since the in vivo fluorescence imaging showed that the self-assembled drugs were mostly enriched in liver and kidney, the drug safety analysis should also consider the liver and kidney function of mice, and it is recommended to supplement the changes in liver and kidney function of mice under different dose concentrations.
8. There are too many figures displayed in the main text. It is recommended that the number of main figures be controlled within 7.
9. It is recommended that the experimental results and design implications of self-assembled drugs be discussed in depth in the discussion section of this article.

Reviewer #3 (Remarks to the Author):

I am satisfied with the answers to the questions and appreciate the additional experimental work and changes of the manuscript. The manuscript has improved a lot.

Title: “Enzyme-Instructed Peptide Self-Assembly as A Cell Membrane Lichen Activating Macrophage-Mediated Cancer Immunotherapy” (**Manuscript ID: NCOMMS-23-40650**)

In answer to the comments raised by Reviewer #1

Comment 1: In all the presented data, the SAMIs treatment was more effective than anti-CD24 or PD-1 group (Fig. 5d-5f, 6d; Fig. S35, S38, S39, S40). We can even notice that there was no difference between the anti-CD24 treatment and the control group, especially in terms of survival benefits in the mice (Fig. 8a-8f). Since the phagocytosis effect mediated by the CD24-Siglec10 axis is also very important and the affinity of peptides (SAMIs) is not as good as that of monoclonal antibody (anti-CD24). More importantly, Barkal et al. (Nature 2019, 572, 392) had proved that anti-CD24 therapy has a certain tumor therapeutic effect, so the authors need to explain why this experimental phenomenon occurs and explained it in detail in the discussion section.

Response to comment 1: Thank you for your insightful comments. The work by Barkal et al. demonstrated the tumor-suppressive potential of anti-CD24 monoclonal antibody therapy, wherein MCF tumor-bearing mice received an initial dose of 200 μg and were subsequently treated every other day at a dose of 400 μg for 2 weeks¹. Differing from their research strategy, the incorporation of anti-CD24 monoclonal antibodies within our PAC-SABIs system served two critical functions: they directly blocked the CD24-Siglec-10 antiphagocytic signaling pathway while simultaneously facilitated the anchoring of PAC-SABIs precursors onto the cancer cell membrane, directing them towards biomimetic and directed surface proliferation like lichens. The formation of subsequent peptide assemblies through ligand-receptor interactions could contribute to the maximization of antibody blocking efficacy. Based on this, we utilized a comparatively low dose of anti-CD24 monoclonal antibodies and confirmed its efficacy in supplying adequate PAC-SABIs to obstruct innate immune checkpoints, concurrently ensuring biosafety. In our study, we administered 15 mg / kg of PAC-SABIs bi-daily over four doses to tumor-bearing mice. Intravenous administration in both the antibody and SAMIs groups was adjusted for the respective concentrations in

PAC-SABIs. Given the substantially lower molecular weight of monomeric peptides relative to that of anti-CD24 antibody proteins, the optimal ratio of 10:1 for peptide to antibody resulted in a PAC-SABIs system enriched with peptide molecules. As detailed in the resubmitted manuscript and response to comment 4 raised by Reviewer #3, these molecules were verified to be instrumental in fostering the development of a fibrous network framework throughout the system, a phenomenon propelled by the extensive self-assembly forces of the peptides. In our regimen of lower doses, we observed no significant tumor growth inhibition or survival benefits with solely antibody therapy in 4T1 and PAN02 tumor-bearing mice.

Despite the prevailing belief in the superior affinity of antibodies over peptides, our findings revealed that SAMIs possessing self-assembly capabilities could organize into nanoarchitectonics with well-defined secondary structures. This led to the manifestation of multivalent effects derived from anti-CD47 peptide sequences, imparting the possibility of high therapeutic efficacy²⁻⁶. Furthermore, in the biodistribution of ^{Cy5.5}SAMIs, we noted a modest enrichment effect within 4T1 and PAN02 subcutaneous tumors, indicating that the self-assembly properties of SAMIs facilitated their partial localization to tumor tissues. Consequently, it was reasonable that SAMIs not only elevated macrophage phagocytic activity in vitro but also displayed substantial antitumor activity in vivo, exceeding the efficacy of anti-CD24 monoclonal antibodies. Moving forward, our research will evaluate the efficacy of various immune checkpoint inhibitors from antibody or peptide self-assembly origins in diverse tumor settings to refine our understanding concerning the comparative treatment merits of antibody versus peptide-based therapies in oncology.

Reference

1. Barkal, A. A. *et al.* CD24 signalling through macrophage Siglec-10 is a target for cancer immunotherapy. *Nature* **572**, 392–396 (2019).
2. Liu, Z. *et al.* Nanoscale optomechanical actuators for controlling mechanotransduction in living cells. *Nat. Methods* **13**, 143–146 (2016).
3. Qi, G. *et al.* Self-Assembled Peptide-Based Nanomaterials for Biomedical Imaging and Therapy. *Adv. Mater. Deerfield Beach Fla* **30**, e1703444 (2018).

4. Hortigüela, V. *et al.* Nanopatterns of Surface-Bound EphrinB1 Produce Multivalent Ligand-Receptor Interactions That Tune EphB2 Receptor Clustering. *Nano Lett.* **18**, 629–637 (2018).
5. Mamuti, M. *et al.* A Polyvalent Peptide CD40 Nanoagonist for Targeted Modulation of Dendritic Cells and Amplified Cancer Immunotherapy. *Adv. Mater. Deerfield Beach Fla* **34**, e2109432 (2022).
6. Chen, W. *et al.* Combined Tumor Environment Triggered Self-Assembling Peptide Nanofibers and Inducible Multivalent Ligand Display for Cancer Cell Targeting with Enhanced Sensitivity and Specificity. *Small Weinh. Bergstr. Ger.* **16**, e2002780 (2020).

Changes in the manuscript on page 32 and 33 (Discussion section)

Discussion

In conclusion, we have developed a CD47 and CD24 bi-target inhibitors, referred to as PAC-SABIs, that can amplify an efficacious macrophage phagocytic capacity and antitumor immune response. The incorporation of active targeting and EISA impart the PAC-SABIs with exceptional spatiotemporal control over bioactivity. The resulting nanostructured network scaffold rapidly form a dynamic lichen-mimetic cell membrane coating, and thus, stimulate the anti-cancer cell phagocytic clearance by macrophages from both mouse and human sources in vitro. Upon confirming the substantial tumor growth inhibition by PAC-SABIs in mice with BC and PC, a series of immune responses are found. These encompass macrophages heightened antitumor phagocytosis, their reorientation to an M1 phenotype, increased secretion of inflammatory cytokines, and enhanced infiltration of CD8⁺ T cells within the typically immunosuppressive environment of the 4T1 tumor. Remarkably, following treatment with PAC-SABIs, we observe an activation of associated pathways of both innate and adaptive immune responses. There is a notable upregulation of genes that serve instrumental roles in immune regulation, inflammatory processes, macrophage phagocytosis, and antigen presentation.

In addition to validating our central hypothesis, we also demonstrate that introducing the anti-CD24 monoclonal antibodies into the PAC-SABIs system propels

the subsequent in situ self-assembly of peptides via ligand-receptor interactions. In comparison with the study by Barkal et al.²⁰, our regimen involving a lower dosage of solitary antibody therapy doesn't yield a significant tumor growth inhibition or survival benefits. Nevertheless, in vitro and in vivo studies have verified its efficacy in supplying adequate PAC-SABIs to obstruct innate immune checkpoints. Besides, with the designed phosphorylated peptide precursors and refined antibody to peptide ratio, PAC-SABIs exhibit long-term stability in the physiological circulatory system and ensure pronounced accumulation and retention in tumor sites. The negligible toxic side effects associated with CD24 and CD47 blockade therapies during PAC-SABIs treatment highlight their prospect for clinical application.

Overall, this work provides a new perspective for the innate immune therapy of solid tumor. In our preliminary exploration, the combination of PAC-SABIs with sequential anti-PD1 therapy has showed promising benefits in synergistic treatment strategies. Future studies might focus on the potential enhancement of this combination with other immunotherapeutic interventions such as varying immune checkpoint inhibitors and adoptive cell therapy. Leveraging the structural versatility of the PAC-SABIs framework, it becomes viable to incorporate new functional modules. This adaptability enables the rational design of a multifaceted modular platform, which holds the potential for tailoring therapy to individual immunotherapeutic needs.

Comment 2: Fig. 7e and 7f lack quantitative analysis results.

Response to comment 2: Thank you for your helpful comments. The quantitative analysis of ex vivo near infrared (NIR) fluorescence images of 4T1 and PAN02 tumors, as well as their corresponding major organs in Fig. 7, has been added to the resubmitted Supporting Information (Supplementary Fig. 30). Ex vivo NIR fluorescence imaging and quantitative analysis revealed that, consistent with in vivo findings, free Cy5.5 was rapidly cleared from the body with negligible accumulation in the tumors and major organs. Despite the observed retention in metabolic organs such as the liver and kidney, distinct variations in tissue distribution were noted between ^{Cy5.5}SABIs and ^{Cy5.5}PAC-SABIs. Specifically, ^{Cy5.5}PAC-SABIs achieved a marked selective enrichment in both

BC and PC tumors. In contrast, $\text{Cy}5.5$ SAMIs presented a significant reduction in tumor accumulation, displaying fluorescence intensities comparable to that of lung tissues. The discernible dissimilarity between these two molecules could be attributed to the presence of targeted ligands and enzyme-instructed self-assembly (EISA), which augmented the precise recognition of $\text{Cy}5.5$ PAC-SABIs towards cancer cells and facilitated their proficient molecular assembly.

Changes in the manuscript on page 18 (Results section)

After the intravenous administration of free $\text{Cy}5.5$, $\text{Cy}5.5$ SAMIs, and $\text{Cy}5.5$ PAC-SABIs, the mice were euthanized at the 48-h mark, and the heart, liver, spleen, lung, kidney, and tumor were harvested for ex vivo imaging (Fig. 5e, f). Ex vivo NIR fluorescence imaging and quantitative analysis revealed that, consistent with in vivo findings, free $\text{Cy}5.5$ was rapidly cleared from the body with negligible accumulation in the tumors and major organs. Despite the observed retention in metabolic organs such as the liver and kidney, distinct variations in tissue distribution were noted between $\text{Cy}5.5$ SAMIs and $\text{Cy}5.5$ PAC-SABIs (Fig. 5e, f, Supplementary Fig. 30). Specifically, $\text{Cy}5.5$ PAC-SABIs achieved a marked selective enrichment in both BC and PC tumors. In contrast, $\text{Cy}5.5$ SAMIs presented a significant reduction in tumor accumulation, displaying fluorescence intensities comparable to that of lung tissues (Supplementary Fig. 30). The discernible dissimilarity between these two molecules could be attributed to the presence of targeted ligands and EISA, which augmented the precise recognition of $\text{Cy}5.5$ PAC-SABIs towards cancer cells and facilitated their proficient molecular assembly. To further observe the distribution of $\text{Cy}5.5$ PAC-SABIs in the BC and PC tissues, we analyzed the $\text{Cy}5.5$ fluorescence of frozen tumor sections. Interestingly, a notable accumulation of $\text{Cy}5.5$ fluorescence was observed within the cancer cell regions of the 4T1 tumor, whereas the fluorescence intensity was diminished in the fibroblast-rich areas (Fig. 5g). This phenomenon was similar within the interior of PAN02 tumors, where $\text{Cy}5.5$ PAC-SABIs exhibited a concentration within the cancer cell area rather than the paracancerous or fibroid regions (Fig. 5g). Moreover, we employed SEM imaging at various magnifications to visualize the in situ self-assembly

morphology of PAC-SABIs. In the 4T1 and PAN02 subcutaneous xenografts and clinical BC tissues, SEM images revealed the formation of an intricate nanofiber network scaffold orchestrated by PAC-SABIs, enveloping the surfaces of malignant cells (Supplementary Fig. 31, 32). These results indicated that the PAC-SABIs' superstructure formation led to their effective accumulation and retention in tumors, which may provide persistent blockade of innate immune checkpoints, thereby inducing sustained macrophage-mediated immune responses.

Changes in the Supporting Information on page 35

Fig. 30 Quantification analysis of the ex vivo NIR fluorescence intensity in tumors and major organs. a Quantitative calculation of average fluorescence signal in 4T1 tumor and major organ area. The error bars represent the mean \pm SD ($n = 3$; $***p < 0.001$; the p value was analyzed by a two-tailed unpaired Student's t-test). **b** Quantitative calculation of average fluorescence signal in

PAN02 tumor and major organ area. The error bars represent the mean \pm SD ($n = 3$; $***p < 0.001$); the p value was analyzed by a two-tailed unpaired Student's t -test).

Comment 3: The Living image results of mice in Fig. 8 need to show all tumor-bearing mice.

Response to comment 3: Thank you for your thoughtful comments. The bioluminescence (BLI) images of all tumor-bearing mice in Fig. 8 have been added to the resubmitted manuscript.

Changes in the manuscript on page 23 and 24 (Results section)

Fig. 6 Antitumor efficacy of PAC-SABIs in vivo. **a** Scheme of the PAC-SABIs therapeutic strategy

for 4T1 orthotopic tumor (Created with BioRender.com). **b** BLIs of 4T1 orthotopic tumor-bearing mice on days 7, 14, 21, and 28 (n = 5). **c** Quantification analysis of 4T1 orthotopic tumor BLI signals in each treatment group. The error bars represent the mean \pm SD (n = 5; ** p < 0.01, *** p < 0.001; the p value was analyzed by a two-tailed unpaired Student's t-test). **d** Kaplan–Meier survival curves of 4T1 orthotopic tumor-bearing mice treated with the indicated formulation (n = 5; ** p < 0.01; the p value was analyzed by log-rank test). **e** Scheme of the PAC-SABIs therapeutic strategy for PAN02 orthotopic tumor (Created with BioRender.com). **f** BLIs of PAN02 orthotopic tumor-bearing mice on days 7, 14, 21, and 28 (n = 5). **g** Quantification analysis of PAN02 orthotopic tumor BLI signals in each treatment group. The error bars represent the mean \pm SD (n = 5; ** p < 0.01, *** p < 0.001; the p value was analyzed by a two-tailed unpaired Student's t-test). **h** Kaplan–Meier survival curves of PAN02 orthotopic tumor-bearing mice treated with the indicated formulation (n = 5; * p < 0.05; the p value was analyzed by log-rank test). **i** Scheme of the macrophage depletion and therapeutic strategy for 4T1 subcutaneous tumor (Created with BioRender.com). **j** Representative flow cytometry plots of tissue-resident macrophages. **k** Quantification analysis of tissue-resident macrophages. Boxplots depict mean and range (n = 4 independent experiments; *** p < 0.001; the p value was analyzed by a two-tailed unpaired Student's t-test). **l** BLIs of 4T1 subcutaneous tumor-bearing mice on days 7, 14, 21, and 28 (n = 5). **m** Quantification analysis of BLI signal of 4T1 subcutaneous tumor in each treatment group. The error bars represent the mean \pm SD (n = 5; * p < 0.05, ** p < 0.01; the p value was analyzed by a two-tailed unpaired Student's t-test). **n** Kaplan–Meier survival curves of 4T1 subcutaneous tumor-bearing mice treated with the indicated formulation (n = 5; ** p < 0.01; the p value was analyzed by log-rank test).

Comment 4: The cytokine levels in the plasma of mice cannot fully reflect the changes in the tumor immune microenvironment after treatment, it is suggested to supplement the levels of cytokines in tumors after different treatment.

Response to comment 4: We appreciate your valuable comments. The levels of various cytokines in 4T1 tumor tissues following different treatments, including tumor necrosis factor alpha (TNF- α), interferon gamma (IFN- γ), interleukin-6 (IL-6), and transforming growth factor beta (TGF- β) have been supplemented in the resubmitted Supporting Information. As illustrated in Supplementary Fig. 42, the pro-inflammatory cytokines

(TNF- α , IFN- γ , and IL-6) in both mouse plasma and tumor tissues showed a significant increase after treatment with PAC-SABIs, while a notable decrease was observed in the anti-inflammatory cytokine TGF- β . These findings strongly indicated that PAC-SABIs held the ability to counteract the anti-inflammatory properties within tumor immune microenvironment.

Changes in the manuscript on page 25 (Results section)

Flow cytometry analyses were performed on GFP⁺ 4T1 tumors to further explore how PAC-SABIs inhibition altered the immune components in TME. Consistent with the IF results, the highest level of in vivo phagocytosis by infiltrating TAMs was achieved in the PAC-SABIs group, which showed 3-fold higher than that in the IgG control group (Fig. 7b, c). As shown in Fig. 7d, e, anti-CD24 mAb, SAMIs, and PAC-SABIs treatments all potentiated the activation state of TAMs towards a pro-inflammatory antitumor phenotype, indicating a potential mechanism of innate IC blockade strategy. Moreover, flow cytometry analysis of T cells in tumor tissues treated with PAC-SABIs demonstrated a significant increase in the percentage of tumor-infiltrating CD8⁺ and CD4⁺ T cells (Fig. 7b, f). We speculated that the phagocytosis of activated macrophages could increase the cross-presentation of tumor antigen to T cells, which enhanced the effectiveness of adaptive antitumor immune response. As the main immunosuppressive cells, the number of FOXP3 regulatory T cells (Tregs) within the tumor following PAC-SABIs treatment was found to be significantly decreased (Fig. 7b, g). This reduction might contribute to the further transition of the macrophage population from a suppressive M2 phenotype to an M1 phenotype. Previous studies have shown that reprogramming the TME can impact the cytokine secretion of immune cells, resulting in heightened levels of pro-inflammatory cytokines such as tumor necrosis factor alpha (TNF- α), interferon gamma (IFN- γ), and interleukin-6 (IL-6), and reduced levels of the anti-inflammatory cytokine transforming growth factor beta (TGF- β). As illustrated in Supplementary Fig. 42, the pro-inflammatory cytokines (TNF- α , IFN- γ , and IL-6) in both mouse plasma and tumor tissues showed a significant increase after treatment with PAC-SABIs, while a notable decrease was observed in the

anti-inflammatory cytokine TGF- β . These findings strongly indicated that PAC-SABIs held the ability to counteract the anti-inflammatory properties within tumor immune microenvironment (TIME).

Changes in the Supporting Information on page 48

Fig. 42 Cytokine levels in the plasma and 4T1 tumor tissue of mice. a-d Cytokine levels (TNF- α , IFN- γ , IL-6, and TGF- β) in the plasma of mice after different treatments determined using ELISA. The error bars represent the mean \pm SD (n = 4 independent experiments; * p < 0.05, ** p < 0.01, *** p < 0.001; the p value was analyzed by a two-tailed unpaired Student's t-test). **e-h** Cytokine levels (TNF- α , IFN- γ , IL-6, and TGF- β) in the 4T1 tumor tissues after different treatments determined using ELISA. The error bars represent the mean \pm SD (n = 4 independent experiments; * p < 0.05, ** p < 0.01, *** p < 0.001; the p value was analyzed by a two-tailed unpaired Student's t-test).

Comment 5: In Fig. 9a, the CD8 positive staining does not look significant, and the tiny positive staining seems to be an edge effect. The authors are advised to repeat the experiment to provide more representative data.

Response to comment 5: Thank you for your valuable comments. Following your recommendation, we have repeated the immunofluorescence (IF) analysis shown in Fig. 7a to mitigate edge effects impacting the CD8 positive staining. The updated IF staining and internal magnification details for the 4T1 tumor sections have been included in the resubmitted manuscript. The results demonstrated a minimal presence of CD8⁺ T cells in the tumor core of the IgG control group, whereas a notable increase in the infiltration depth and number of CD8⁺ T lymphocytes was observed in the tumor tissue treated with PAC-SABIs.

Changes in the manuscript on page 24 and 25 (Results section)

To examine the regulatory effects of PAC-SABIs in vivo, the changes that occurred in the 4T1 TME were analyzed after intravenous administration. H&E staining of xenograft sections in both the IgG control and anti-CD24 mAb groups revealed that the cancer cells were densely arranged in bands and clusters, with disordered arrangement (Fig. 7a). In the PAC-SABIs group, a notable presence of inflammatory cells and tissue necrosis was observed within the tumor interstitium, showing a greater extent of inflammatory cell infiltration and tissue necrosis compared to the SAMIs group (Fig. 7a). Then, we conducted IF staining and observed a noteworthy manifestation of macrophages engaging in phagocytic and cytotoxic activities against cancer cells in the PAC-SABIs group, characterized by extensive macrophage infiltration and internalization of cancer cell components (Fig. 7a). The nuclear density within this particular region was lower than that in adjacent regions, indicating a reduction in cancer cell density. Conversely, this tumoricidal phenomenon was less frequently observed in the remaining treatment groups. In addition, the IF results demonstrated a minimal presence of CD8⁺ T cells in the tumor core of the IgG control group, whereas a notable increase in the infiltration depth and number of CD8⁺ T lymphocytes was observed in the tumor tissue treated with PAC-SABIs (Fig. 7a).

Changes in the manuscript on page 27 and 28 (Results section)

Fig. 7 Immune response in vivo. **a** Representative H&E and IF staining of cytokeratin 19 (CK19), F4/80⁺ macrophages and CD8⁺ T cells for the corresponding 4T1 tumor tissues after different treatments. Scale bars are marked in the figures. **b** Representative flow cytometry plots depicting CD45⁺CD11b⁺F4/80⁺ macrophages phagocytosis, CD11b⁺F4/80⁺CD206^{hi} macrophages, CD11b⁺F4/80⁺CD86^{hi} macrophages, CD45⁺CD3⁺CD4⁺/CD8⁺ T cells, and CD45⁺CD3⁺CD4⁺Foxp3⁺ T cells in 4T1 tumors after different treatments. **c-g** Quantification analysis of CD45⁺CD11b⁺F4/80⁺ macrophages phagocytosis, CD11b⁺F4/80⁺CD206^{hi} macrophages, CD11b⁺F4/80⁺CD86^{hi} macrophages, CD45⁺CD3⁺CD4⁺/CD8⁺ T cells, and CD45⁺CD3⁺CD4⁺Foxp3⁺ T cells in 4T1 tumors after different treatments. The error bars represent

the mean \pm SD (n = 4 independent experiments; * p < 0.05, ** p < 0.01, *** p < 0.001; the p value was analyzed by a two-tailed unpaired Student's t-test). **h** Identification of differentially expressed genes in 4T1 tumors collected from the mice after treating with IgG Control or PAC-SABIs (n = 5). **i** Number of differential genes enriched in GO term of mice treated with PAC-SABIs versus IgG Control (n = 5). **j** GO enrichment analysis of the differential pathways treated with PAC-SABIs versus IgG Control (n = 5). **k** Volcano plot for the transcriptome sequencing of 4T1 tumors treated with PAC-SABIs versus IgG Control (n = 5).

Comment 6: Since the PAC-SABIs could block the CD47 and CD24 signaling and are composed of multiple components, the authors cannot reveal changes in the immune microenvironment simply by detecting the changes of macrophages and CD8⁺ T cells in tumor. It is recommended to do some RNA-seq to reveal the changes in the immune transcriptomic after drug treatment of tumor cells in vivo (Adv. Mater. 2023, 35, 2306281).

Response to comment 6: Thank you for your insightful comments on the necessity of a multifaceted evaluation of the tumor immune microenvironment (TIME) treated with PAC-SABIs. Acknowledging PAC-SABIs' inhibitory effect on CD47 and CD24 signaling and its resultant complexity in the TIME, the RNA-seq analyses on PAC-SABIs-treated 4T1 breast cancer in vivo were performed to complement our existing data on macrophage and CD8⁺ T cell changes. As illustrated in the Fig. 7h, a total of 1276 differential genes were identified, encompassing 486 upregulated genes and 790 downregulated ones. Gene Ontology (GO) assessment elucidated that a predominant fraction of upregulated genes was substantially enriched in immune related pathways such as innate and adaptive immune responses (Fig. 7i). Notably, the three most significant pathways showed intimate connection to activation of immune response, lymphocyte mediated immunity, and phagocytosis (Fig. 7i, j). Moreover, genes central to regulatory roles in immunity, inflammatory processes, macrophage phagocytotic capacity, and antigenic presentation experienced marked upregulation subsequent to PAC-SABIs therapy (Fig. 7k). Consequently, the transcriptomic outcomes provided additional evidence to substantiate the effectiveness of PAC-SABIs' innate immune

checkpoint blockade strategy in eliciting potent antitumor immune activity.

Changes in the manuscript on page 26 (Results section)

Meanwhile, the RNA-seq analyses on PAC-SABIs-treated 4T1 breast cancer in vivo were performed for an extensive evaluation of its resultant intricate alterations in the TIME. As illustrated in the Fig. 7h, a total of 1276 differential genes were identified, encompassing 486 upregulated genes and 790 downregulated ones. Gene Ontology (GO) assessment elucidated that a predominant fraction of upregulated genes was substantially enriched in immune related pathways such as innate and adaptive immune responses (Fig. 7i). Notably, the three most significant pathways showed intimate connection to activation of immune response, lymphocyte mediated immunity, and phagocytosis (Fig. 7i, j). Moreover, genes central to regulatory roles in immunity, inflammatory processes, macrophage phagocytotic capacity, and antigenic presentation experienced marked upregulation subsequent to PAC-SABIs therapy (Fig. 7k). Consequently, the transcriptomic outcomes provided additional evidence to substantiate the effectiveness of PAC-SABIs' innate immune checkpoint blockade strategy in eliciting potent antitumor immune activity.

Changes in the manuscript on page 27 and 28 (Results section)

Fig. 7 Immune response in vivo. **a** Representative H&E and IF staining of cytokeratin 19 (CK19), F4/80⁺ macrophages and CD8⁺ T cells for the corresponding 4T1 tumor tissues after different treatments. Scale bars are marked in the figures. **b** Representative flow cytometry plots depicting CD45⁺CD11b⁺F4/80⁺ macrophages phagocytosis, CD11b⁺F4/80⁺CD206^{hi} macrophages, CD11b⁺F4/80⁺CD86^{hi} macrophages, CD45⁺CD3⁺CD4⁺/CD8⁺ T cells, and CD45⁺CD3⁺CD4⁺Foxp3⁺ T cells in 4T1 tumors after different treatments. **c-g** Quantification analysis of CD45⁺CD11b⁺F4/80⁺ macrophages phagocytosis, CD11b⁺F4/80⁺CD206^{hi} macrophages, CD11b⁺F4/80⁺CD86^{hi} macrophages, CD45⁺CD3⁺CD4⁺/CD8⁺ T cells, and CD45⁺CD3⁺CD4⁺Foxp3⁺ T cells in 4T1 tumors after different treatments. The error bars represent

the mean \pm SD (n = 4 independent experiments; * p < 0.05, ** p < 0.01, *** p < 0.001; the p value was analyzed by a two-tailed unpaired Student's t-test). **h** Identification of differentially expressed genes in 4T1 tumors collected from the mice after treating with IgG Control or PAC-SABIs (n = 5). **i** Number of differential genes enriched in GO term of mice treated with PAC-SABIs versus IgG Control (n = 5). **j** GO enrichment analysis of the differential pathways treated with PAC-SABIs versus IgG Control (n = 5). **k** Volcano plot for the transcriptome sequencing of 4T1 tumors treated with PAC-SABIs versus IgG Control (n = 5).

Changes in the manuscript on page 32 (Discussion section)

In conclusion, we have developed a CD47 and CD24 bi-target inhibitors, referred to as PAC-SABIs, that can amplify an efficacious macrophage phagocytic capacity and antitumor immune response. The incorporation of active targeting and EISA impart the PAC-SABIs with exceptional spatiotemporal control over bioactivity. The resulting nanostructured network scaffold rapidly form a dynamic lichen-mimetic cell membrane coating, and thus, stimulate the anti-cancer cell phagocytic clearance by macrophages from both mouse and human sources in vitro. Upon confirming the substantial tumor growth inhibition by PAC-SABIs in mice with BC and PC, a series of immune responses are found. These encompass macrophages heightened antitumor phagocytosis, their reorientation to an M1 phenotype, increased secretion of inflammatory cytokines, and enhanced infiltration of CD8⁺ T cells within the typically immunosuppressive environment of the 4T1 tumor. Remarkably, following treatment with PAC-SABIs, we observe an activation of associated pathways of both innate and adaptive immune responses. There is a notable upregulation of genes that serve instrumental roles in immune regulation, inflammatory processes, macrophage phagocytosis, and antigen presentation.

Comment 7: Since the in vivo fluorescence imaging showed that the self-assembled drugs were mostly enriched in liver and kidney, the drug safety analysis should also consider the liver and kidney function of mice, and it is recommended to supplement the changes in liver and kidney function of mice under different dose concentrations.

Response to comment 7: Thank you for your valuable comments. Consequent to your suggestions, we advanced our assessment of PAC-SABIs' hepatorenal toxicity by consistently tracking key blood biochemical markers such as alanine aminotransferase (ALT), aspartate aminotransferase (AST), Creatinine, and Blood Urea Nitrogen (BUN) in healthy BALB/c mice for a period of one month following intravenous injection of PAC-SABIs at differential doses. Under the treatment regime using 15 mg / kg PAC-SABIs, as well as at double and triple the initial dose concentrations, all recorded parameters were observed to be within the normal reference range (Supplementary Fig. 38), further supporting the favorable biosafety of PAC-SABIs.

Changes in the manuscript on page 21 (Results section)

Generally, the toxicity of inhibitors, especially the blood toxicity associated with CD47 blockade therapy, is a key criterion affecting clinical transformation. To evaluate the systemic toxicity of PAC-SABIs, complete blood count was obtained from healthy BALB/c mice after the intravenous injection of PAC-SABIs. The main parameters included red blood cell, hemoglobin, hematocrit, mean corpuscular hemoglobin, mean corpuscular hemoglobin concentration, mean corpuscular volume, mean platelet volume and platelets. All of these parameters were shown to be within normal reference ranges (Supplementary Fig. 37). Considering PAC-SABIs' biodistribution in the liver and kidneys, the key blood biochemical markers such as ALT, AST, creatinine and BUN in healthy BALB/c mice were monitored continuously post intravenous injection of PAC-SABIs at varying doses. Under the treatment regime using 15 mg / kg PAC-SABIs, as well as at double and triple the initial dose concentrations, all recorded parameters were maintained within the normal reference spectrum (Supplementary Fig. 38). Subsequently, following approximately one month of treatment, we collected the main organs of mice in each group, and sliced them for H&E staining. No notable histological alterations or pathological lesions were observed (Supplementary Fig. 39). These results exhibited high biocompatibility characteristics of PAC-SABIs in mice without significant systemic toxicity.

Changes in the Supporting Information on page 44

Fig. 38 Hepatorenal toxicity analyses. a-d Blood biochemical analyses including alanine aminotransferase (ALT), aspartate aminotransferase (AST), Creatinine, and Blood Urea Nitrogen (BUN) after PAC-SABIs treatment (15 mg / kg). The error bars represent the mean \pm SD (n = 3). e-h Blood biochemical analyses including ALT, AST, Creatinine, and BUN after PAC-SABIs treatment (30 mg / kg). The error bars represent the mean \pm SD (n = 3). i-l Blood biochemical analyses including ALT, AST, Creatinine, and BUN after PAC-SABIs treatment (45 mg / kg). The error bars represent the mean \pm SD (n = 3).

Comment 8: There are too many figures displayed in the main text. It is recommended that the number of main figures be controlled within 7.

Response to comment 8: Thank you for your helpful comments regarding the number of main figures in our manuscript. Following your recommendation, we have relocated Fig. 3 and Fig. 6 of our initial 10 main text figures to the Supplementary Information. Additionally, Fig. 3m has been integrated into Fig. 4 for better coherence. In adherence to the journal's guidelines, which allow for up to 10 display items (figures and/or tables) depending on the word count, our article currently includes 8 figures in the main text. Although this slightly exceeds your recommended number by one, our decision to maintain these 8 figures is rooted in their indispensable contribution to the manuscript's clarity, coherence, and scientific integrity. We hope that our modifications will meet

with your approval and understanding.

Changes in the manuscript on page 11 (Results section)

As illustrated in Fig. 3a, CD47 and CD24 function as innate ICs, diminishing the phagocytic action of macrophages on cancer cells in a synergistic manner. We hypothesize that by utilizing the interface receptor ligand interaction and ALP catalysis, PAC-SABIs can precisely identify the cancer cell membrane and establish stable assemblies. This process efficiently inhibits CD47 and CD24 signals, regulating TAM phagocytic activity and, eventually, boosting antitumor immune responses. Having determined the effective assembly of PAC-SABIs under static external conditions, we proceeded to investigate the self-assembly behavior of PAC-SABIs within BC and PC cellular environments using the NBD fluorescence. Firstly, the widespread expression of ALP in 4T1 and PAN02 subcutaneous tumor, as well as clinical BC and PC samples, was verified using the BCIP/NBT color development kit (Supplementary Fig. 18). Then, 4T1 and PAN02 cells were co-incubated with NBD-labeled PAC-SABIs at 37 °C, and changes in NBD fluorescence were directly monitored using confocal laser scanning microscopy (CLSM) imaging at different time points (Fig. 3b, c). As anticipated, the treated 4T1 and PAN02 cells exhibited a time-dependent augmentation in NBD fluorescence emission on the cellular surface. Following a 30-min co-incubation period, the fluorescence appeared as a diffuse signal surrounding the cancer cells as PAC-SABIs bound to the membrane's outer surface. Then, a prompt aggregation of fluorescent clusters on the membrane was observed at the 60-min mark. By the 120-min interval, fluorescence became uniform and encompassed the cell surface, signifying the in situ self-assembly of PAC-SABIs on cancer cell membrane (Fig. 3b, c). This observation was further supported by the findings from differential interference contrast (DIC) and CLSM imaging (Fig. 3d). As shown in Fig. 3e-g, the green fluorescence from NBD exhibited a strong co-localization with the red fluorescence from Dil dye, suggesting that a majority of PAC-SABI molecules were assembled and localized on the 4T1 cell membrane. Under identical conditions, 4T1 cells treated with SAMIs exhibited robust intracellular green fluorescence, reflecting the internalization

of SAMIs into the cells, a phenomenon commonly observed when other nanostructures are incubated with live cells (Supplementary Fig. 19). After confirming that anti-CD24 mAbs effectively disrupted the CD24-Siglecs interaction (Supplementary Fig. 20), we conducted a blocking experiment utilizing these anti-CD24 mAbs. The surface fluorescence signals of pretreated 4T1 cells were significantly reduced during dynamic incubation with PAC-SABIs, indicating that the specificity of interaction between CD24 and PAC-SABIs promoted the membrane in situ self-assembly process (Supplementary Fig. 21). Additionally, as shown in Supplementary Fig. 22, TEM revealed the morphologies of PAC-SABIs assembled on the top surface of cell membranes. We also observed the formation of superstructure networks over time on the 4T1 and PAN02 cell membranes treated with PAC-SABIs through SEM imaging. During the co-incubation process from 30 to 120 min, PAC-SABIs, like the proliferation process of lichens, first attached to the cell's adherent edge, then migrated along the cell surface, and finally formed a 3D network scaffold covering the membrane (Fig. 3h, i).

Changes in the manuscript on page 14 and 15 (Results section)

Fig. 3 In situ self-assembly of PAC-SABIs on BC and PC cell membranes. **a** Schematic illustration of tumor immune quiescent microenvironment, in situ assembly of PAC-SABIs on the cancer cell surface, and blockage of CD47 and CD24 phagocytic checkpoints by PAC-SABIs (Created with BioRender.com). **b** Time-dependent CLSM images of 4T1 cells treated with NBD-labeled PAC-SABIs. Scale bar: 20 μm . **c** Time-dependent CLSM images of PAN02 cells treated with NBD-labeled PAC-SABIs. Scale bar: 20 μm . **d** Merged DIC and fluorescence images of 4T1 cell treated with NBD-labeled PAC-SABIs for 120 min. Scale bar: 20 μm . **e** CLSM images of 4T1 cell treated with Dil dye (red) and NBD-labeled PAC-SABIs for 120 min. Scale bar: 20 μm . **f, g** Fluorescence distribution of NBD-labeled PAC-SABIs on 4T1 cell. **h** Time-dependent SEM images

of 4T1 cells treated with PAC-SABIs. The red arrows point to PAC-SABIs on cell membrane. Scale bar: 1 μ m. **i** Time-dependent SEM images of PAN02 cells treated with PAC-SABIs. The red arrows point to the PAC-SABIs on cell membrane. Scale bar: 1 μ m. **j** CLSM images of 3D 4T1 spheroids treated with $Cy5.5$ PAC-SABIs, Calcein-AM, and Hoechst 33342. Scale bar: 50 μ m. **k** CLSM images of 3D 4T1 spheroids along the z-axis position. Scale bar: 50 μ m.

Changes in the Supporting Information on page 22

Fig. 17 Expression of CD47 and CD24 in BC and PC. a Heatmap showing the normalized expression of CD47, CD24, and PD-L1 in a pan-cancer cohort. **b-d** Expression of CD47, CD24 and PD-L1 in BC and PC compared to matched normal tissue by analyzing GEPIA³⁸ ($p < 0.05$; the *p* value was analyzed by a two-tailed unpaired Student's t-test). **e** Overall survival for PC patients (n = 178) with high versus low CD47 expression as defined by median (Kaplan-Meier survival analysis was utilized and the *p* value was analyzed by log-rank test). **f** Overall survival for BC patients (n = 1070) with high versus low CD24 expression as defined by median (Kaplan-Meier survival analysis

was utilized and the p value was analyzed by log-rank test). **g** UMAP dimension 1 and 2 plots displaying cells from a primary sample of BC, and cells colored by cluster identity (n = 20000 single cells). **h** CD44, CD47, CD24, CD274, SIRP α , and Siglec-10 expression overlaid onto UMAP space. **i, j** IF analysis of CD47 and CD24 in Patient 1. **k, l** IF analysis of CD47 and CD24 in Patient 2.

Changes in the Supporting Information on page 33

Fig. 28 PAC-SABIs promote phagocytic clearance of cancer cells by HDDMs. **a** Schematic illustration of HDDMs generation and stimulation (Created with BioRender.com). **b** IF staining of CD11b and CD206 in HDDMs after stimulation with TCM for 48 h. Scale bar: 50 μ m. **c** Flow cytometry-based measurement of CD206 expression by HDDMs stimulated with TCM (blue) vs. cell medium control (red). **d** Normalized phagocytosis of MDA-MB-231 cells, in the presence of IgG control, anti-CD24 mAb, SAMIs and PAC-SABIs. The error bars represent the mean \pm SD (n = 3 donors; * p < 0.05, ** p < 0.01; the p value was analyzed by a two-tailed unpaired Student's t-test).

e Representative CLSM images of in vitro phagocytosis of MDA-MB-231 cells (pHrodo-red⁺, red) by HDDMs (Calcein-AM, green). Scale bar: 50 μm. **f** Representative 3D CLSM image reconstruction of in vitro phagocytosis of MDA-MB-231 cells (pHrodo-red⁺, red) by HDDMs (Calcein-AM; green).

Comment 9: It is recommended that the experimental results and design implications of self-assembled drugs be discussed in depth in the discussion section of this article.

Response to comment 9: We appreciate your thoughtful comments. The discussion section of the resubmitted manuscript has been expanded to provide a more comprehensive examination of the experimental results and design implications of PAC-SABIs. Thank you once again for your critical feedback, which has significantly refined the depth and quality of our work.

Changes in the manuscript on page 32 and 33 (Discussion section)

Discussion

In conclusion, we have developed a CD47 and CD24 bi-target inhibitors, referred to as PAC-SABIs, that can amplify an efficacious macrophage phagocytic capacity and antitumor immune response. The incorporation of active targeting and EISA impart the PAC-SABIs with exceptional spatiotemporal control over bioactivity. The resulting nanostructured network scaffold rapidly form a dynamic lichen-mimetic cell membrane coating, and thus, stimulate the anti-cancer cell phagocytic clearance by macrophages from both mouse and human sources in vitro. Upon confirming the substantial tumor growth inhibition by PAC-SABIs in mice with BC and PC, a series of immune responses are found. These encompass macrophages heightened antitumor phagocytosis, their reorientation to an M1 phenotype, increased secretion of inflammatory cytokines, and enhanced infiltration of CD8⁺ T cells within the typically immunosuppressive environment of the 4T1 tumor. Remarkably, following treatment with PAC-SABIs, we observe an activation of associated pathways of both innate and adaptive immune responses. There is a notable upregulation of genes that serve instrumental roles in immune regulation, inflammatory processes, macrophage

phagocytosis, and antigen presentation.

In addition to validating our central hypothesis, we also demonstrate that introducing the anti-CD24 monoclonal antibodies into the PAC-SABIs system propels the subsequent in situ self-assembly of peptides via ligand-receptor interactions. In comparison with the study by Barkal et al.²⁰, our regimen involving a lower dosage of solitary antibody therapy doesn't yield a significant tumor growth inhibition or survival benefits. Nevertheless, in vitro and in vivo studies have verified its efficacy in supplying adequate PAC-SABIs to obstruct innate immune checkpoints. Besides, with the designed phosphorylated peptide precursors and refined antibody to peptide ratio, PAC-SABIs exhibit long-term stability in the physiological circulatory system and ensure pronounced accumulation and retention in tumor sites. The negligible toxic side effects associated with CD24 and CD47 blockade therapies during PAC-SABIs treatment highlight their prospect for clinical application.

Overall, this work provides a new perspective for the innate immune therapy of solid tumor. In our preliminary exploration, the combination of PAC-SABIs with sequential anti-PD1 therapy has showed promising benefits in synergistic treatment strategies. Future studies might focus on the potential enhancement of this combination with other immunotherapeutic interventions such as varying immune checkpoint inhibitors and adoptive cell therapy. Leveraging the structural versatility of the PAC-SABIs framework, it becomes viable to incorporate new functional modules. This adaptability enables the rational design of a multifaceted modular platform, which holds the potential for tailoring therapy to individual immunotherapeutic needs.

REVIEWERS' COMMENTS

Reviewer #1 (Remarks to the Author):

Although the revised manuscript has been improved to some extent, there have been many similar studies, and the innovation of this study is not satisfactory. I still think this paper does not meet the requirements of “Nature communication”.